# Muscle-specific stress fibers give rise to sarcomeres in cardiomyocytes

Aidan M Fenix[1], Abigail C Neininger[1], Nilay Taneja[1], Karren Hyde[1], Mike R Visetsouk[2], Ryan J Garde[2], Baohong Liu[3], Benjamin R Nixon[4], Annabelle E Manalo[1], Jason R Becker[4], Scott W Crawley[5], David M Bader[1], Matthew J Tyska[1], Qi Liu[3], Jennifer H Gutzman[2], Dylan T Burnette[1]*

[1]Department of Cell and Developmental Biology, Vanderbilt University, Nashville, United States; [2]Department of Biological Sciences, Cell and Molecular Biology, University of Wisconsin Milwaukee, Milwaukee, United States; [3]Department of Biomedical Informatics, Vanderbilt University Medical Center, Nashville, United States; [4]Department of Medicine, Vanderbilt University Medical Center, Nashville, United States; [5]Department of Biological Sciences, The University of Toledo, Toledo, United States

**Abstract** The sarcomere is the contractile unit within cardiomyocytes driving heart muscle contraction. We sought to test the mechanisms regulating actin and myosin filament assembly during sarcomere formation. Therefore, we developed an assay using human cardiomyocytes to monitor sarcomere assembly. We report a population of muscle stress fibers, similar to actin arcs in non-muscle cells, which are essential sarcomere precursors. We show sarcomeric actin filaments arise directly from muscle stress fibers. This requires formins (e.g., FHOD3), non-muscle myosin IIA and non-muscle myosin IIB. Furthermore, we show short cardiac myosin II filaments grow to form ~1.5 µm long filaments that then 'stitch' together to form the stack of filaments at the core of the sarcomere (i.e., the A-band). A-band assembly is dependent on the proper organization of actin filaments and, as such, is also dependent on FHOD3 and myosin IIB. We use this experimental paradigm to present evidence for a unifying model of sarcomere assembly.
DOI: https://doi.org/10.7554/eLife.42144.001

*For correspondence:
dylan.burnette@vanderbilt.edu

**Competing interests:** The authors declare that no competing interests exist.

## Introduction

At its core, a sarcomere is composed of 'thick' myosin II filaments, and 'thin' actin filaments (*Figure 1A*) (*Au, 2004*). One sarcomere is measured from Z-line to Z-line, which contain α-actinin 2 (*Figure 1A*). The proper establishment of cardiac sarcomeres during development and their subsequent maintenance is critical for heart function. Previous studies in cultured myocytes have shown the presence of actin bundles called 'stress fiber-like structures' similar in appearance to classic stress fibers (*Dlugosz et al., 1984*). These stress fibers were often found to be close to the edge of the myocyte with sarcomeres existing further from the edge (*Rhee et al., 1994*). These studies proposed that the stress fibers served as a template for the formation of sarcomeres (*Dlugosz et al., 1984*; *Rhee et al., 1994*; *Sanger et al., 2005*). The original model that proposed this was called the Templating Model (*Dlugosz et al., 1984*), and was proposed before it was known these stress fibers contained both non-muscle and sarcomeric proteins (*Rhee et al., 1994*). Beyond non-muscle myosin IIB (NMIIB), which is present in non-muscle cells, stress fibers in muscle cells contain muscle specific proteins, such as α-actinin, tropomyosin, troponins, and tropomodulin (*Almenar-Queralt et al., 1999*; *Rhee et al., 1994*; *Sanger et al., 2005*). Each of these proteins have non-muscle paralogs, which likely serve similar functions (*Bryce et al., 2003*; *Colpan et al., 2013*; *Côté, 1983*; *Gunning et al., 2015*; *Lim et al., 1986*; *Sjöblom et al., 2008*). Partly in response to the presence of

muscle specific proteins in stress fibers, the Templating Model was modified to the 'Pre-Myofibril Model' (*Rhee et al., 1994*; *Sanger et al., 2005*). Even though these models have different names and are often presented as mutually exclusive, they are very similar in their predictions. Specifically, both models posit an actin bundle that appears structurally similar to a stress fiber will acquire a row of sarcomeres over time to become a 'myofibril' (*Dlugosz et al., 1984*; *Rhee et al., 1994*; *Sanger et al., 2005*) (*Figure 1A*). There is a vast amount of localization data in fixed cardiomyocytes to support these models. However, there is very little dynamic data in live cells that suggests stress fibers give rise to sarcomeres. The strongest dynamic support comes from imaging fluorescently tagged α-actinin 2 in myocytes. Time montages from chick skeletal myotubes showed small puncta of α-actinin 2 adding to pre-existing Z lines (*McKenna et al., 1986*). Subsequently, a time montage was used to show a similar phenomenon occurring in chick cardiomyocytes (*Dabiri et al., 1997*).

Some in vivo data support the Template/Pre-Myofibril Model, while others do not. In strong support of the Template/Pre-Myofibril Model, static images of chick heart tissue have essentially revealed every structure described in primary cultured chick cardiomyocytes (*Du et al., 2008*). The presence of NMIIB-containing stress fibers in the cardiomyocytes was particularly clear (*Du et al., 2008*). NMIIB germline knockout (KO) mice were also reported to have fewer and disorganized sarcomeres via EM (*Tullio et al., 1997*). On the other hand, several studies have called into question the role of stress fibers in sarcomere assembly. First, several studies examining cardiomyocytes within mouse or chick heart tissue did not find stress fibers containing NMIIB (*Ehler et al., 1999*; *Kan-O et al., 2012*; *Ma et al., 2009*). In addition, a conditional KO mouse that removes NMIIB genetically at P9 apparently still had striated sarcomere structures (*Ma et al., 2009*). Finally, a conditional heart KO of the other major paralog of NMII, NMIIA, was also reported to have no apparent defects in heart formation (*Conti et al., 2004*; *Conti et al., 2015*). Taken together, the lack of clear data showing stress fibers in cardiomyocytes and inconsistencies for a role of NMII in sarcomere assembly calls into question whether the Template/Pre-Myofibril Model is a viable construct for understanding sarcomere assembly (*Sanger et al., 2005*; *Sparrow and Schöck, 2009*).

There is further data to suggest that a mechanism other than that described in the Template/Pre-Myofibril model could be driving sarcomere assembly. This alternative model—called the 'Stitching Model'—is based on the idea that parts of a sarcomere are assembled independently and then brought together (i.e., stitched) (*Holtzer et al., 1997*; *Lu et al., 1992*; *Sanger et al., 2005*). In support of the Stitching Model, studies in *Drosophila* have shown the presence of small myosin filaments following knockdown (KD) of separate Z-line components (*Rui et al., 2010*). These data suggest that myosin filaments can assemble independently of Z-lines. Indeed, there are also electron micrographs that appear to show stacks of myosin II filaments (i.e., A-bands) without detectable actin filaments in skeletal muscle (*Holtzer et al., 1997*; *Lu et al., 1992*; *Sanger et al., 2005*). Examination of electron micrographs also supports the idea that bodies containing Z-line components and actin filaments—called 'I-Z-I' bodies—could also exist in skeletal muscle without apparent myosin II filaments (*Holtzer et al., 1997*; *Lu et al., 1992*; *Sanger et al., 2005*). Based on this data, it was proposed that stitching could occur through sequential assembly by adding new I-Z-I bodies and myosin II filaments (*Holtzer et al., 1997*; *Lu et al., 1992*; *Sanger et al., 2005*).

The Template/Pre-Myofibril Model and Stitching Model have been proposed to be mutually exclusive explanations of how sarcomeres arise. The Template/Pre-Myofibril Model predicts that multiple sarcomeres will appear approximately simultaneously along the length of a stress fiber, while the Stitching Model would predict that sarcomeres will appear adjacently one by one, sequentially (see original models in (*Dlugosz et al., 1984*; *Holtzer et al., 1997*; *Rhee et al., 1994*)). Here, we leverage our discovery that immature human induced pluripotent stem cell-derived cardiomyocytes (hiCMs) completely disassemble and then reassemble their sarcomeres following plating to test these possibilities. Using this assay, we show that sarcomeres are assembled directly from actin stress fiber templates, and we refer to these stress fibers as Muscle Stress Fibers (MSFs). Our data suggest sarcomere assembly is dependent on the formin actin filament nucleator, FHOD3, non-muscle myosin IIA and non-muscle myosin IIB. Surprisingly, our data do not fully support either the Template/Pre-Myofibril Model or Stitching Model, but rather some aspects of each. As such, we now propose a unified model of sarcomere assembly based on the formation of MSFs and their subsequent transition into sarcomere-containing myofibrils.

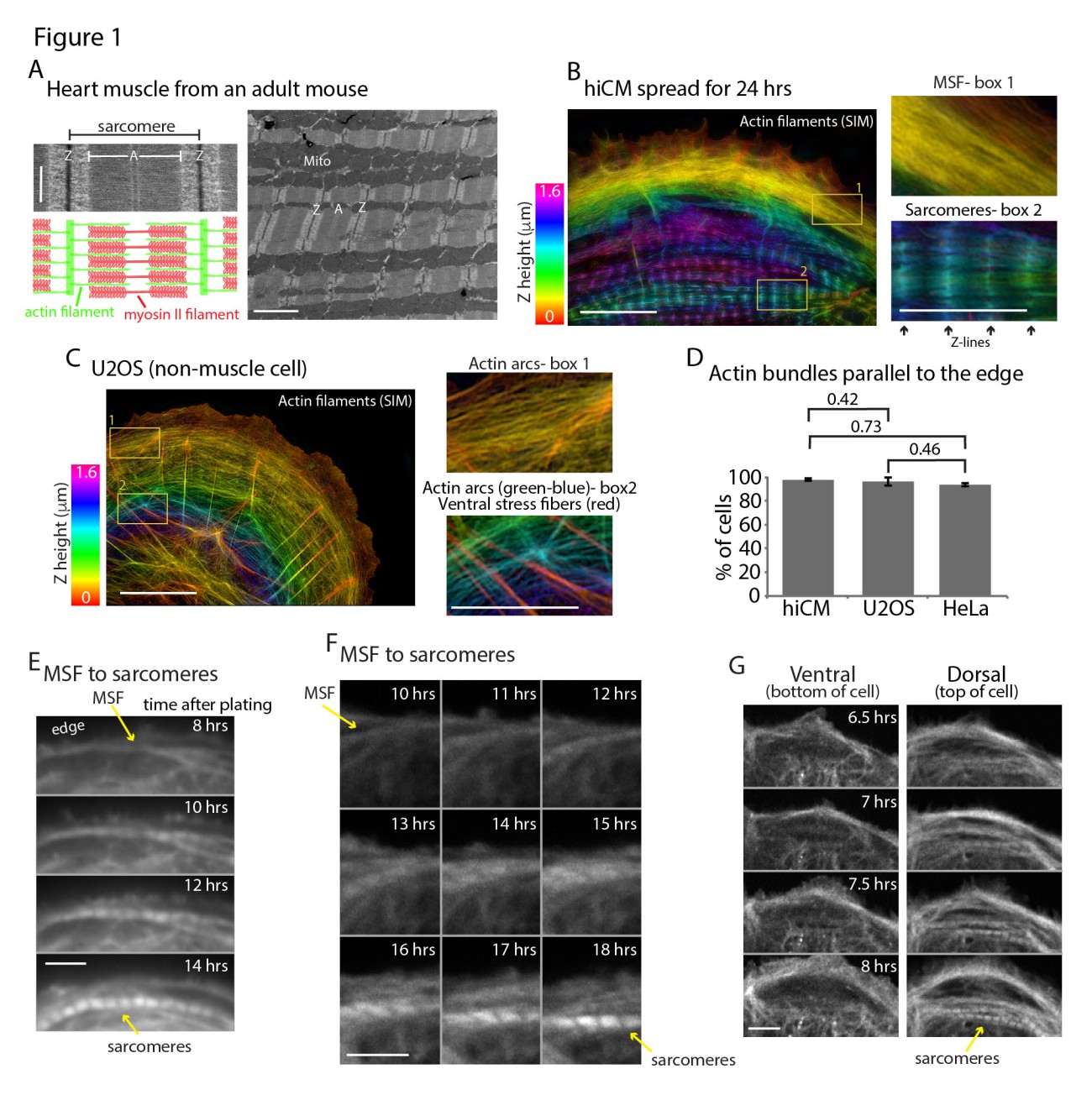

**Figure 1.** Sarcomeres arise directly from Muscle Stress Fiber (MSF) precursors. (**A**) Electron microscopy (EM) schematic of a cardiac sarcomere from adult mouse. Electron dense regions on the borders of a sarcomere are Z-discs (Z), while the core of the sarcomere is composed of thin actin filaments and thick myosin II filaments (**A**). Multiple sarcomeres aligned adjacently form a myofibril (lower mag EM, right). (**B**) hiCM allowed to spread for 24 hr following plating and imaged with SIM. hiCM has been stained for actin and color coding is a representation of height (Z plane) within the cell following 3D imaging (Z-height, left). Notice the clear stress fiber and sarcomere-like actin organization at the front and rear of the cell in box 1 and 2, respectively. (**C**) Spread U2OS cell color coded for Z as in *Figure 1B*, displaying prominent actin arc stress fibers behind leading edge of cell and imaged with SIM. Box 1 shows actin arcs just behind the leading edge of cell, while box 2 shows actin arcs on dorsal surface in cell body (green and blue colored actin), while ventral stress fibers (red colored actin) are on bottom surface of cell. (**D**) Percentage of hiCMs, U2OS, and HeLa cells with actin arc stress fibers. hiCMs; 1372 cells over three experiments. U2OS; 37 cells over four experiments. HeLa; 186 cells over four experiments. (**E**) Wide-field time lapse of hiCM transfected with Lifeact-mEmerald to visualize actin. MSF at front of hiCM undergoes retrograde flow and acquires sarcomeres (yellow arrows). (**F**) Laser-scanning confocal microscopy of hiCM expressing Lifeact-mApple showing MSF to sarcomere transition. hiCM lacks sarcomeres at first time point, and MSF at edge of cell undergoes retrograde flow and acquires sarcomeres (yellow arrows). (**G**) 3D laser-scanning confocal microscopy of hiCM expressing Lifeact-mApple forming sarcomeres. Note how ventral surface (left montage) contains no sarcomere

*Figure 1 continued on next page*

*Figure 1 continued*

structures, while sarcomere assembly occurs on the dorsal surface of cell (right montage). Scale Bars; (**A**) 500 nm high mag (left), 2 µm low mag (right); (**B**) 10 µm low mag, 5 µm high mag insets; (**C**) 10 µm low mag, 5 µm high mag insets; (**E**), (**F**), (**G**), 10 µm. P-values denoted in graphs.

DOI: https://doi.org/10.7554/eLife.42144.002

The following figure supplement is available for figure 1:

**Figure supplement 1.** hiCMs do not contain sarcomeres at early time points post plating.

DOI: https://doi.org/10.7554/eLife.42144.003

## Results

### Development of an assay to test sarcomere assembly

To address how cardiac sarcomeres are assembled, we used hiCMs as a model system (see Materials and methods) (*Takahashi et al., 2007*). We first noted the actin filaments in hiCMs, which had spread for 24 hr, had two distinct organizations, muscle stress fibers (MSFs) and sarcomere-containing myofibrils (*Figure 1B*). Spread hiCMs displayed MSFs at the leading edge and organized sarcomere structures in the cell body (*Figure 1B*). Strikingly, super-resolution imaging revealed the MSFs in hiCMs resembled a classic actin stress fiber found in non-muscle cells, referred to as actin arcs (*Figure 1C and D*) (*Heath, 1983*; *Hotulainen and Lappalainen, 2006*). Actin arcs are stress fibers on the dorsal (top) surface of the cell that are parallel to the leading edge and stain continuously with fluorescent phalloidin (*Figure 1C*). Similarly, both MSFs and sarcomeres in hiCMs are on the dorsal surface (*Figure 1B*). We next sought to test the concept that a MSF obtained sarcomeres as predicted by the Templating/Pre-Myofibril Model.

To test whether MSFs give rise to sarcomeres, we needed to develop a sarcomere assembly assay. We noticed that hiCMs which had been freshly plated (1.5–4 hr post plating) contained no sarcomeres at either the cell edge or cell body, as visualized by SIM (*Figure 1—figure supplement 1A*). Loss of sarcomere structure was confirmed by visualizing multiple sarcomeric proteins, including actin, beta cardiac myosin II (βCMII), α-actinin 2, and TroponinT (*Figure 1—figure supplement 1A*). Though hiCMs did not contain sarcomeres at early time points post plating, hiCMs did display MSFs at the cell edge (*Figure 1—figure supplement 1A*). hiCMs subsequently assembled sarcomere structures over the course of 24 hr (*Figure 1—figure supplement 1B*). We next sought to test if MSFs template sarcomeres. Indeed, time-lapse microscopy of hiCMs expressing the actin probe Lifeact-mEmerald (*Riedl et al., 2008*) revealed that MSFs acquire sarcomeres over time, with the first sarcomeres appearing between 4 and 16 hr (*Figure 1E and F* and *Video 1*). Not surprisingly, this transition occurs on the dorsal surface of the cell (*Figure 1G*). Importantly, to visualize sarcomere assembly in live hiCMs, we began our imaging at early time points post plating (i.e., 1.5–4 hr). At these time points, hiCMs do not contain sarcomeres (*Figure 1—figure supplement 1A*), and this ensured we were visualizing the initial sarcomere assembly event, and not sarcomere rearrangement or reorganization.

To further characterize our sarcomere assembly assay, we used α-actinin 2, which is a classic marker of Z-lines (*Luther, 2009*). Endogenous α-actinin 2 localized to both MSFs and sarcomeres (*Figure 2A*). Small puncta of α-actinin 2 localized to MSFs, while sarcomeres had linear α-

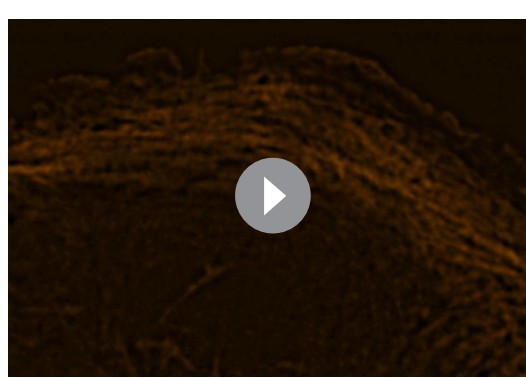

**Video 1.** Actin filaments in a hiCM assembling sarcomeres. hiCM transfected with Lifeact-mApple and imaged with SIM. MSFs undergo retrograde flow and transition to sarcomere containing myofibrils towards cell body. Lookup table: orange hot. 30.5 by 20.9 µm. Video length: 9.5 hr.

DOI: https://doi.org/10.7554/eLife.42144.004

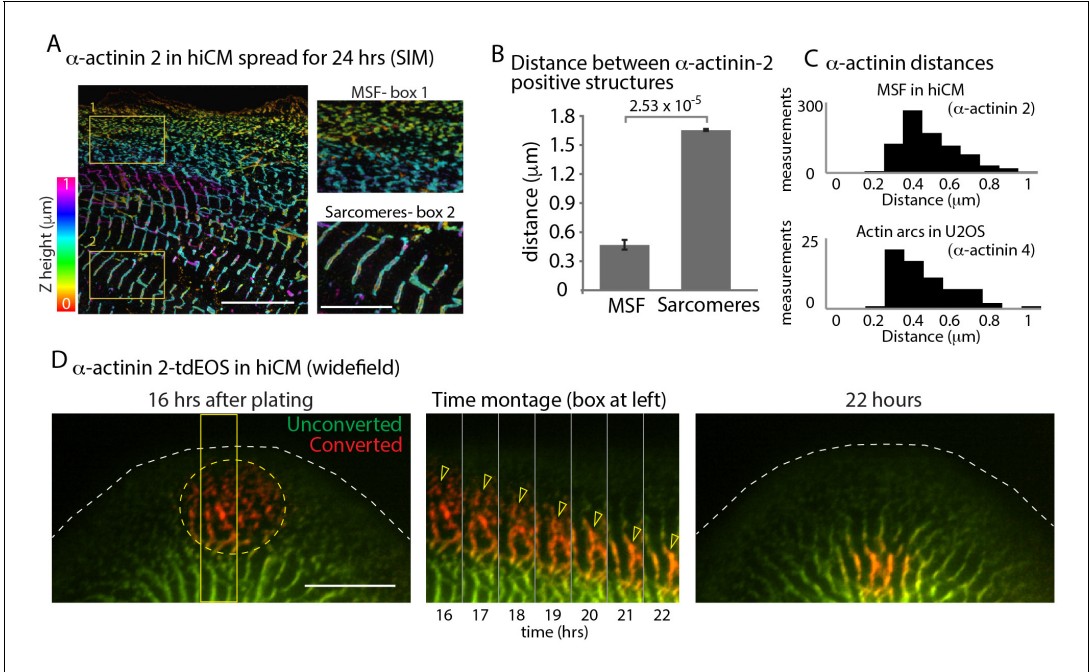

**Figure 2.** α-Actinin 2 spacing and dynamics in hiCMs. (**A**) Color coded representation for Z-height of endogenous α-actinin 2 in MSFs (box 1) and sarcomeres (box 2) of hiCM imaged with SIM. Note difference in structure and spacing of α-actinin 2 in MSFs (box 1) and sarcomeres (box 2) (**B**) Distance between α-actinin 2 structures in MSFs and sarcomeres. MSFs; 14 cells, three experiments, 827 measurements, Sarcomeres; 15 cells, three experiments, 527 measurements. Distance between structures increases as MSFs transition to sarcomeres. (**C**) Histogram depicting distribution of distances between α-actinin 2 structures in MSFs in hiCMs (top) and α-actinin 4 found in actin arcs of non-muscle cells (bottom). Distribution is similar between cell types. (**D**) Wide-field montage of photoconversion of α-actinin 2-tdEOS in hiCM. MSFs at leading edge of the cell were photoconverted (green to red) and imaged over time. Montage (middle) depicts α-actinin 2-tdEOS puncta of MSFs (hollow yellow arrow heads) transition into sarcomere structures (middle, right). Scale Bars; (**A**) 10 μm low mag, 5 μm high mag insets. (**B**), 10 μM. P-values denoted in graphs.

DOI: https://doi.org/10.7554/eLife.42144.005

The following figure supplement is available for figure 2:

**Figure supplement 1.** SIM of NMIIA and α-actinin 2localizations in hiCM

DOI: https://doi.org/10.7554/eLife.42144.006

actinin 2 which labeled Z-lines (*Figure 2A*). As has been shown in other systems (*Dabiri et al., 1997*; *Du et al., 2008*), the spacing between α-actinin 2 puncta increases during the MSF to sarcomere transition, with the spacing of α-actinin 2 ~ 0.5 μm in MSFs and ~1.7 μm in sarcomeres (*Figure 2B*). In hiCMs, α-actinin 2 puncta in MSFs alternates with NMII, as has been shown for other systems (*Figure 2—figure supplement 1*) (*Ehler et al., 1999*; *Hotulainen and Lappalainen, 2006*; *Rhee et al., 1994*; *Sanger et al., 2005*). Interestingly, the spacing of α-actinin 2 puncta associated with MSFs in hiCMs was very similar to the spacing of α-actinin 4 (i.e., a non-muscle paralog of α-actinin) in actin arcs in U2OS cells (*Figure 2C*). If MSFs were serving as a template for sarcomeres, we asked whether the α-actinin 2 molecules in MSFs were also being incorporated into the sarcomere structures. Previous data suggest α-actinin 2 puncta join existing Z-lines (*Dabiri et al., 1997*; *McKenna et al., 1986*). To test this hypothesis, we utilized a photo-convertible probe, tdEOS, which converts from green to red fluorescence to specifically mark the α-actinin 2 puncta of MSFs (*Nienhaus et al., 2006*; *Wiedenmann et al., 2004*). We found that a subset of photo-converted α-actinin 2-tdEOS puncta were indeed incorporated into Z-lines (*Figure 2D*). Collectively, these results strongly suggest MSFs give rise to sarcomeres in hiCMs.

## Actin retrograde flow in hiCMs and non-muscle cells

We next wanted to further investigate the similarities between MSFs and actin arcs. Actin arc stress fibers in non-muscle cells undergo robust 'retrograde flow' away from the edge of the cell as can be seen in U2OS cells, a classic model of mesenchymal migration (*Figure 3A and C*) (*Hotulainen and*

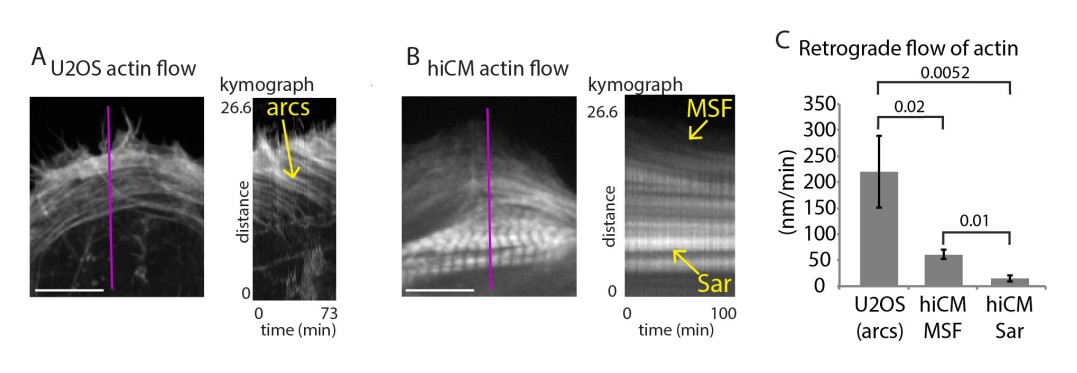

**Figure 3.** Retrograde flow of actin in non-muscle cells and hiCMs. (A) Still of U2OS cell expressing Lifeact-mEmerald (left) imaged with spinning disk confocal. Kymograph (right) taken from purple line of left image. Note robust movement of actin arc stress fibers (yellow arrow). (B) Still of hiCM expressing Lifeact-mApple (left) imaged with spinning disk confocal. Kymograph (right) taken from purple line of left image. Note slower movement of MSF in hiCM compared to actin arcs in U2OS cell, and stationary nature of sarcomeres (Sar). Gamma image correction of 0.5 was used to display relatively bright (i.e., sarcomeres) and dim (i.e, MSF) structures. (C) Quantification of actin stress fiber translocation rates in U2OS cells and hiCMs. U2OS; 3 cells over three experiments. hiCMs; 12 cells over three experiments. Scale Bars; (A), (B), 10 μM. P-values denoted in graph.
DOI: https://doi.org/10.7554/eLife.42144.007

*Lappalainen, 2006*; *Ponti et al., 2004*). Kymography measurements found that actin arcs in U2OS cells moved at ~200 nm/min, in agreement with previously published findings (*Figure 3A and C*) (*Ponti et al., 2004*). We found MSFs also underwent retrograde flow (*Figure 3B and C*). Strikingly, however, kymography revealed MSFs in hiCMs moved significantly slower than actin arcs in U2OS cells (*Figure 3C*). This was the first indication that actin arcs in non-muscle cells are different than MSFs in hiCMs. We next wanted to define the mechanisms governing MSFs and their acquisition of sarcomeres. As the mechanisms of actin arc formation and maintenance have been well studied (*Burnette et al., 2014*; *Hotulainen and Lappalainen, 2006*; *Murugesan et al., 2016*), we were interested in using our assay to test whether the same mechanisms driving actin arc dynamics were governing MSF dynamics.

## Formins, but not the Arp2/3 complex, are required for MSF-based sarcomere formation

The Arp2/3 complex is well known to be required for actin arc formation in non-muscle cells (*Hotulainen and Lappalainen, 2006*). To test the role of the Arp2/3 complex during sarcomere assembly, we allowed hiCMs to spread in the presence of CK666, an inhibitor of the Arp2/3 complex (*Nolen et al., 2009*). Surprisingly, hiCMs allowed to spread in the presence of CK666 formed robust MSFs and sarcomeres comparable to untreated control cells (*Figure 4A*, inset, and *4B*). Cells were quantified as containing sarcomeres if they contained three parallel Z lines in a row each separated from the adjacent Z line by 1 μm – 2.5 μm. An α-actinin 2 localization was defined as a Z line if it was as least 2x the length of the microscope's resolution limit (see Materials and methods). In addition, the spacing between Z-lines between control and CK666 hiCMs was unchanged (*Figure 4C*). We also found the retrograde flow of MSFs was unchanged between control and CK666-treated hiCMs (*Figure 4D and E*). To confirm inhibition of the Arp2/3 complex by CK666, we examined the endogenous localization of the Arp2/3 complex with and without CK666 treatment. The strong localization of the Arp2/3 complex at the edge of control hiCMs was absent in CK666 treated hiCMs (*Figure 4F and G*). To further confirm this observation, we analyzed the loss of p16b, a subunit of the Arp2/3 complex, from the leading edge of hiCMs via CK666 in live hiCMs. Indeed, hiCMs showed rapid loss of p16b-mEGFP from the leading edge following administration of CK666 (*Figure 4H and I*). The delocalization of the Arp2/3 complex from the leading edge is consistent with inactivation by CK666, as shown previously in non-muscle cells (*Henson et al., 2015*). Taken together, our data suggest that the Arp2/3 complex does not need to be localized at the leading edge for sarcomeres to be assembled.

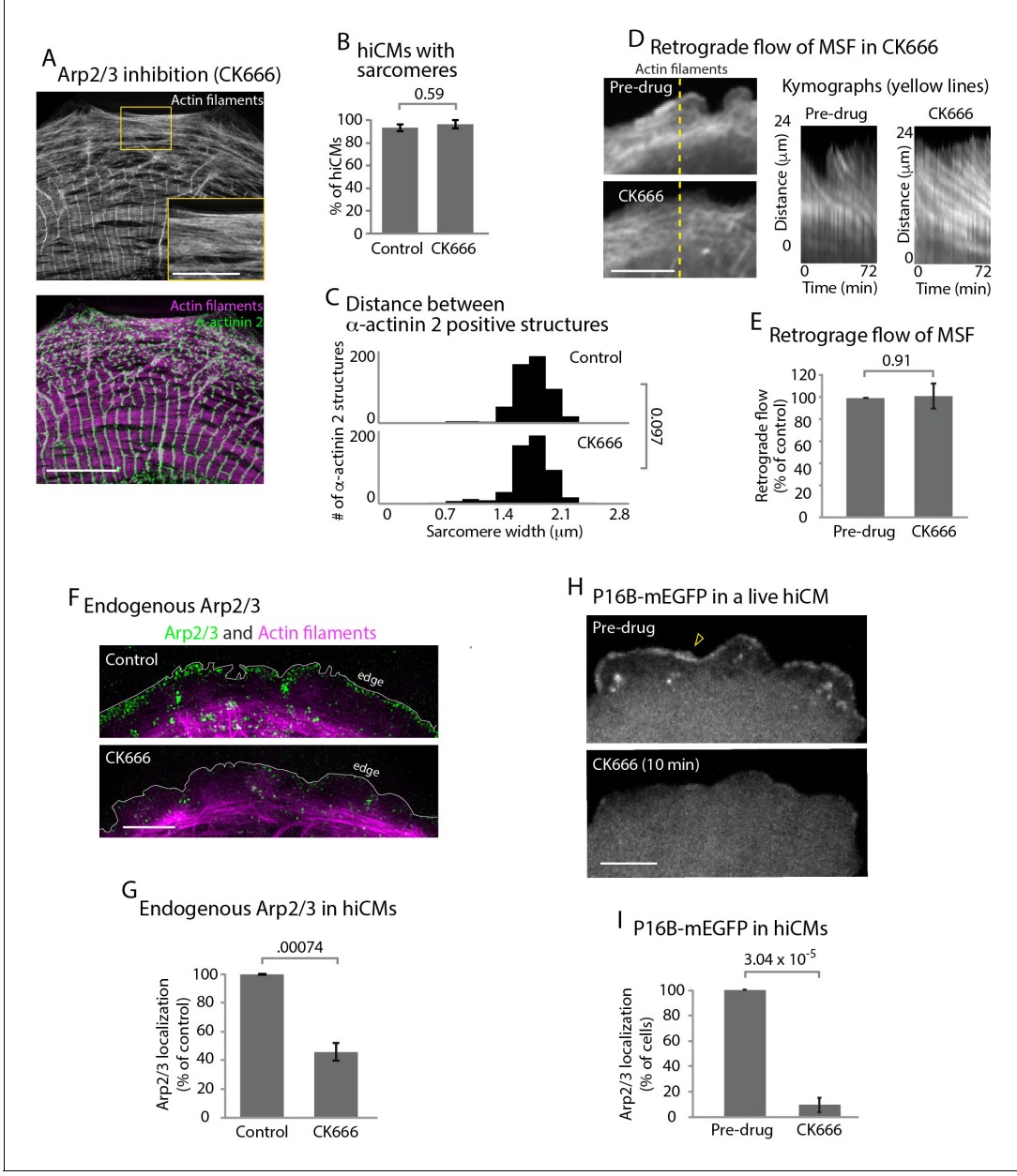

**Figure 4.** The Arp2/3 complex is not required for sarcomere assembly. (**A**) hiCM allowed to spread for 24 hr in the presence of 25 μM CK666, labeled with actin and α-actinin 2 (i.e., Z-lines) and imaged with SIM. Box indicates presence of MSFs. (**B**) Quantification of percentage of cells with sarcomeres at 24 hr post plating in control and 25 μM CK666. Control: 76 cells, 10 experiments; 25 μM CK666: 41 cells, three experiments. (**C**) Histogram of distribution of distances between α-actinin 2 Z-lines. Note tight distribution of Z-lines in both conditions. Control: 14 cells, three experiments, 317 measurements. 25 μM CK666: 16 cells, three experiments, 530 measurements. (**D**) Stills of hiCM expressing Lifeact-mApple pre (top) and post (bottom) addition of 25 μM CK666 and imaged with spinning disk confocal. Kymographs (right) taken from dotted yellow line (left). (**E**) Rates of retrograde flow of hiCMs depicted as percent change in CK666 from pre-drug condition. 8 cells over three experiments. (**F**) Localization of the Arp2/3 complex in control (top) and 25 μM CK666 treated (bottom) hiCMs imaged with SIM. Note loss of Arp2/3 at the edge of CK666 treated hiCM. Cells spread for 24 hr in presence of 25 μM CK666 as in ***Figure 4A*** (**G**) Quantification of loss of the Arp2/3 complex from the leading edge of hiCMs. Control; 36 cells over three experiments. 25 uM CK666; 29 cells over three experiments. (**H**) Live hiCM expressing P16B-mEGFP (a component of the Arp2/3 complex) and imaged with spinning disk confocal. Localization of P16B-mEGFP at leading edge in pre-drug control (top) is acutely lost after addition of 25 μM CK666 (bottom). (**I**) Quantification of hiCMs displaying localization of the Arp2/3 complex (P16B-mEGFP) pre- and post-25μM CK666 in live hiCMs (as in ***Figure 4H***). 27 cells over three experiments. Scale bars; (**A**) 10 μm low mag, 5 μm high mag inset. (**D**), (**F**), (**H**), 10 μm. P-values denoted in graphs.
DOI: https://doi.org/10.7554/eLife.42144.008

In addition to the Arp2/3 complex, formin-mediated actin polymerization has been shown to be crucial for actin arc formation and dynamics in multiple cell types (*Hotulainen and Lappalainen,*

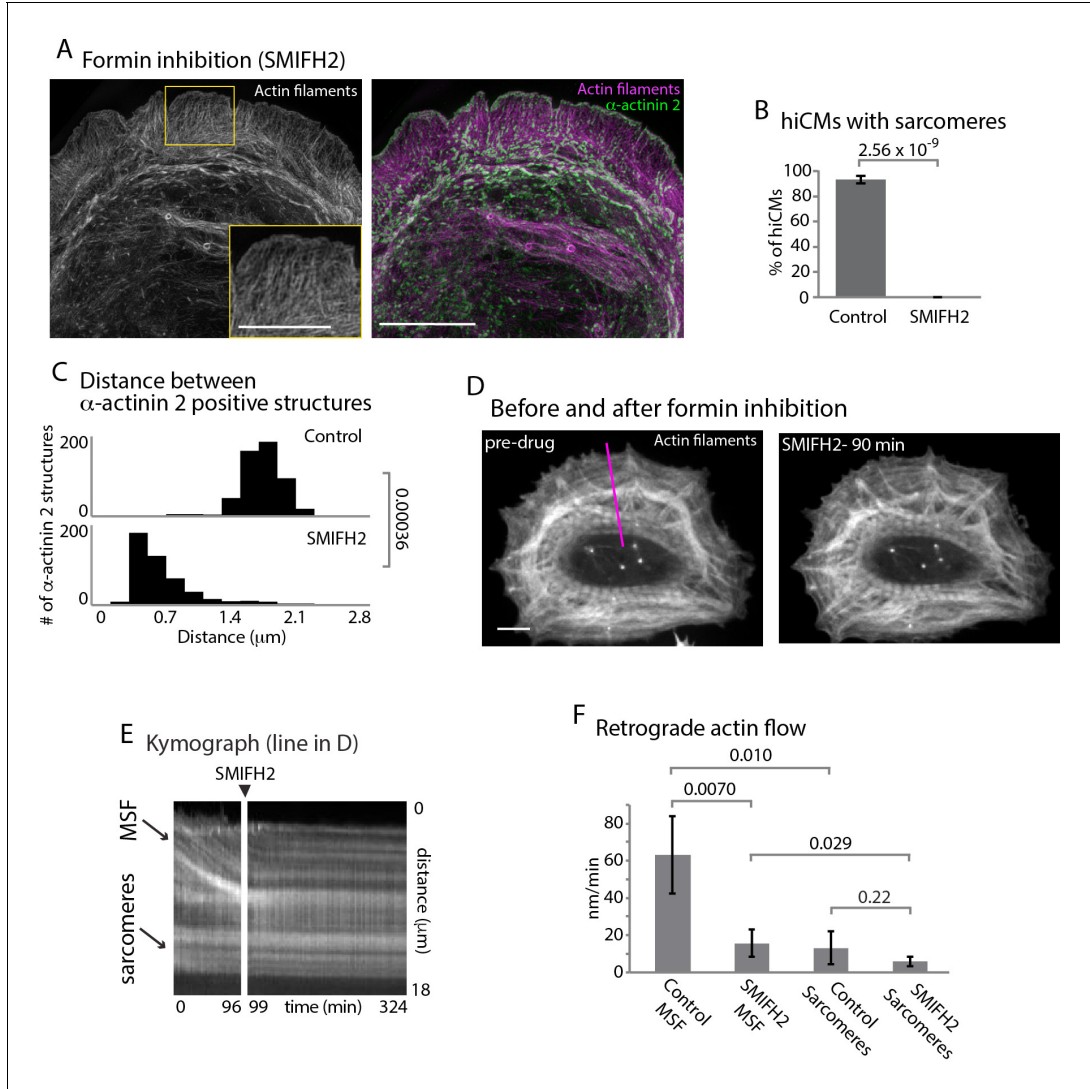

**Figure 5.** Formins are required for sarcomere assembly and MSF dynamics. (**A**) hiCM allowed to spread in the presence of 25 µM SMIFH2 for 24 hr, labeled with actin and α-actinin and imaged with SIM. Box indicates loss of transverse MSFs behind leading edge of hiCM. (**B**) Quantification of percentage of cells with sarcomeres at 24 hr post plating. Control: 76 cells, 10 experiments; 25 µM SMIFH2, 16 cells, three experiments (**C**) Histogram of distribution of distances between α-actinin 2 Z-lines. Control: 14 cells, three experiments, 317 measurements, 25 µM SMIFH2: 11 cells, three experiments, 468 measurements. (**D**) Stills from live hiCM expressing Lifeact-mApple which were spread for 24 hr and assembled sarcomeres. hiCM before (left) and 90 min following addition of 25 µM SMIFH2 (right) (drug administered 24 hr after spreading) and imaged with spinning disk microscopy. Note how sarcomeres and overall actin architecture remains unperturbed at 90 min post 25 µM SMIFH2. (**E**) Kymographs of MSF and sarcomere retrograde flow taken from purple line in (**D**). Note immediate loss of retrograde flow following addition of 25 µM SMIFH2. (**F**) Quantification of actin retrograde flow in hiCMs pre and post addition of 25 µM SMIFH2. Control: 12 cells, three measurements from each cell, three experiments; 25 µM SMIFH2, 12 cells, three measurements from each cell, three experiments. Scale Bars: (**A**) 10 µm low mag, 5 µm high mag inset. (**D**), 10 µm. P-values denoted in graphs.

DOI: https://doi.org/10.7554/eLife.42144.009

The following figure supplements are available for figure 5:

**Figure supplement 1.** hiCM spreading in the presence of 25 µM SMIFH2 .
DOI: https://doi.org/10.7554/eLife.42144.010

**Figure supplement 2.** hiCMs assemble sarcomeres following washout of SMIFH2.
DOI: https://doi.org/10.7554/eLife.42144.011

*2006*; *Murugesan et al., 2016*). As a starting point to test whether formins are required for sarcomere assembly, we allowed hiCMs to spread in the presence of a pan-inhibitor of formin-mediated actin polymerization, small molecule inhibitor of formin homology domain 2, SMIFH2 (*Rizvi et al., 2009*). SMIFH2 has been shown to stop formin-mediated actin polymerization, actin arc formation and retrograde flow in non-muscle cells (*Henson et al., 2015*; *Murugesan et al., 2016*; *Rizvi et al., 2009*). We found hiCMs spreading in the presence of SMIFH2 completely failed to form sarcomeres (*Figure 5A and B*, and *Figure 5—figure supplement 1*). This effect was reversible, as sarcomeres formed after the removal of SMIFH2 (*Figure 5—figure supplement 2*). Distances between α-actinin 2 structures were also significantly decreased in hiCMs treated with SMIFH2, with the distribution of α-actinin 2 more closely resembling MSFs than sarcomeres (*Figures 5C* and *2C*). However, the alignments of the α-actinin 2 puncta were not similar to MSFs in control hiCMs, as they were not periodic (*Figures 5A* and *2A*). This result strongly suggested formins are required for sarcomere assembly.

We next asked if formin inhibition was affecting either the MSFs or sarcomeres directly. To test this, we allowed hiCMs to spread for 24 hr (after they have established sarcomeres) and imaged their actin cytoskeleton via live-cell microscopy before and after administering SMIFH2 (*Figure 5D*). Following addition of SMIFH2, formation of new MSFs was immediately blocked, along with retrograde flow of existing MSFs (*Figure 5D–5F*). However, we did not detect any changes in sarcomere structure over the short time of the experiment, and hiCMs continued to beat in the presence of SMIFH2 (note sarcomere structure in *Figure 5D*). As there are 15 mammalian formin genes, we next asked what specific formin was required for sarcomere assembly.

We performed RNA sequencing analysis of mRNA isolated from hiCMs. Normalized read counts revealed that one formin, FHOD3, was expressed higher than all other formins (*Figure 6A*). Indeed, previous data from isolated rat cardiomyocytes have shown FHOD3 as crucial for sarcomere maintenance (*Iskratsch et al., 2010*; *Kan-O et al., 2012*; *Taniguchi et al., 2009*). Rat cardiomyocytes containing myofibrils subsequently lost their myofibrils following FHOD3 knockdown (*Iskratsch et al., 2010*; *Taniguchi et al., 2009*). However, the role of FHOD3 during de novo sarcomere assembly has not been tested. Therefore, we sought to use our assay to directly test if the formin FHOD3 was required for MSF based sarcomere assembly. We knocked down FHOD3 using siRNA, and hiCMs were unable to assemble sarcomeres following plating (*Figure 6B and C*). Interestingly, KD of the two most highly expressed formins after FHOD3, DAAM1 and DIAPH1, did not stop sarcomere assembly (*Figure 6C* and *Figure 6—figure supplement 1*). However, there are clear defects in the actin organization at the cell edge and in the sarcomeres (*Figure 6—figure supplement 1*). As FHOD3 had the most prominent phenotype, we decided to focus our further analysis on this condition. In line with pan-formin inhibition, the actin organization and spacing between α-actinin 2 in FHOD3 KD hiCMs highly resembled hiCMs spread in the presence of SMIFH2 (*Figure 5A–5C* and *Figure 6B–6D*).

Based on our results, if FHOD3 is involved in sarcomere assembly, it should localize to MSFs. FHOD3-mEGFP localized to both MSFs at the edge of hiCMs, and then becomes increasingly organized away from the leading edge of the cell where sarcomeres are located (*Figure 6E*). This localization is consistent with a role for FHOD3 in mediating the transition from MSFs to sarcomeres. Taken together, our data show that the formin FHOD3 localizes to both MSFs and sarcomeres, and is required for de novo sarcomere assembly. We next wanted to investigate other potential mechanisms regulating sarcomere assembly.

## Non-muscle myosin II is required for cardiac sarcomere actin filaments

In addition to actin nucleators, non-muscle myosin II (NMII) activity has been shown to be required for actin arc formation and organization in non-muscle cell types (*Hotulainen and Lappalainen, 2006*; *Medeiros et al., 2006*). Thus, we asked whether NMII was required for MSF formation and/or the MSF to sarcomere transition. We first localized the two major paralogs of NMII in humans, NMIIA and NMIIB, in spread hiCMs (*Vicente-Manzanares et al., 2009*). Both NMIIA and NMIIB localize to actin arcs in non-muscle cells (*Kolega, 1998*). Consistent with this, both NMIIA and NMIIB localized to MSFs, and were restricted from the middle of the cell where sarcomeres were localized (*Figure 7A and B*). Indeed, time-lapse microscopy revealed NMIIA filaments formed at the edge of hiCMs and underwent retrograde flow as in non-muscle cells (*Figure 7C*). However, NMIIA remained at the edge of hiCMs and was restricted from the cell body where sarcomeres are formed (*Figure 7C* and *Video 2*). NMIIB also remained at the edge of hiCMs and was restricted from the

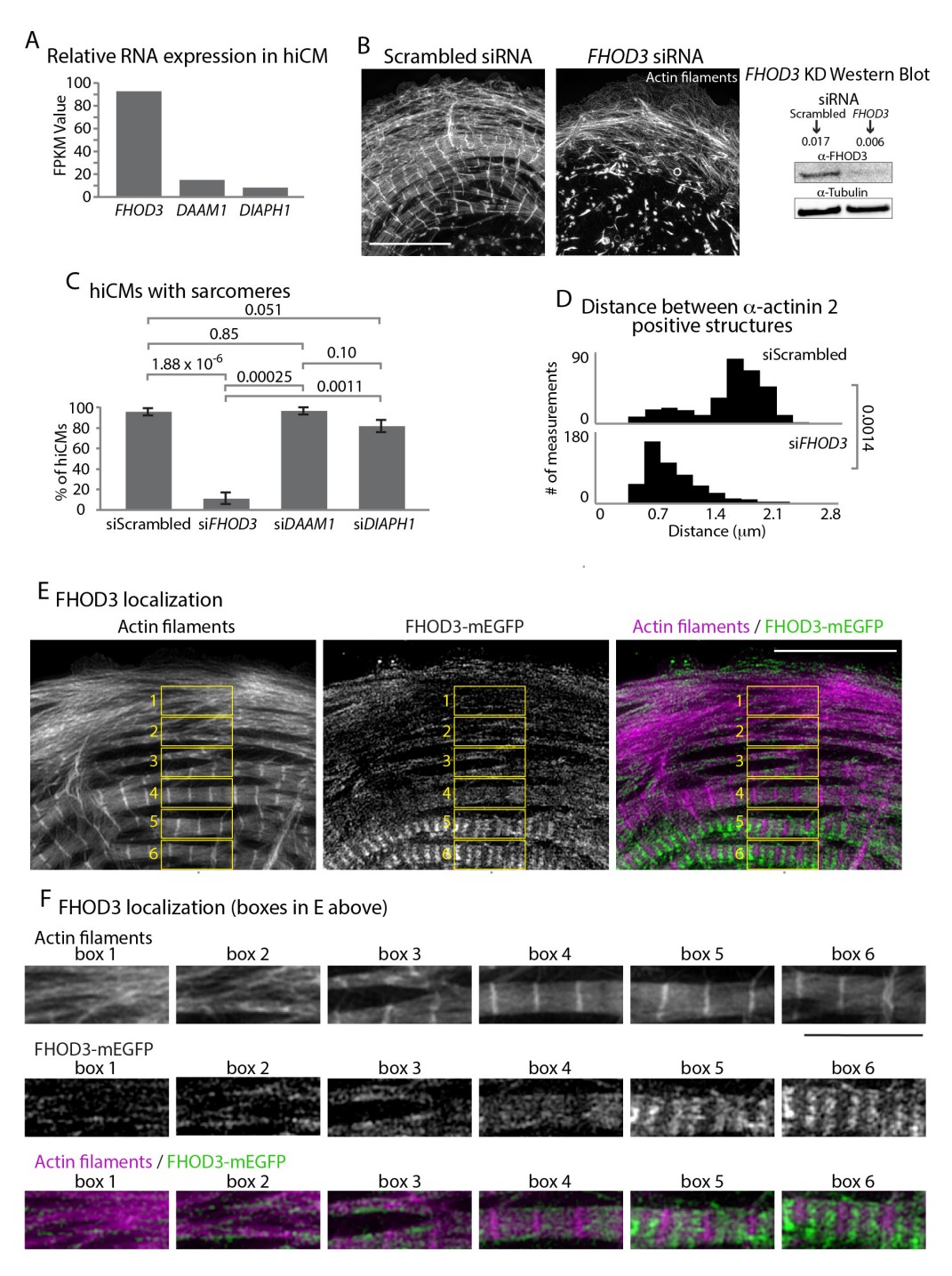

**Figure 6.** Formin FHOD3 is required for sarcomere assembly. (**A**) Normalized Frames Per Kilobase Million (FPKM) of mRNA expression of the top three expressed formins in hiCMs, three experiments, three separate runs. (**B**) Actin of siRNA scramble control and siRNA *FHOD3* hiCMs allowed to spread for 24 hr and imaged with SIM. Note loss of sarcomeres comparable to SMIFH2 treatment in FHOD3 KD hiCMs (*Figure 5A*). Western blot (right) denotes protein loss in FHOD3 KD. (**C**) Quantification of percentage of cells with sarcomeres at 24 hr post plating in scramble control, si*FHOD3*, si*DAAM1*, and si*DIAPH1* hiCMs. Control: 76 cells, 10 experiments; siRNA FHOD3: 33 cells, three experiments; siRNA DAAM1: 29 cells, three experiments; siRNA Dia1: 26 cells, three experiments. (**D**) Histogram of distribution of distances between α-actinin 2 Z-lines. Control: 14 cells, three experiments, 317 measurements; si*FHOD3*: 15 cells, three experiments, 488 measurements. (**E**) hiCM transfected with FHOD3-mEGFP, fixed at 24 hr, stained for Actin and imaged with SIM. Boxes 1-6 depict localization of FHOD3-mEGFP along MSFs (boxes 1 and 2) and in sarcomeres (boxes 4, 5, and 6). (**F**) Boxes from (**E**) showing increased organization of FHOD3-mEGFP from MSFs to sarcomeres. Note increasingly organized structure, and localization between Z-lines of FHOD3-mEGFP. Scale bars; (**B**), (**E**), 10 μm. (**F**), 5 μm. P-values denoted in graphs.

*Figure 6 continued on next page*

*Figure 6 continued*

DOI: https://doi.org/10.7554/eLife.42144.012

The following figure supplement is available for figure 6:

**Figure supplement 1.** hiCMs assemble sarcomeres following knockdown (KD) of DAAM1 and DIAPH1.

DOI: https://doi.org/10.7554/eLife.42144.013

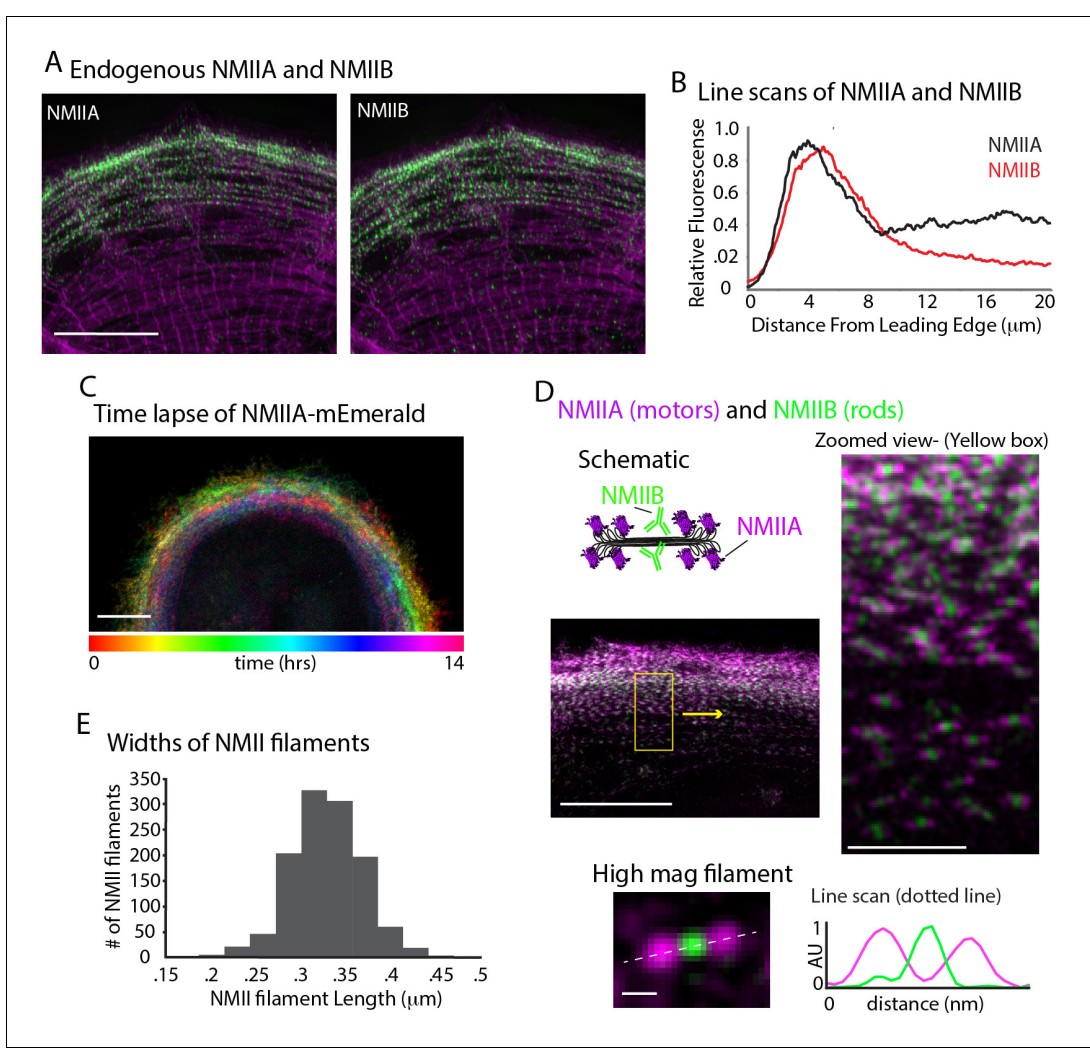

**Figure 7.** NMII Localization and Dynamics in hiCMs. (**A**) Localization of endogenous NMIIA (left) and NMIIB (right) in the same hiCM and imaged with SIM. Both NMIIA and NMIIB localize to MSFs at the leading edge of hiCMs. (**B**) Line scans starting from edge of hiCMs showing localization of NMIIA (black) and NMIIB (red). Note NMIIA is localized slightly,~1 μm in front of NMIIB. NMIIA: 15 cells, two experiments; NMIIB: 32 cells, four experiments. (**C**) Color projection of time-lapse of hiCM expressing NMIIA-mEmerald and imaged with laser-scanning confocal. Note how NMIIA-mEmerald remains at the edge of hiCMs. (**D**) hiCM transfected with NMIIA-mEmerald (N-terminal motors), stained for endogenous NMIIB C-terminal rod domain (cartoon schematic and middle left), and imaged with SIM. High-mag views of NMIIA-NMIIB co-filaments (right) from yellow box (middle left). High mag view of single NMIIA-NMIIB co-filament (bottom) and line scan across white dotted line, from N-terminal motors (purple) and C-terminal rod domains (green). (**E**) Quantification of NMII co-filament length. Histogram displays the distribution of NMII co-filament lengths (motor-domain to motor-domain). Scale Bars; (**A**), (**C**), 10 μm, (**D**) 10 μm low mag (left), 2 μm 'zoomed view' inset (right), 200 nm 'high mag filament' inset (bottom).
DOI: https://doi.org/10.7554/eLife.42144.014

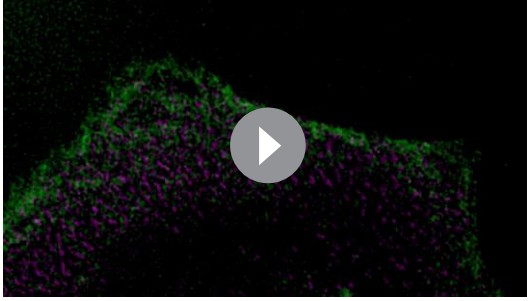

**Video 3.** NMIIB filament dynamics during sarcomere assembly. hiCM transfected with pHalo-NMIIB and α-actinin 2-mEmerald and imaged with wide-field microscopy. Note how NMIIB-Halo filaments form at the edge of hiCMs, are localized to MSFs, but are restricted from sarcomeres during sarcomere assembly. 47 by 27 μm. Video Length: 7 hr.
DOI: https://doi.org/10.7554/eLife.42144.016

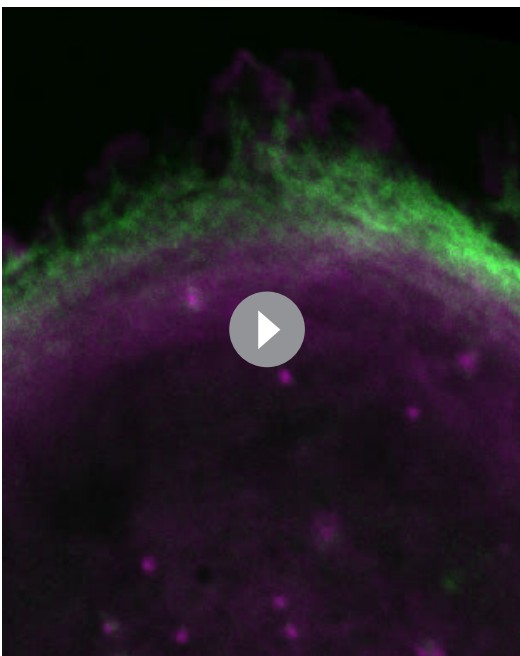

**Video 2.** NMIIA filament dynamics during sarcomere assembly. hiCM transfected with NMIIA-mEmerald and Lifeact-mApple and imaged with 3D laser-scanning confocal. Note how NMIIA-mEmerald filaments form at the edge of hiCMs, are localized to MSFs, but are restricted from sarcomeres during sarcomere assembly. 28 by 35 μm. Video length: 14 hr.
DOI: https://doi.org/10.7554/eLife.42144.015

cell body where sarcomeres are formed (*Video 3*). The vast majority of NMIIA and NMIIB filaments overlapped, except at the very leading edge where NMIIA is localized slightly ahead of NMIIB in hiCMs (*Figure 7A and B*). Super-resolution microscopy revealed that most NMII filaments contained NMIIA and NMIIB (*Figure 7D*). NMIIA and NMIIB co-filaments have previously been reported in non-muscle cells (*Beach et al., 2014*; *Shutova et al., 2014*). Measurements of the lengths of NMII co-filaments in hiCMs showed lengths agreeing with previously published measurements in non-muscle cells (*Figure 7D and E*) (*Beach et al., 2014*; *Shutova et al., 2014*). Taken together, these data suggest NMII organization and dynamics appear similar in MSFs of hiCMs as in actin arcs of non-muscle cells.

Given the presence of both NMIIA and NMIIB in each filament within MSFs, we next asked whether NMIIA and/or NMIIB were required for sarcomere assembly. NMIIA has previously been shown to be required for actin arc assembly in non-muscle cells (*Figure 8—figure supplement 1*) (*Burnette et al., 2014*; *Fenix et al., 2016*). Thus, we hypothesized that NMIIA would likely be the key paralog required for MSF formation and subsequent sarcomere assembly. Surprisingly, KD of NMIIA did not result in a complete inhibition of MSF or sarcomere assembly, although the sarcomeres in NMIIA KD hiCMs were disorganized (*Figure 8A–8E* and *Figure 8—figure supplements 2* and *3*). Notably, NMIIA KD hiCMs displayed a similar distribution of distances between α-actinin 2 structures compared to control hiCMs (*Figure 8D*). This measurement shows that though there are fewer and more disorganized sarcomeres in the NMIIA KD, their widths as measured from Z-line to Z-line are similar to control hiCMs. However, NMIIA KD cells had significantly shorter Z-lines compared to control hiCMs (*Figure 8E*). Taken together, this data suggests NMIIA is involved in sarcomere assembly and organization.

It has previously been shown NMIIB is not required for actin arc formation in non-muscle cells (*Kuragano et al., 2018*; *Shutova et al., 2017*) (*Figure 8—figure supplement 1*). We hypothesized that NMIIB would not be required for MSFs or sarcomere assembly in hiCMs. Surprisingly, NMIIB KD resulted in a complete inability of hiCMs to form sarcomeres after plating (*Figure 8A–8D* and *Figure 8—figure supplements 2* and *4*). In addition, the width between α-actinin 2 structures was significantly smaller than control hiCMs (*Figure 8D*). These results argue NMIIB is also a major player required for sarcomere assembly in hiCMs. To further confirm that myosin II is required for sarcomere assembly, we then pharmacologically inhibited all myosin II paralogs in hiCMs with blebbistatin

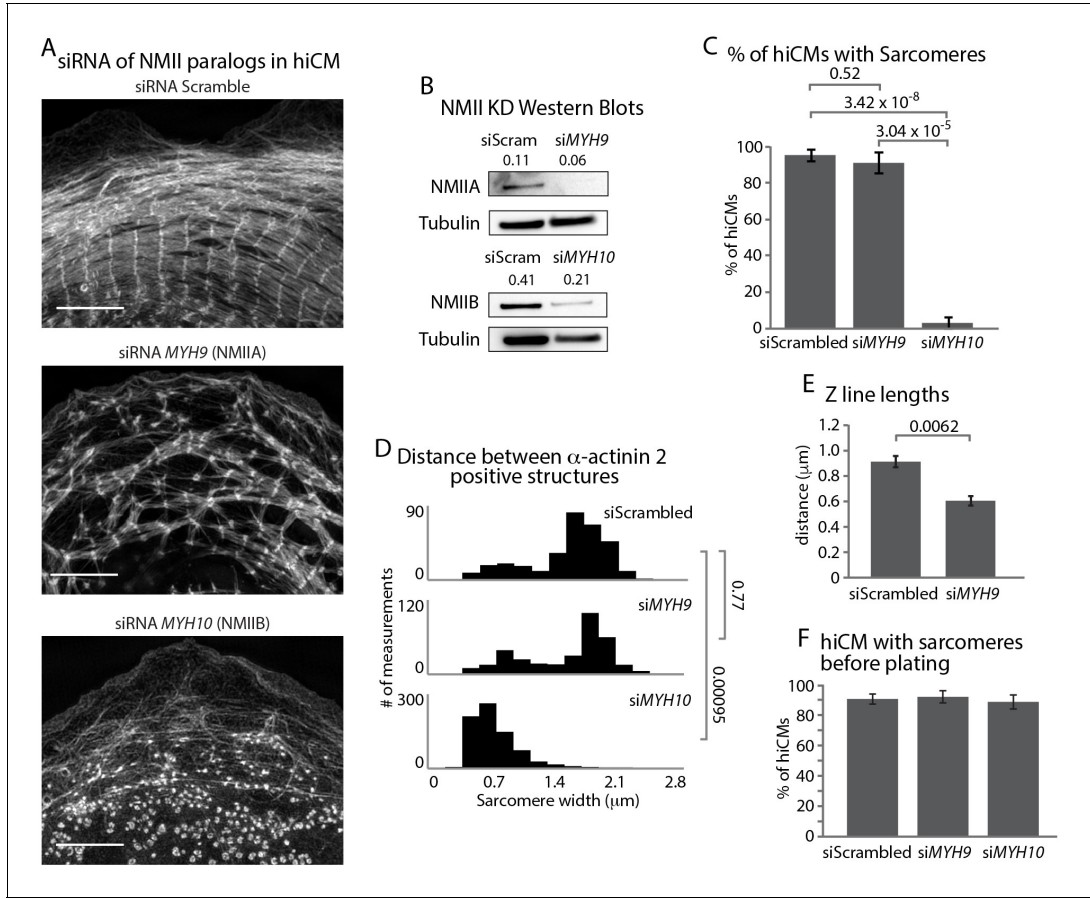

**Figure 8.** NMIIA and NMIIB are Required for Sarcomere Assembly in hiCMs. (**A**) Actin of representative scramble control (top), NMIIA KD (siRNA *MYH9*, middle), and NMIIB KD (siRNA *MYH10,* bottom) hiCMs allowed to spread for 24 hr and imaged with SIM. NMIIA KD hiCMs (middle) display disorganized sarcomeres, while NMIIB KD hiCMs (bottom) display no actin-based sarcomeres. (**B**) Representative western blots of 2 separate experiments showing knockdown of NMIIA (siRNA *MYH9*, top) and NMIIB (siRNA *MYH10*, bottom). (**C**) Percentage of scramble control, NMIIA KD (siRNA *MYH9*), and NMIIB KD (siRNA *MYH10*) hiCMs with actin based sarcomeres at 24 hr spread. Control: 49 cells, six experiments; NMIIA KD: 34 cells, three experiments; NMIIB KD: 59 cells, four experiments. (**D**) Histogram of distribution of α-actinin 2 structures in scramble control, NMIIA KD (si*MYH9*) and NMIIB KD (si*MYH10*) hiCMs. siScrambled: 554 measurements, 14 cells, three experiments; si*MYH9*: 332 measurements, 15 cells, three experiments; si*MYH10*: 772 measurements, 15 cells, three experiments. (**E**) Quantification of Z-line lengths in scramble control and NMIIA KD (si*MYH9*) hiCMs. Control: 22 cells, four experiments; NMIIA: 14 cells, three experiments. (**F**) Quantification of hiCMs with sarcomeres in scramble control, NMIIA KD (si*MYH9*), and NMIIB KD (si*MYH10*) hiCMs before re-plating. siScrambled: 772 cells, four experiments; si*MYH9*: 642 cells, two experiments; si*MYH10*: 385 cells, two experiments. Scale bar; (**A**) 10 μm. P-values denoted in graphs.

DOI: https://doi.org/10.7554/eLife.42144.017

The following figure supplements are available for figure 8:

**Figure supplement 1.** NMIIA but not NMIIB is required for actin arc formation in HeLa cells.

DOI: https://doi.org/10.7554/eLife.42144.018

**Figure supplement 2.** Knockdown of NMIIA and NMIIB in hiCMs.

DOI: https://doi.org/10.7554/eLife.42144.019

**Figure supplement 3.** Live montage of spreading NMIIA KD hiCM.

DOI: https://doi.org/10.7554/eLife.42144.020

**Figure supplement 4.** Live montage of spreading NMIIB KD hiCM.

DOI: https://doi.org/10.7554/eLife.42144.021

**Figure supplement 5.** Live montage of hiCM spreading in 100 μM blebbistatin.

DOI: https://doi.org/10.7554/eLife.42144.022

**Figure supplement 6.** NMIIA KD and NMIIB KD hiCMs contain sarcomeres before plating.

DOI: https://doi.org/10.7554/eLife.42144.023

(*Straight et al., 2003*). hiCMs spreading in the presence of blebbistatin were unable to assemble sarcomere structures (*Figure 8—figure supplement 5*). While these defects in sarcomere assembly were dramatic, we noticed that hiCMs treated with siRNA against NMIIA or NMIIB were still beating before re-plating. This implied that the pre-existing sarcomeres of the hiCMs were still intact after KD before plating. Therefore, we immuno-localized α-actinin 2 to visualize sarcomeres in hiCMs before plating. Surprisingly, we found there were no differences between control, NMIIA, and NMIIB KD cells before plating (*Figure 8F* and *Figure 8—figure supplement 6*) Collectively, this data would suggest NMIIA and NMIIB are required for de novo sarcomere formation, but not homeostasis (i.e., turnover) of pre-existing sarcomeres.

## NMIIB and FHOD3 are required for organized A-band formation

Thus far, our results highlight the importance of formin-mediated actin polymerization and NMII for proper actin filament architecture during sarcomere assembly. We next wanted to address how the thick, β Cardiac Myosin II (βCMII) filaments at the core of the sarcomere (i.e., A-band, *Figure 1A*) assemble. Therefore, we started by localizing endogenous βCMII and NMIIB filaments (*Figure 9A*). βCMII predominately localized behind NMIIB in organized sarcomere structures and showed a peak localization ~15 microns behind the leading edge of the cell, with a slight area of overlap with NMIIB (*Figure 9A and B*). We noted that the area of overlap contained NMIIB-βCMII co-filaments (*Figure 9C and D*, and *Figure 9—figure supplement 1*). In addition, we also found NMIIA-βCMII co-filaments in hiCMs (*Figure 9—figure supplement 2*). To our knowledge, this is the first time a myosin II filament-species has been reported that contains a non-muscle and muscle paralog inside cells. Furthermore, we also found NMIIB-βCMII co-filaments in mouse and human heart tissue, indicating NMIIB-βCMII co-filaments are present in vivo (*Figure 9C* and *Figure 9—figure supplement 1*). The co-filaments containing NMIIB and βCMII were of similar length to NMIIA/B filaments (*Figure 9D*). Indeed, we noticed that near the leading edge of the cell, βCMII filaments are typically smaller and not organized into stacks resembling A-bands (*Figure 9A and C–E*). This suggests βCMII filaments are polymerized at the edge and subsequently grow larger as they move away from the leading edge (*Figure 9E*). The presence of NMII before βCMII filaments grow into larger filaments led us to test the hypothesis that NMII would play a role in βCMII filament formation.

To test if NMIIB was also required for βCMII filament and A-band formation, we depleted hiCMs of NMIIB and localized βCMII 24 hr after plating. Compared to control hiCMs, NMIIB KD-hiCMs displayed a significant decrease in the ability to form A-band-like structures (as defined in the Materials and methods) and a reduced overall number of βCMII filaments (*Figures 9F, G, I and K*). Although βCMII filaments formed, they were highly disorganized compared to control cells, as assessed by Fourier transform (*Pasqualini et al., 2015*) (*Figure 9G*). As the actin puncta left over after NMIIB KD contained α-actinin 2 (*Figure 9H*), we hypothesized that these puncta could also be bound to βCMII filaments. Indeed, we found that βCMII filaments spanned the distance between closely spaced puncta (*Figure 9I*). This data suggested that even when the actin cytoskeleton is severely disrupted, there are still mechanisms leading to the association between it and βCMII filaments. We next sought to expand upon this observation and test whether the observed defects in βCMII filament assembly in the NMIIB KD hiCMs were caused by the disruption of the actin cytoskeleton. To test this, we allowed hiCMs to form sarcomeres for 18 hr, then treated hiCMs with the actin monomer sequestration agent Latrunculin B for 6 hr (*Spector et al., 1983*; *Wakatsuki et al., 2001*). Previous studies in non-muscle cells have shown that latrunculin treatment clears the actin from most of the cell and leaves aggregates of actin filaments scattered throughout the cytoplasm (*Ayscough et al., 1997*; *Gronewold et al., 1999*). We also found aggregates of actin filaments in the periphery of hiCM treated with latrunculin (*Figure 9—figure supplement 3*). βCMII filaments appeared to exclusively localize to these aggregates but were not in organized into A-bands (*Figure 11I J*). Taken together, these results argue that the organization of actin filaments is a major factor in the organization of βCMII filaments. As FHOD3 KD also resulted in severely disorganized actin filament architecture, we localized βCMII in this condition. Indeed, FHOD3 KD hiCM also had disorganized βCMII filaments compared to control hiCMs (*Figure 9—figure supplement 4*).

To further investigate the mechanisms of A-band assembly, we created a full length, human βCMII construct containing a mEGFP tag on the motor domain (i.e., N-terminal) (*Figure 11A*). This construct properly integrated into both single filaments and more mature myofibrils (*Figure 11A B*). In non-muscle U2OS cells, A-band-like stacks of NMIIA filaments are often formed through a process

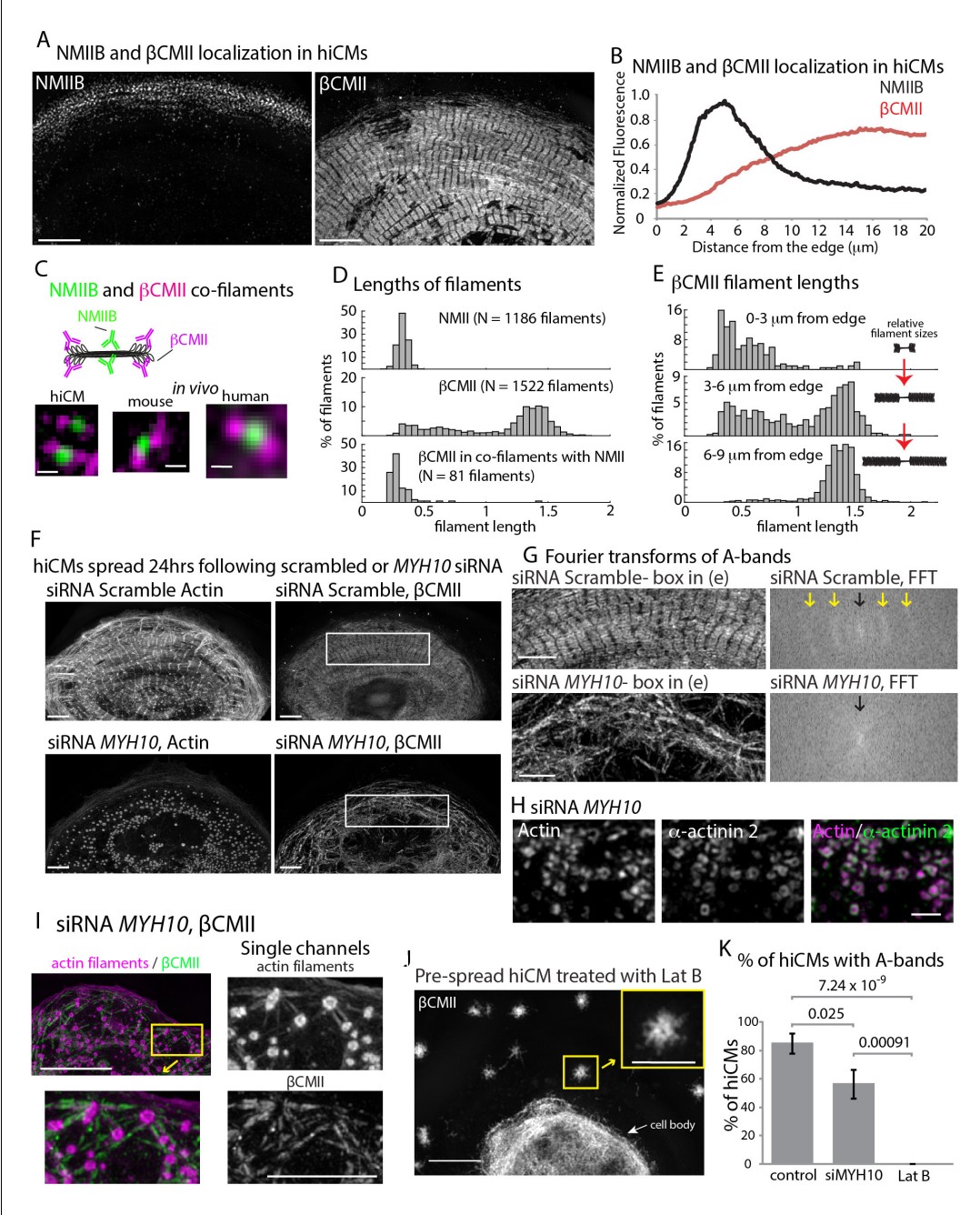

**Figure 9.** β Cardiac Myosin II (βCMII) Filament Assembly in hiCMs. (**A**) Endogenous localization of NMIIB (left) and βCMII in the same hiCM and imaged with SIM. (**B**) Averaged line-scans of NMIIB (**Figure 7B**) and βCMII localization in hiCMs spread for 24 hr. Note the peak fluorescence of βCMII is more towards the cell body than peak fluorescence of NMIIB. 23 cells from four experiments were used for βCMII localization. (**C**) Schematic (top) of NMIIB-β CMII co-filaments. High-mag views of βCMII-NMIIB co-filaments (bottom). Endogenous staining of hiCM (bottom, left) for βCMII (N-terminal motors) and NMIIB (rod domain) and imaged with SIM. Mouse and human tissue (bottom middle, and bottom right, respectively) stained for βCMII (motors) and NMIIB (rod domain) and imaged with SIM and Zeiss 880 with Airyscan, respectively. (**D**) Histograms displaying width of NMII filaments (top), βCMII filaments (middle), and NMIIB-βCMII co-filaments in hiCMs. Measurements made from motor-domain to motor domain as in **Figure 7D and E**. (**E**) Histograms displaying distribution of βCMII filaments widths with respect to their location in hiCMs. Note βCMII filaments tend to grow larger as they move towards the center of the cell. Measurements were not taken from 'mature' sarcomere structures in highly organized A-bands. (**F**) Actin and βCMII of scramble control hiCM (top) and NMIIB KD (si*MYH10*) hiCM (bottom) spread for 24 hr. Note loss of organized A-bands but presence of βCMII filaments in NMIIB KD hiCM. (**G**) Fourier transforms of βCMII signal from white boxes in **Figure 11F** from scramble control and NMIIB KD hiCMs (above and below respectively). Yellow arrows indicate sarcomeric periodicity in scramble control hiCMs, which is lacking in NMIIB KD cells. (**H**) Actin and α-actinin 2 localized in a hiCM after NMIIB KD. (**I**) High mag views of actin and βCMII in NMIIB KD (siRNA *MYH10*) hiCM imaged with SIM. βCMII

*Figure 9 continued on next page*

*Figure 9 continued*

filaments localize to residual actin filaments. (J) βCMII in hiCM spread for total of 24 hr, with the final 6 hr in 5 µM Latrunculin B and imaged with SIM. Notice lack of βCMII A-bands in the periphery of hiCM, and large βCMII filament aggregates (yellow box). (K) Percentage of scramble control, NMIIB KD (siRNA *MYH10*), and 5 µM Latrunculin B hiCMs with βCMII A-bands. Control: 26 cells, three experiments; NMIIB KD: 26 cells, two experiments; Latrunculin B: 11 cells, three experiments. Cell bodies were not analyzed in the latrunculin experiment due to the density of βCMII localization. Scale Bars; (A) 10 µm, (C) 200 nm, (F), (G) 5 µm, (H) 1 µm. (I) 10 µm low mag, 5 µm high mag insets. (J) 10 µm low mag, 5 µm high mag inset. P-values denoted in graphs.

DOI: https://doi.org/10.7554/eLife.42144.024

The following figure supplements are available for figure 9:

**Figure supplement 1.** Myosin II co-filaments in hiCMs and in vivo.
DOI: https://doi.org/10.7554/eLife.42144.025

**Figure supplement 2.** Endogenous localization of NMIIA and βCMII in hiCMs.
DOI: https://doi.org/10.7554/eLife.42144.026

**Figure supplement 3.** Actin, βCMII, and DNA localization in hiCMs treated with Latrunculin A.
DOI: https://doi.org/10.7554/eLife.42144.027

**Figure supplement 4.** βCMII filament assembly is perturbed in NMIIA and FHOD3 KD hiCMs.
DOI: https://doi.org/10.7554/eLife.42144.028

called 'Expansion' (*Fenix et al., 2016*). During Expansion, NMIIA filaments that are close to each other (i.e., in a tight bundle) move away from each other in space but remain part of the same ensemble, where they can be aligned in a stack similar to muscle myosin II in the A-band. In addition to Expansion, NMIIA filaments also, but more rarely, 'Concatenated' (*Fenix et al., 2016*). Concatenation is defined by spatially separated NMIIA filaments moving towards one another to create a stack. To test how βCMII filament stacks form, we repeated our live-cell sarcomere formation assay using our βCMII-mEGFP construct. In contrast to our previous results with NMIIA, we found the major physical mechanism of βCMII filament stack formation to be concatenation, where pre-existing βCMII filaments ran into one another and stitched together to form the A-band (*Figure 11C E* and *Video 4*). A small percentage of hiCMs showed an expansion event of βCMII-mEGFP, however this was significantly less frequent than in non-muscle cells and did not appear to result in a more organized A-band (*Figure 11D E*). Indeed, each of the hiCMs quantified in *Figure 11E* showed only one expansion event.

## Discussion

The goal of this study was to address a decades-old question concerning the origin of sarcomeres assembling in cardiomyocytes. A number of labs have proposed sarcomeres arise from a stress fiber-like precursor, while others propose models where separate sarcomere components sequentially stitch together to form sarcomeres (*Holtzer et al., 1997*; *Lin et al., 1994*; *Lu et al., 1992*; *Rhee et al., 1994*; *Rui et al., 2010*; *Sanger et al., 2005*). Based on previous actin filament staining and NMII localization to these stress fibers, our initial hypothesis was that the mechanisms underlying actin filament polymerization and organization would be related to those governing the non-muscle stress fibers known as actin arcs (see model in *Figure 12*). In support of this hypothesis, both previous reports in non-human cardiomyocytes and our own work with hiCMs show that the actin organization of these precursors appears identical to actin arcs (*Heath, 1983*; *Rhee et al., 1994*; *Sanger et al., 2005*; *Tojkander et al., 2012*). Indeed, both MSFs and actin arcs contain NMII, stain continuously with phalloidin, are on the dorsal surface of the cell, and display retrograde flow in which they move away from the cell's edge (*Figure 12*). However, our study also revealed distinct differences between the regulation and dynamics of MSFs and actin arcs.

It has been well established that the actin filaments of actin arcs in non-muscle cells require both nucleation mediated by the Arp2/3 complex and formins (*Henson et al., 2015*; *Hotulainen and Lappalainen, 2006*; *Murugesan et al., 2016*). Our data would suggest the Arp2/3 is not formally required for MSF or sarcomere assembly in hiCMs. However, the Arp2/3 complex is localized to the edge of hiCMs, and future work will be needed to elucidate its role in cardiac biology. In contrast to the Arp2/3 complex, our data suggest that formins are required for MSFs and sarcomere assembly. Specifically, we found the formin FHOD3 has a major role in MSF and sarcomere assembly

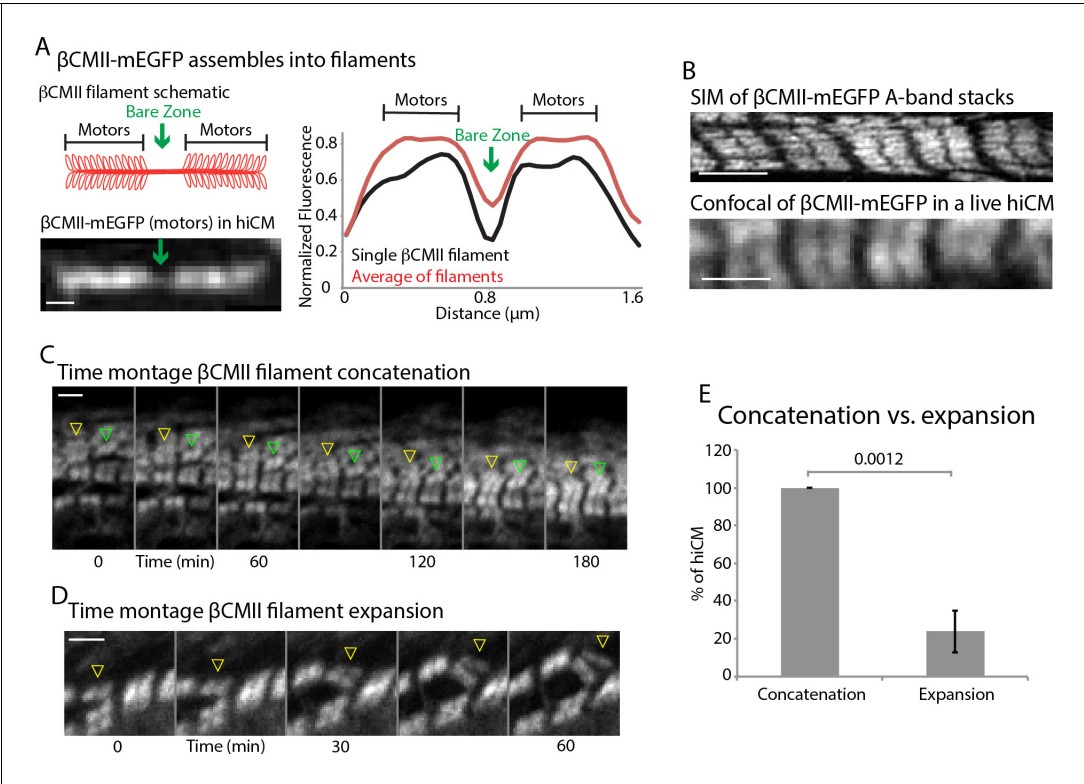

**Figure 11.** βCMII filaments concatenate to form larger A-band structures. (**A**) Cartoon of βCMII filament (left, above). N-terminal tagged human βCMII-mEGFP filament expressed in hiCM (left, below). Gap in signal represents bare-zone lacking motors (green arrows). βCMII single filament and A-band filament (βCMII filaments found within organized A-bands) widths measured by line scans (right). βCMII Filaments: 16 filaments, three experiments. βCMII myofibrils: 28 myofibrils, three experiments. Note more level 'plateau' of signal from motors in A-band βCMII filaments. (**B**) SIM of representative βCMII-mEGFP myofibril in hiCM (top) and laser scanning confocal (bottom). (**C**) Representative montage showing two separate concatenation events. Yellow arrowhead denotes a large stack of βCMII-mEGFP filaments concatenating with a smaller stack of βCMII-mEGFP filaments as they undergo retrograde flow. Green arrowhead denotes smaller βCMII-mEGFP filament concatenating with larger βCMII-mEGFP stack as they undergo retrograde flow. Both events result in larger and more organized βCMII-mEGFP filament stack (i.e. the A-band). (**D**) Example of βCMII-mEGFP filament splitting event. Note how small βCMII-mEGFP stack splits to create two smaller βCMII-mEGFP filaments and does not result in larger or more organized βCMII-mEGFP filament stacks. (**E**) Quantification of % of hiCMs which display concatenation or expansion events of βCMII filaments. Scale Bars; (**A**) 200 nm, (**B**), (**C**) and (**D**) 2 μm. P-values denoted in graph.

DOI: https://doi.org/10.7554/eLife.42144.029

(*Figure 12*). This adds to the already established role of FHOD3 in sarcomere homeostasis (i.e., turn-over) (*Iskratsch et al., 2010*; *Kan-O et al., 2012*; *Taniguchi et al., 2009*). Future work will be required to elucidate the precise mechanism of how and where FHOD3 potentially nucleates actin filaments in sarcomeres. Canonical sarcomeric actin filaments have their barbed ends embedded within a Z-line (see schematic in *Figure 1A*). As such, this is where we would have predicted FHOD3 would localize. However, our SIM data show that FHOD3 does not localize to Z lines (*Figure 6F*, boxes 4, 5 and 6). Instead, FHOD3 appears to localize on either side of each Z-line. This could indicate that FHOD3 is only transiently associated with the barbed ends of canonical sarcomeric actin filaments. Alternatively, there could barbed ends of actin filaments in this region, which could indicate that the organization and/or dynamics of actin filaments within a sarcomere is more complex. Finally, while FHOD3 KD displayed the most severe phenotype, KD of DAAM1 and DIAPH1 in hiCMs resulted in both sarcomeric and non-sarcomeric defects in actin architecture. This agrees with previous literature showing roles for multiple formins in explanted mouse cardiomyocytes in maintenance of sarcomere structure (*Rosado et al., 2014*). The diverse roles of formins during sarcomere assembly and subsequent maintenance should be explored in future work.

We also show that NMIIA and NMIIB are required for proper sarcomere assembly in our hiCM model system. Interestingly, our data suggest that each motor may be playing a different role(s) in

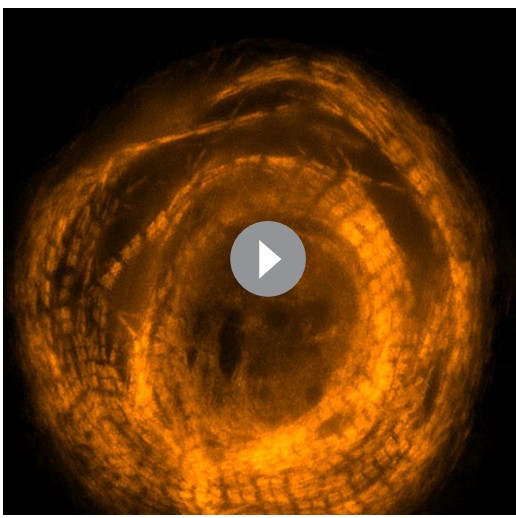

**Video 4.** βCMII filaments in a hiCM assembling sarcomeres. hiCM transfected with βCMII-mEGFP and imaged with SIM. Note filaments concatenating. Lookup table: orange hot. 30.7 by 29.7 μm. Video length: 7.5 hr.
DOI: https://doi.org/10.7554/eLife.42144.030

sarcomere assembly. NMIIB KD resulted in no detectable sarcomere assembly (*Figure 8* and *Figure 8—figure supplement 4*), suggesting that NMIIB could be required for the initial and possibly subsequent steps of assembly. On the other hand, NMIIA KD hiCMs were able to assemble sarcomeres, although these were wispy with significantly shorter Z-lines (*Figure 8* and *Figure 8—figure supplement 3*). This NMIIA KD-phenotype could be a result of several possibilities including a role of NMIIA in sarcomere maturation, alignment, or stability. Obviously, this is not an exhaustive list of potential mechanisms. In addition to showing NMIIA and NMIIB were required for sarcomere assembly, we show NMIIA and NMIIB form myosin II co-filaments with βCMII. βCMII filaments found in co-filaments were relatively small, (~300 nm), and subsequently grew larger as they transitioned to the larger filaments of the A-band (~1.6 microns), and lost NMII. Individual βCMII filaments (or a small bundle below the resolution limit of our imaging modality) concatenate to form the βCMII filament stacks of the A-band. This process requires the presence of actin tracks (*Figure 12*).

Collectively, our data provide new mechanistic and dynamic insight surrounding sarcomere assembly, and highlights key differences between classic, well-studied non-muscle stress fibers, and MSFs in cardiac myocytes (*Figure 12*). In addition, our model unifies certain aspects of previously proposed models of sarcomere assembly (*Figure 12*). These previously proposed models, while presented as mutually exclusive from one another, are actually quite similar in certain respects. Highlighting this, our data support a major feature shared between the Template Model and Pre-Myofibril Model. Specifically, that stress fibers are precursors to sarcomere containing myofibrils. These stress fibers were originally called 'stress fiber-like structures' when the Template Model was proposed (*Dlugosz et al., 1984*; *Sanger et al., 2005*). Later, they were renamed to Pre-Myofibrils in lieu of more thorough characterization (*Rhee et al., 1994*; *Sanger et al., 2005*). It was shown these stress fibers did not contain the same proteins as non-muscle stress fibers (*Rhee et al., 1994*). However, most of the known proteins in stress fibers found in cardiomyocytes have paralogs in non-muscle stress fibers. As such, we find 'stress fiber-like structures' an apt description. Therefore, we suggest simply calling these structures 'stress fibers' is preferable. In this study, we are comparing regulatory mechanisms of the non-muscle stress fiber referred to as actin arcs, to sarcomere precursors in cardiac myocytes. Thus, we decided to call them Muscle Stress Fibers (MSFs) to avoid confusion.

Our data support the concept that MSFs transition into sarcomere-containing myofibrils over time. This is the way the cartoon models of both the Template and Pre-Myofibril Models are presented (see original models (*Dlugosz et al., 1984*; *Rhee et al., 1994*)). An open question remains as to whether there is true templating during this process. The original Template Model posited that 'stress fiber-like structures' [MSFs] would disappear after they were used as a template to build a myofibril (*Dlugosz et al., 1984*). While some components of MSFs (e.g., α-actinin 2) do appear to persist during the MSF to sarcomere transition, others clearly do not (e.g., NMIIA/B). Furthermore, our data suggest that NMIIA/B filaments could be themselves a template for the addition of βCMII as all three paralogs can be found in co-filaments together in the region where NMIIA/B and βCMII overlap. In addition to the Templating/Pre-myofibril Models, our data also support aspects of the Stitching Model. The Stitching Model originally proposed that pre-assembled components of the sarcomere (i.e., A-bands and I-Z-I bodies) would stitch together sequentially to form a myofibril (*Holtzer et al., 1997*). We did not detect pre-formed I-Z-I bodies or A-bands in our assay (*Figure 1—figure supplement 1*) or sequential assembly of sarcomeres to form a myofibril (*Figure 1*). However,

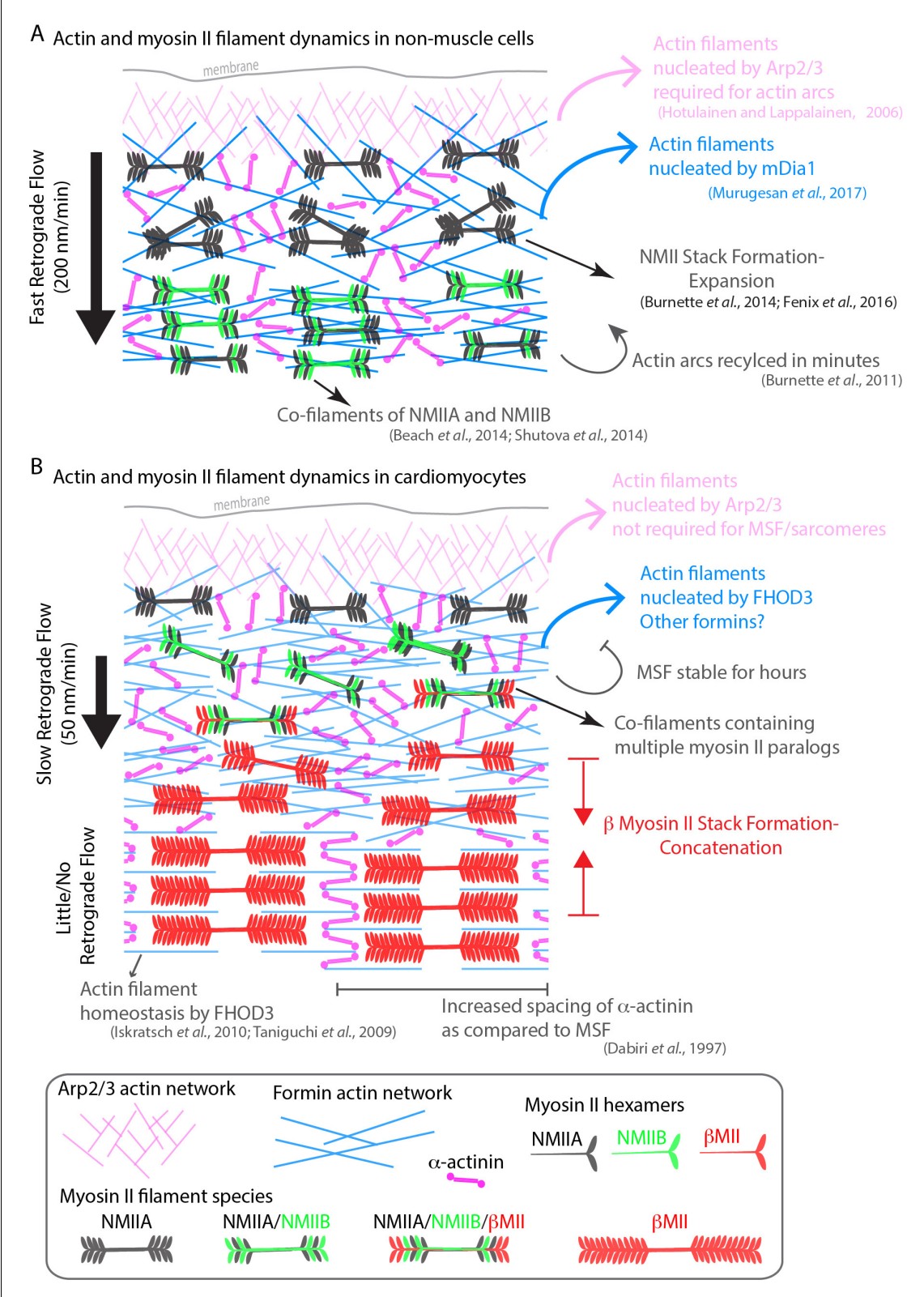

**Figure 12.** Model of actomyosin stress fiber formation in non-muscle cells and human cardiomyocytes. (**A**) Actin and myosin II stress fiber formation in non-muscle cells. Actin stress fibers are formed via the Arp2/3 complex and the formin mDia1. NMIIA is the predominant isoform at the leading edge of non-muscle cells, and stress fiber formation is NMIIA dependent. Non-muscle cells display robust retrograde flow of actin stress fibers and display rapid turnover. Large NMIIA stacks are formed via growth and expansion of smaller NMIIA filaments. Citations leading to this model are presented in
*Figure 12 continued on next page*

*Figure 12 continued*

the cartoon. (**B**) Model of actin and myosin II stress fiber formation in human cardiomyocytes. Sarcomeres are templated by Muscle Stress Fibers (MSFs). MSFs do not require the Arp2/3 complex, and require the formin FHOD3. MSFs display slow retrograde flow compared with non-muscle stress fibers. Both NMIIA and NMIIB are localized to the edge of hiCMs, and display prominent NMII co-filaments. NMIIB-βCMII co-filaments are also present with MSFs. Large βCMII filament stacks form via concatenation and stitching of individual βCMII filaments.

DOI: https://doi.org/10.7554/eLife.42144.031

The following figure supplement is available for figure 12:

**Figure supplement 1.** Sarcomeres form on dorsal surface of hiCMs and subsequently move towards ventral surface.

DOI: https://doi.org/10.7554/eLife.42144.032

certain aspects of βCMII dynamics warrant comparison to the Stitching Model. We found separate βCMII filaments concatenated and 'stitched' together to form larger βCMII filament stacks (i.e., the A-band) (*Figure 11C*).

Our data show the transition of a MSF to a sarcomere-containing myofibril occurs on the dorsal (top) surface of the cell. However, a recent study also imaging iPSC derived cardiac myocytes claims that sarcomeres are formed on the ventral (bottom) surface of the cell near extracellular matrix adhesions (*Chopra et al., 2018*). This group also report that the first sarcomeres forms between 24– and 48 hr after plating, well after we detect the first sarcomeres appearing (*Chopra et al., 2018*). This led us to question what could be leading to these two seemingly opposite results. Importantly, it appears that this group was imaged the ventral (bottom) surface of their myocytes, as the focus of their study was on focal adhesions. Close inspection of their time-lapse movies revealed faint and blurred structures corresponding to sarcomeric patterns that show up in the frame right before the appearance of sarcomeres. This supports the notion that they are imaging sarcomeres that are coming into focus, and not assembling 'de novo'. To test this idea, we imaged hiCMs with 3D confocal microscopy after they had been plated for 24 hr (*Figure 12—figure supplement 1*). While we also see similar patterns of sarcomeres appearing on the ventral surface as (*Chopra et al., 2018*), our data revealed these sarcomeres are moving down from the dorsal surface and not assembling on the ventral surface (*Figure 12—figure supplement 1* and *Video 5*). Of interest, the phenomenon of actin arcs moving down to the ventral surface of non-muscle cells has also been reported previously (*Gao et al., 2012*; *Hotulainen and Lappalainen, 2006*). Finally, (*Chopra et al., 2018*) also claim that neither NMIIA nor NMIIB are required for sarcomere assembly. We also find this strange. While their double NMIIA/NMIIB knockout (KO) cardiomyocyte cell line has α-actinin 2 positive structures, they do not contain continuous labeled Z-lines aligned parallel to each other comparable to the control cell line. The authors did not measure Z-line lengths, spacing, or other criteria needed to define sarcomeres. These discrepancies between our study and theirs, including the role of NMII, need to be harmonized, as it will directly affect our interpretation of future in vivo data concerning sarcomere assembly.

Data from in vivo studies attempting to answer the question of NMII contribution to sarcomere assembly in mice have been difficult to interpret (*Sparrow and Schöck, 2009*). Germline NMIIA KO mice fail to gastrulate and thus sarcomere assembly is impossible to examine (*Conti et al., 2004*). A conditional NMIIA KO mouse was generated using a promoter which is activated after heart formation has begun, and no heart defects were noted (*Conti et al., 2015*). A germline NMIIB KO mouse formed a functional heart, though most pups died before birth due to heart failure (*Tullio et al., 1997*). Only one high magnification image of the KO animal was shown, which demonstrated severe sarcomere disorganization (*Tullio et al., 1997*). Of interest, the NMIIB KO animals also showed highly

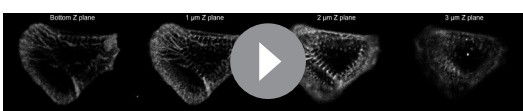

**Video 5.** Sarcomeres traveling from dorsal to ventral surface of hiCM hiCM plated for 24 hr and expressing α-actinin 2-mCherry. Four views from four Z-planes imaged using laser-scanning confocal microscopy. In the first frame, no sarcomeres can be visualized on the bottom of the cell (left most frame), but can clearly be seen in the 2 μm Z plane (middle right frame). As the video continues, the sarcomeres in the 2 μm Z plane travel towards the ventral (bottom Z plane) surface of the hiCM, which can be readily seen as the myofibril comes into focus first in the 1 μm Z plane (middle left), and then in the Bottom Z plane (left). Height: 38.2 μm. Video length: 17.5 hr.

DOI: https://doi.org/10.7554/eLife.42144.033

increased NMIIA protein levels compared to controls (*Tullio et al., 1997*). Thus, the observed capacity of this animal to form sarcomere like structures could be due to genetic compensation by NMIIA. A conditional KO NMIIB mouse has also been made (*Ma et al., 2009*). While the authors showed an impressive NMIIB KD in the cerebellum via a neuronal specific driver, there were high levels of NMIIB protein in the heart at the time of analysis in the heart specific KO (*Ma et al., 2009*). In addition, the heart specific conditional KO is driven off of the alpha myosin heavy chain promoter, which switches on after sarcomere assembly has begun and indeed after the heart fields have begun to beat (*Ma et al., 2009*; *Ng et al., 1991*). Complicating the issue further, NMIIB has been difficult to localize in tissue. We believe this to be a result of paraffin embedding and subsequent paraffin removal and rehydration protocols. For example, we successfully localized NMIIB in formalin fixed human and mouse tissue (*Figure 9* and *Figure 9—figure supplement 1*), but failed to localize NMIIB in paraffin embedded tissue. Future comparisons between in vivo and in vitro data sets will need to be addressed.

Intriguing and unanswered questions also remain to be tested. For example, how are the observed gradients of NMII and βCMII in cardiac myocytes established and maintained? Such questions have been previously difficult to impossible to test due complicated and technically challenging model systems. Here, we present a relatively easy to use model system to test these questions. The hiCMs we use in this study are commercially available. Our Methods outline protocols to transfect with DNA for protein expression and siRNA for protein knockdown, and trypsinization protocols for re-plating. Going forward, we believe studies using the experimental setup we describe here will not only continue to clarify previous models but also reveal new insights into both sarcomere assembly, and cardiac cell biology.

## Materials and methods

**Key resources table**

| Reagent type (species) or resource | Designation | Source or reference | Identifiers | Additional information |
|---|---|---|---|---|
| Cell line (human) | iCell Cardiomyocytes2 | Cellular Dynamics International | | ipsc derived cardiac myocytes |
| Cell line (human) | Cor.4U Cardiomyocytes | Ncardia (formerly Axiogenesis) | | ipsc derived cardiac myocytes |
| Cell line (human) | HeLa | American Type Culture Collection (ATCC) | CCL-2 | |
| Cell line (human) | U2-OS | American Type Culture Collection (ATCC) | HTB-96 | |
| Recombinant DNA reagent | pHalo-NMHC IIA-C18 | this paper | | see Materials and methods for details on creation |
| Recombinant DNA reagent | pHalo-NMHC IIB-C18 | this paper | | see Materias and methods for details on creation |
| Recombinant DNA reagent | α-actinin-2-mEmerald | Addgene | 53988 | |
| Recombinant DNA reagent | α-actinin-2-tdEOS | Addgene | 57577 | |
| Recombinant DNA reagent | NMIIA-(N-terminal)-mEGFP | Addgene | 11347 | |
| Recombinant DNA reagent | Lifeact-mEmerald | Gift from Michael Davidson | | |
| Recombinant DNA reagent | Lifeact-mApple | Gift from Michael Davidson | | |
| Recombinant DNA reagent | NMIIB-(N-terminal)-mEmerald | Addgene | 54192 | |

*Continued on next page*

*Continued*

| Reagent type (species) or resource | Designation | Source or reference | Identifiers | Additional information |
|---|---|---|---|---|
| Recombinant DNA reagent | FHOD-mEGFP (lacking T(D/E)5XE exon) | Gift from Dr. Elizabeth Ehler | *Iskratsch et al. (2010)* | |
| Recombinant DNA reagent | βCMII-mEGFP | Genescript (Piscataway, NJ, USA) this paper | | see Materials and methods for details on creation |
| Recombinant DNA reagent | p16b-eGFP | Gift from Dr. Jenniffer Lippincott-Schwartz and Dr. Pekka Lappalainen | Lai, F. P., et al 2008. *Koestler et al., 2013.* | |
| Antibody | rabbit polyclonal IgG anti-NMIIA | Biolegend | P909801 | Used at 1:1000 |
| Antibody | rabbit polyclonal IgG anti-NMIIB | Cell Signaling | P3404S | Used at 1:200 |
| Antibody | rabbit polyclonal IgG anti-NMIIB | Biolegend | 909901 | Used at 1:200 |
| Antibody | mouse monoclonal IgG anti-βCMII | Iowa Hybridoma Bank | A4.1025 | Used at 1:2 |
| Antibody | rabbit polyclonal IgG anti-α-actinin-2 | Sigma-Aldrich | clone EA-53 | Used at 1:200 |
| Antibody | rabbit polyclonal IgG anti-p34-Arc/ArpC2 (Arp2/3 complex) | Millipore Sigma | 07–227 | Used at 1:100 |
| Antibody | rabbit polyclonal IgG anti-DAAM1 | Bethyl Laboratories | HPA026605 | Used at 1:100 |
| Antibody | rabbit polyclonal IgG anti-DIAPH1 | Sigma-Aldrich | A300-078A | Used at 1:100 |
| Antibody | mouse monoclonal IgGanti-cardiac troponin T | Santa Cruz Biotechnology | CT3, sc-20025 | Used at 1:50 |
| Antibody | goat anti mouse IgG Alexa Fluor 568 | ThermoFisher Scientific | A-11004 | Used at 1:100 |
| Antibody | goat anti rabbit IgG Alexa Fluor 568 | ThermoFisher Scientific | A-11011 | Used at 1:100 |
| Antibody | goat anti mouse IgG Alexa Fluor 488 | ThermoFisher Scientific | AA28175 | Used at 1:100 |
| Antibody | goat anti rabbit IgG Alexa Fluor 647 | ThermoFisher Scientific | A-21244 | Used at 1:100 |
| Antibody | goat anti mouse IgM Alexa Fluor 488 | ThermoFisher Scientific | A-21042 | Used at 1:100 |
| Antibody | mouse monoclonal IgG anti-FHOD3 | Santa Cruz Biotechnology | G-5, sc-374601 | Used at 1:100 |
| Sequence-based reagent | SmartPool siRNA DIAPH1 | GE Dharmacon | E-010347-00-0005 | |
| Sequence-based reagent | SmartPool siRNA DAAM1 | GE Dharmacon | E-012925-00-0005 | |
| Sequence-based reagent | SmartPool siRNA FHOD3 | GE Dharmacon | E-023411-00-0005 | |
| Sequence-based reagent | SmartPool siRNA MYH10 (NMIIB) | GE Dharmacon | E-023017-00-0010 | |
| Sequence-based reagent | SmartPool siRNA MYH9 (NMIIA) | GE Dharmacon | E-007668-00-0005 | |
| Commercial assay or kit | Rneasy Mini Kit | Qiagen | 74104 | |

*Continued on next page*

*Continued*

| Reagent type (species) or resource | Designation | Source or reference | Identifiers | Additional information |
|---|---|---|---|---|
| Commercial assay or kit | QIAprep spin Miniprep Kit (250) | Qiagen | 27106 | |
| Chemical compound, drug | Blebbistatin | Sigma-Aldrich | B0560 | |
| Chemical compound, drug | LatrunculinB | Sigma-Aldrich | L5288 | |
| Chemical compound, drug | LatrucluinA | Sigma-Aldrich | L5163 | |
| Chemical compound, drug | CK666 | Sigma-Aldrich | SML0006 | |
| Chemical compound, drug | SMIFH2 | Sigma-Aldrich | S4826 | |
| Chemical compound, drug | TransIT-TKO | Mirus Bio | MIR 2150 | |
| Chemical compound, drug | FuGENE HD | Promega | E2311 | |
| Chemical compound, drug | Alexa Fluor 488 Phalloidin | ThermoFisher Scientific | A12379 | |
| Chemical compound, drug | Viafect | promega | E4981 | |
| Software, algorithm | (Fiji Is Just) ImageJ | NIH (open source) | | |

## Contact for reagent and resource sharing

Further information and requests for resources and reagents should be directed to and will be fulfilled by the Lead Contact, D.T.B. (dylan.burnette@vanderbilt.edu)

## Experimental model and subject details

### Cell line growth and experimental conditions

Human induced pluripotent stem cell cardiomyocytes (hiCMs) were purchased from either Axiogenesis (Cor.4u, Ncardia Cologne, Germany) or Cellular Dynamics International (iCell caridomyocytes[2], Madison, WI). Cells were cultured as per manufacturer's instructions. hiCMs were cultured in 96 well plates and maintained in proprietary manufacturer provided media until ready for experimental use. Knockdown experiments in hiCMs were started 4 days after the initial thaw. See Plating Assay for more detailed protocol for cell plating. See below for detailed information on protein transfection and siRNA mediated knockdown

U-2 OS and HeLa cells (HTB-96; American Type Culture Collection, Manassas, VA) cells were cultured in 25 $cm^2$ cell culture flasks (25-207; Genessee Scientific Corporation, San Diego, CA) with growth medium comprising DMEM (10–013-CV; Mediatech, Manassas, VA) containing 4.5 g/L L-glutamine, D-glucose, and sodium pyruvate and supplemented with 10% fetal bovine serum (F2442; Sigma-Aldrich, St. Louis, MO). For protein expression experiments in U-2 OS cells, cells were transiently transfected with FuGENE 6 (E2691; Promega, Madison, WI) according to the manufacturer's instructions. Knockdown of NMIIA and NMIIB was performed as previously, with the Accell SMARTpool siRNA to human MYH9, MYH10 or Accell scrambled control purchased from Thermo Fisher Scientific (Waltham, MA) in combination with Lipofectamine 2000 Transfection Reagent (cat# 11668027, Thermo Fisher Scientific, Waltham, MA) for HeLa cells. For both live and fixed cell microscopy, cells were plated and imaged on 35 mm glass bottom dishes with a 10 mm micro-well #1.5 cover glass (Cellvis, Mountain View, CA) coated with 25 µg/mL laminin (114956-81-9, Sigma-Aldrich).

## Plasmids

Plasmids encoding NMIIA-(N-terminal)-mEGFP (11347; Addgene, Cambridge, MA) with mEGFP on the N-terminus of NMIIA heavy chain were used as described previously (*Chua et al., 2009*). Plasmid

encoding Lifeact-mEmerald and Lifeact-mApple were gifts from Michael Davidson. Plasmid encoding NMIIB-(N-terminal)-mEmerald was purchased from Addgene (54192; Addgene, Cambridge, MA). The plasmid encoding α-actinin 2-tdEOS was purchased from Addgene (57577; Addgene, Cambridge, MA). Plasmid encoding human βCMII was synthesized by Genscript (Piscataway, NJ, USA). Briefly, the wild-type human MYH7 (βCMII) sequence from the National Center for Biotechnology Information (NCBI, Bethesda, MD, USA) was cloned into a pUC57 along with Gateway DNA recombination sequences in order to facilitate rapid fluorescent protein integration and swapping (Gateway Technology, ThermoFisher Scientific, Waltham, MA). mEGFP containing a previously published linker sequence was added to the βCMII plasmid using Gateway Vector Conversion System with One Shot *ccdB* Survival Cells (ThermoFisher Scientific, Waltham, MA) for the βCMII-(N-terminal)-mEGFP construct used in this study. FHOD3-mEGFP plasmid lacking the T(D/E)5XE exon was a gift from Elizabeth Ehler. This construct has previously been shown to localize to sarcomeres in neonatal rat cardiomyocytes (*Iskratsch et al., 2010*) (see top left panel of Figure 2A in reference). *Assembling pHalo-NMHC IIB-C18:* pEmerald-NMHC IIB-C18 (Addgene, #54192) was the kind gift of Michael Davidson. Halo tag cDNA was PCR amplified from pHalo-N1 and subcloned into the Age I/BspEI mEmerald site of pEmerald-NMHC IIB-C18, replacing mEmerald, to generate the sequence verified pHalo-NMHC IIB-C18 construct. *Assembling pHalo-NMHC IIA-C18 (NMIIA-HaLo):* The NheI/BspEI Halo cDNA fragment from the pHalo-NMHC IIB-C18 construct was subcloned into the NheI/BspEI mEmerald site of pEmerald-NMHC IIA-C18 (Addgene, #54190) (*Burnette et al., 2014*), replacing mEmerald, to generate the sequence verified pHalo-NMHC IIA-C18 construct. The pEGFP-p16b plasmid (to visualize the Arp2/3 complex) was a gift from Dr. Pekka Lappalainen and previously published (*Koestler et al., 2013*; *Lai et al., 2008*).

## Halo-tag Labeling

hiCM expressing Halo-tag constructs were labeled with JF Dye 585 at 1:100 dilution in pre-warmed culture medium for 1 hr. The plate was then washed by gently exchanging with fresh medium 3 times (*Grimm et al., 2017*; *Grimm et al., 2015*)

## Cell line authentication

The HeLa cell line used in this study was a gift of Dr. DA Weitz (Harvard University). The Burnette lab had this line authenticated by Promega and ATCC using their 'Cell Line Authentication Service' in 2015. The methods and test results received from Promega and ATCC are as follows:

"Methodology: Seventeen short tandem repeat (STR) loci plus the gender determining locus, Amelogenin, were amplified using the commercially available PowerPlex 18D Kit from Promega. The cell line sample was processed using the ABI Prism 3500xl Genetic Analyzer. Data were analyzed using GeneMapper ID-X v1.2 software (Applied Biosystems). Appropriate positive and negative controls were run and confirmed for each sample submitted.'

"Data Interpretation: Cell lines were authenticated using Short Tandem Repeat (STR) analysis as described in 2012 in ANSI Standard (ASN-0002) Authentication of Human Cell Lines: Standardization of STR Profiling by the ATCC Standards Development Organization (SDO)'

"Test Results: The submitted profile is an exact match for the following ATCC human cell line(s) in the ATCC STR database (eight core loci plus Amelogenin):CCL-2 (HeLa)'

The U2 OS cell line used in this study was purchased directly from ATCC by the Burnette lab in 2014 (ATCC, HTB-96). Mycoplasma Monitoring: Both HeLa and U2 OS cell lines were checked for potential mycoplasma infection using either DAPI or Heoscht throughout the course of this study.

## Live-Cell sarcomere assembly visualization assay

To visualize sarcomere formation, we developed a repeatable method, which can be used to visualize any fluorescently tagged protein during sarcomere formation in hiCMs. Prior to performing this assay, cells are maintained in desired culture vessel (our hiCMs were maintained in 96 well plates). In this assay, cells are transiently transfected with desired fluorescently tagged protein, trypsinized, plated on desired imaging dish, and imaged to observe sarcomere formation. A sarcomere assembly assay proceeds as follows. First, hiCMs are transfected with desired fluorescently tagged protein as described below (Viafect, overnight transfection). Transfection mix is washed out with culture media. Cells are then detached from culture vessel using trypsinization method described below. Cells are

then plated on a desired culture vessel (in this study, 10 mm #1.5 glass bottom dishes were used, CellVis, Mountain View, CA) for live cell imaging. Imaging vessels were pre-coated with 25 µg/mL laminin for 1 hr at 37° C and washed with 1x PBS containing no $Mg^{2+}/Ca^{2+}$. Cells were allowed to attach for ~1.5 hr and media was added to fill the glass bottom dish. Cells were then imaged using desired imaging modality at 3–6 hr post plating. The 3–6 hr time window was optimal, as cells had not yet established sarcomeres, but were healthy enough to tolerate fluorescence imaging. This powerful assay can be adapted to visualize the kinetics during sarcomere formation of any fluorescently tagged protein. We have used this assay to visualize actin (Lifeact), FHOD3, βCMII, α-actinin 2, NMIIA, and NMIIB. The latter two proteins being large (i.e.,>250 KD).

## Trypsinization of hiCMs

To trypsinize Cellular Dynamics hiCMs, manufacturers recommendations were used, as follows. All volumes apply were modified from 24 well format for 96 well plates. hiCMs were washed 2x with 100 uL 1x PBS with no $Ca^{2+}/Mg^{2+}$ (PBS*). PBS* was completely removed from hiCMs and 40 uL 0.1% Trypsin-EDTA with no phenol red (Invitrogen) was added to hiCMs and placed at 37° C for 2 min. Following incubation, culture dish was washed 3x with trypsin inside well, rotated 180 degrees, and washed another 3x. Trypsinization was then quenched by adding 160 µL of culture media and total cell mixture was placed into a 1.5 mL Eppendorf tube. Cells were spun at 1000gs for 5 min, and supernatant was aspirated. Cells were then re-suspended in 200 uL of culture media and plated on a 10 mm glass bottom dish pre-coated with 25 µg/mL laminin for 1 hr. Cells were then allowed to attach for at least 1 hr, and 2–3 mLs of culture media with or without drug was added to cells.

To trypsinize Axiogenesis hiCMs, manufacturers recommendations were used, as follows. Cells were washed 2x with 500 µL PBS*. Cells were placed in 37° C incubator for 7 min in PBS*. Following 7 min, PBS* was aspirated and 40 µL 0.5% Trypsin (Invitrogen) was placed on cells for 3 min in 37° C incubator. Following 3 min, 160 µL full Cor.4u media was used to quench trypsinization and re-suspend cells. Cells were then plated on pre-coated glass bottom dish and media added 1.5 hr later to dilute trypsin and fill chamber. Note* this trypsinization protocol has since been modified by Axiogenesis (now Ncardia). See manufacturer for new protocol.

It is important to note that the trypsinization protocol is based off of the hiCMs manufacturers protocols for cell plating, and these hiCMs have been trypsinized prior to functional assays of cardiomyocyte performance and characterization (e.g., drug response, electro-physiology, maturation, etc...) (*Fine et al., 2013*; *Ivashchenko et al., 2013*; *Mioulane et al., 2012*).

## Transient transfection of hiCMs

Cellular Dynamics hiCMs were transfected via modification of manufacturer's recommendations as follows. Volumes used are for transfection in 96 well plates. 2 µL of total 200 ng plasmid (containing fluorescently tagged protein of interest, diluted in Opti-MEM) and 2 µL 1:5 diluted Viafect (Promega, E4981, in Opti-MEM) was added to 6 µL Opti-MEM. Entire mixture of 10 µL was added to single 96 well of hiCMs containing freshly exchanged 100 µL full culture media. Transfection was allowed to go overnight (~15 hr), and washed 2x with full culture media. For transfection of multiple probes, 2 µL of 200 ng plasmid was used for each probe together with 4 µL 1:5 diluted Viafect, into 2 µL Opti-MEM and mixture was applied to cells as above.

Axiogenesis hiCMs were transfected via modification of manufacturer's recommendations as follows. A 3.5:1 Fugene to DNA ratio was used to transfect Axiogenesis hiCMs. 1.2 µL Fugene +0.33 µg DNA per 96 well into 5 µL serum free Cor.4u media was incubated for 15 min at room temperature. 95 µL full Cor.4u media was added to mixture and entire mixture was added to hiCMs (on top of 100 µL already in well). For transfection of 2 separate plasmids, 3.5:1 ratio was used for both plasmids and additional volume was subtracted from 95 µL dilution. Note* this transfection protocol has since been modified by Axiogenesis. See manufacturer for new protocol.

## Protein knockdown

Cellular Dynamics hiCMs were used for knockdown experiments via modification of manufacturers recommendations. Volumes used are for siRNA application in 96 well plates. Dharmacon SmartPool siRNA (GE Dharmacon, Lafayette, CO) targeted to MYH9 (NMIIA), MYH10 (NMIIB), FHOD3, DAAM1, and DIAPH1 were used (E-007668-00-0005, E-023017-00-0010, E-023411-00-0005,

E-012925-00-0005, and E-010347-00-0005, respectively). To achieve KD, a master mixture of 100 µl Opti-MEM (ThermoFisher, Waltham, MA)+4 µl Transkit-TKO (Mirus Bio, Madison WI)+5.5 µl 10 µM siRNA was incubated for 30 min at room temperature. 80 µl of fresh, pre-warmed media was added to hiCMs. Following incubation of siRNA mixture, 8.3 µl of mixture was added to each individual well of 96 well plate. hiCMs were then incubated for 2 days at 37° C. hiCMs were then washed 2x with fresh, pre-warmed media. To achieve KD of NMIIA, NMIIB, FHOD3, DAAM1, and DIAPH1, 3 rounds of siRNA mediated KD described above were necessary. Following 3 rounds of scramble control siRNA treatment, hiCMs still beat and maintained sarcomere structure (see *Figure 8—figure supplement 6*).

## Pharmacological experiments

For all pharmacological experiments, drugs were added to pre-warmed and equilibrated media before adding to hiCMs. For sarcomere assembly assays, 25 µM SMIFH2 (Sigma-Aldrich, S4826), 25 µM CK666 (Sigma-Aldrich, SML0006), and 100 µM Blebbistatin (Sigma-Aldrich, B0560) (separately) were added 1.5 hr after plating to allow hiCMs to attach to the substrate. hiCMs were subsequently fixed (see below) at 24 hr post plating. For live-cell experiments (as in *Figure 5D*), media in imaging container was aspirated using a vacuum system, and media containing drug was added to hiCMs on the microscope stage. This allowed the same hiCM to be imaged pre and post-drug application. For the LatrunculinB experiment (*Figure 9J*), cells were allowed to spread for 18 hr in normal media, and media containing 5 µM LatrunculinB (Sigma-Aldrich, L5288) was added for 6 hr and hiCMs were fixed (for a total spreading time of 24 hr).

## Western blotting

Gel samples were prepared by mixing cell lysates with LDS sample buffer (Life Technologies, #NP0007) and Sample Reducing Buffer (Life Technologies, #NP00009) and boiled at 95°C for 5 min. Samples were resolved on Bolt 4–12% gradient Bis-Tris gels (Life Technologies, #NW04120BOX). Protein bands were blotted onto a nylon membrane (Millipore). Blots were blocked using 5% NFDM (Research Products International Corp, Mt. Prospect, IL, #33368) in TBST. Antibody incubations were also performed in 5% NFDM in TBST. Blots were developed using the Immobilon Chemiluminescence Kit (Millipore, #WBKLS0500).

## Fixation and immunohistochemistry

hiCMs and HeLa cells were fixed with 4% paraformaldehyde (PFA) in PBS at room temperature for 20 min and then extracted for 5 min with 1% Triton X-100% and 4% PFA in PBS as previously described (*Burnette et al., 2014*). Cells were washed three times in 1 × PBS. To visualize endogenous βCMII and transfected proteins in fixed cells, hiCMs were live-cell extracted before fixation as described previously to reduce background and non-cytoskeletal myosin II filaments (i.e., the soluble pool). Briefly, a cytoskeleton-stabilizing live-cell extraction buffer was made fresh containing 2 ml of stock solution (500 mM 1,4-piperazinediethanesulfonic acid, 25 mM ethylene glycol tetraacetic acid, 25 mM MgCl2), 4 ml of 10% polyoxyethylene glycol (PEG; 35,000 molecular weight), 4 ml H$_2$O, and 100 µl of Triton X-100, 10 µM paclitaxel, and 10 µM phalloidin. Cells were treated with this extraction buffer for 1 min, followed by a 1 min wash with wash buffer (extraction buffer without PEG or Triton X-100). Cells were then fixed with 4% PFA for 20 min. After fixation, the following labeling procedures were used: for actin visualization, phalloidin-488 in 1 × PBS (15 µl of stock phalloidin per 200 µl of PBS) was used for 3 hr at room temperature. For immunofluorescence experiments, cells were blocked in 10% bovine serum albumin (BSA) in PBS. Primary antibodies were diluted in 10% BSA. NMIIA antibody (BioLegend, P909801) was used at 1:1000. NMIIB antibody (Cell Signaling, 3404S and BioLegend 909901) were used at 1:200. βCMII antibody (Iowa Hybridoma Bank, A4.1025) was used undiluted from serum. α-actinin 2 (Sigma-Aldrich, clone EA-53) antibody was used at 1:200. FHOD3 antibody (Santa Cruz Biotechnology, G-5, sc-374601) was used at 1:100. Arp2/3 antibody (Anti-p34-Arc/ARPC2, Millipore Sigma, 07–227) was used at 1:100. DAAM1 antibody (Bethyl Laboratories, A300-078A) was used at 1:100. DIAPH1 antibody (Sigma-Aldrich, HPA026605) was used at 1:100. Cardiac Troponin T antibody (Santa Cruz Biotechnology, CT3, sc-20025) was used at 200 µg/ml. Secondary antibodies were diluted in 10% BSA at 1:100 and centrifuged at 13,000 rpm for 10 min before use. Cells were imaged in VECTASHIELD Antifade Mounting Media with DAPI (H-

1200, VECTOR LABORATORIES, Burlingame, CA, USA). To label both NMIIA and NMIIB in the same sample (as in *Figure 7A*), NMIIA primary antibody was directly labeled using a primary antibody labeling kit from Biotium (Mix-n-Stain CF Antibody Labeling Kits, Biotium, Inc. Fremont, CA). Following manufacturers protocol, NMIIA was primary labeled, and stain was visually compared to standard immunofluorescence protocol (see above) to validate localization pattern.

## RNA-seq data analysis

RNA-seq reads were aligned to the human reference genome hg19 using STAR (*Dobin et al., 2013*) and quantified by featureCounts (*Liao et al., 2014*). Read counts were normalized by the Relative Log Expression (RLE) method and FPKM values for each sample were generated by DESeq2 (*Love et al., 2014*).

## Structured illumination microscopy

Two SIM microscopes were used in this study (individual experiments were always imaged using the same microscope). SIM imaging and processing was performed on a GE Healthcare DeltaVision OMX equipped with a 60×/1.42 NA oil objective and sCMOS camera at room temperature. SIM imaging and processing was also performed on a Nikon N-SIM structured illumination platform equipped with an Andor DU-897 EMCCD camera and a SR Apo TIRF (oil) 100 × 1.49 NA WD 0.12 objective at room temperature.

## Spinning disk microscopy

Spinning disk confocal images were taken on a Nikon Spinning Disk equipped with a Yokogawa CSU-X1 spinning disk head, Andor DU-897 EMCCD camera and 100x Apo TIRF (oil) 1.49 NA WD 0.12 mm objective at 37 degrees C and 5%$CO_2$.

## Laser-scanning confocal microscopy

Laser-scanning confocal images were taken on a Nikon A1R laser scanning equipped with a 60x/1.40 Plan Apo Oil objective at 37 degrees C and 5%$CO_2$. Confocal images for *Figure 9C* (in vivo, right) and *Figure 9—figure supplement 1C-D* were taken on a Zeiss 880 with AiryScan equipped with a 63x/1.40 Plan-Apochromat Oil objective at room temperature.

## Wide-field microscopy

Wide-field images were taken on a high-resolution wide-field Nikon Eclipse Ti equipped with a Nikon 100x Plan Apo 1.45 NA oil objective and a Nikon DS-Qi2 CMOS camera. Data presented in *Figure 8—figure supplement 6* was acquired on an Essen Incucyte S3 (Ann Arbor, MI) Live-Cell Analysis system with the 20x objective. Photo-conversion experiments were performed as previously described (*Fenix et al., 2016*).

## Quantification and statistical analysis

### Sarcomere assembly quantification

To quantify percent of hiCMs with sarcomeres (i.e., *Figure 4B*), the actin cytoskeleton (via fluorescently labeled phalloidin) was imaged using structured illumination microscopy. hiCMs were quantified as containing sarcomere structures if they contained at least one myofibril containing at least 3 Z-discs (bright phalloidin staining which overlaps with α-actinin 2 staining as in *Figure 4A*) spaced ~1.5–2 μm apart. By these metrics our quantification of sarcomere formation is not a measure of sarcomere maturity or alignment, but a measure of the hiCMs ability to assemble the building blocks of the sarcomere (i.e., the thin actin filaments) in response to a perturbation. Thus, while NMIIA KD hiCMs clearly form unaligned, disorganized, and fewer sarcomeres and myofibrils than control hiCMs, they still maintain the ability to assemble sarcomere structures, which is reflected in our quantification (*Figure 8A and C*). In the same vein, we realize our quantification is a very liberal quantification of sarcomere assembly. While we don't expect a small array of sarcomeres to represent a functionally capable cardiomyocyte, we are investigating the early steps of sarcomere assembly and the ability of hiCMs to form the basic actin-myosin structure of the sarcomere.

## β cardiac myosin II (βCMII) filament and stack quantification

A similar methodology was used to quantify βCMII A-band filament stacks using endogenous βCMII staining and SIM instead of the actin cytoskeleton. A βCMII filament was quantified as the minimum SIM resolvable βCMII unit which had a bipolar organization (a filament with motor domains on each side), as represented in *Figure 11A*. hiCMs were quantified as containing βCMII A-band filament stacks if they contained even one βCMII filament stack in the cell. A βCMII filament stack was defined as being thicker than the minimum SIM resolvable βCMII filament, indicating multiple SIM resolvable βCMII filaments. Indeed, by this metric, βCMII filament stacks have more resolvable 'motor-domains' than βCMII filaments.

For actin arc and MSF retrograde flow rates (as in *Figure 3*), 3 regions of interest (ROIs) were used per cell. ROIs were drawn using the line tool in FIJI starting from in front of the leading edge (to ensure new MSF formation was captured) to the cell body where sarcomere structures were localized. ROIs were then used to measure MSF translocation rates using the kymograph tool (line width = 3) on hiCMs which had been aligned using StackReg function. Kymographs generated in this manner were then manually measured by counting pixels on the X axis (distance) and the Y axis (time) for a distance/time measurement resulting in translocation rates. This method is similar to previously described methods of actin arc translocation in non-muscle cells.

To measure distance between α-actinin 2 structures, the line tool and measure tool in FIJI (Fiji is Just ImageJ) was used. Lines were drawn, positions recorded (using ROI tool), and distances measured between α-actinin 2 structures. Multiple regions per cell for α-actinin 2 in MSFs, and whole cells for distances between sarcomeres were used.

To measure Z-line sizes in hiCMs, the line tool and measure tool in FIJI was used. Lines were drawn on individual Z-lines and their positions were recorded using the ROI tool. This gave a measurement of Z-line size and position within the cell. For Z-line measurements, such as *Figure 8E*, all Z-lines which could be reliably measured were measured.

## Immunofluorescence localization quantification

To quantify localization of NMIIA, NMIIB, and βCMII, line scans starting from the edge of the cell were taken for every cell and the normalized average localizations were used to average the number of indicated experiments for the final localization patterns depicted in the graph (as in *Figure 7B*).

To quantify Arp2/3 intensity, a similar but altered strategy was taken. four separate boxes for each cell were placed along the edge of the cell (i.e., the lamellipodium) using the actin channel for guidance. These boxes were then used to measure average intensity of the anti-p34 channel (i.e., the Arp2/3 complex). Background subtracted averages for each cell in control and CK666 treated hiCMs were used to quantify percent decrease as depicted in *Figure 4F*.

## Cells used for this study

In order to have comparable results, cells used for this study were standardized based on morphology. Specifically, spread non-muscle cells and hiCMs with a broad leading edge and lamella were chosen as previously shown in studies of both non-muscle and muscle contractile system formation (*Burnette et al., 2014*; *Rhee et al., 1994*). This also facilitated the ability to observe the MSF to sarcomere transition in live hiCMs, and is recommended for future studies investigating sarcomere assembly. All experiments measuring sarcomere assembly were conducted at least 3 times, separately, and cells were imaged using SIM. Though this results in a relatively small number of cells for some of the experiments, we believe super-resolution imaging modalities such as SIM offer invaluable insight into sarcomere assembly (*Gustafsson, 2005*). Indeed, sub-diffraction imaging is required to reliably localize myosin II co-filaments both in vitro and in vivo. Furthermore, as has been seen in *Drosophila*, even high-resolution imaging modalities such as laser-scanning confocal microscopy are not sufficient to detect subtle, yet important, structural changes in response to perturbation (*Fernandes and Schöck, 2014*).

## Statistics

Statistical significance was calculated using 2-tailed, unpaired Students T-tests performed in Excel. All independent experiments represent biological replicates. Error bars in all graphs represent standard error of the mean (SEM). For all graphs depicting % of cells (for example, *Figure 1D*), number

of cells and experiments is indicated in figure legend. Percents displayed represent the average of the averages of the all experiments performed. For example, if controls cells displayed 95%, 90%, and 100% of all cells displaying actin arcs in three separate experiments, the represented percentage in the graph would be 95%. SEM was calculated by dividing the standard deviation by the square root of the number of separate experiments. For actin translocation rates at least three measurements per cell were used to calculate the average translocation rates per cell. Translocation rates were then calculated in same manner as described above. Line-scan graphs represent normalized relative fluorescence.

## Acknowledgements

We thank the M Tyska lab and the Epithelial Biology Center (EBC) at Vanderbilt University for invaluable discussions and reagents. Janice Williams of the Cell Imaging Shared Resources was instrumental in acquiring the EM images in *Figure 1A*. We give a special thanks to Sean Schaffer and Bryan Millis of the Cell Imaging Shared Resources and Nikon Center of Excellence at Vanderbilt University for help with and maintenanc of the SIM and spinning disk microscopes, and Anthony Tharp and Josh Luffman for the CDB CORE Equipment. The authors declare no competing financial interests.

This work was supported by Vanderbilt University School of Medicine Molecular Biophysics Training Grant T32 GM08320 to AMF, an American Heart Association pre-doctoral fellowship #16PRE29100014 to AMF, a NIH F31 pre-doctoral fellowship F31 HL136081 to AMF, a Career Development Award from the National Cancer Institute SPORE in GI Cancer P50 CA095103 to DTB, an American Heart Association Scientist Development Grant #17SDG33460353 to DTB a MIRA Grant R35 GM125028 from NIGMS to DTB, RO1 HL037675 from NHLBI to DMB, a K08 HL116803 to JRB, and an University of Wisconsin Research Growth Initiative grant to JHG.

## Additional information

### Funding

| Funder | Grant reference number | Author |
| --- | --- | --- |
| National Heart, Lung, and Blood Institute | F31 HL136081 | Aidan M Fenix |
| American Heart Association | 16PRE29100014 | Aidan M Fenix |
| National Heart, Lung, and Blood Institute | K08 HL116803 | Jason R Becker |
| National Heart, Lung, and Blood Institute | RO1 HL037675 | David M Bader |
| University of Wisconsin-Milwaukee | ResearchGrowth Initiative | Jennifer H Gutzman |
| National Institute of General Medical Sciences | R35 GM125028 | Dylan Tyler Burnette |
| National Cancer Institute | P50 CA095103 | Dylan Tyler Burnette |
| American Heart Association | 17SDG33460353 | Dylan Tyler Burnette |

The funders had no role in study design, data collection and interpretation, or the decision to submit the work for publication.

### Author contributions

Aidan M Fenix, Formal analysis, Funding acquisition, Validation, Investigation, Visualization, Methodology, Writing—original draft, Writing—review and editing; Abigail C Neininger, Formal analysis, Investigation, Visualization, Writing—review and editing; Nilay Taneja, Formal analysis, Investigation, Writing—review and editing; Karren Hyde, Resources, Methodology; Mike R Visetsouk, Investigation, Methodology; Ryan J Garde, Investigation; Baohong Liu, Formal analysis; Benjamin R Nixon, Annabelle E Manalo, Jason R Becker, Scott W Crawley, David M Bader, Resources; Matthew J Tyska, Resources, Writing—review and editing; Qi Liu, Formal analysis, Methodology; Jennifer H Gutzman,

Resources, Funding acquisition, Investigation, Methodology, Writing—review and editing; Dylan T Burnette, Conceptualization, Resources, Formal analysis, Funding acquisition, Investigation, Visualization, Methodology, Writing—original draft, Writing—review and editing

### Author ORCIDs
Benjamin R Nixon (iD) http://orcid.org/0000-0003-1840-0179
Jason R Becker (iD) http://orcid.org/0000-0002-2107-8179
Jennifer H Gutzman (iD) http://orcid.org/0000-0002-7725-6923
Dylan T Burnette (iD) http://orcid.org/0000-0002-2571-7038

### Decision letter and Author response
Decision letter https://doi.org/10.7554/eLife.42144.043
Author response https://doi.org/10.7554/eLife.42144.044

## Additional files

### Supplementary files
• Transparent reporting form
DOI: https://doi.org/10.7554/eLife.42144.034

### Data availability
Sequencing data have been deposited in GEO under accession codes GSE119743. All other data generated or analysed during this study are included in the manuscript and supporting files.

The following dataset was generated:

| Author(s) | Year | Dataset title | Dataset URL | Database and Identifier |
|---|---|---|---|---|
| Fenix AM, Burnette DT | 2018 | Cardiomyocyte mRNA Content | https://www.ncbi.nlm.nih.gov/geo/query/acc.cgi?acc=GSE119743 | NCBI Gene Expression Omnibus, GSE119743 |

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
