## [Decision Letter]

[Editors’ note: a previous version of this study was rejected after peer review, but the authors submitted for reconsideration. The first decision letter after peer review is shown below.]

Thank you for submitting your work entitled "Muscle stress fibers are essential sarcomere precursors and are regulated differently than non-muscle stress fibers" for consideration by *eLife*. Your article has been reviewed by a Senior Editor, a Reviewing Editor, and three reviewers. The reviewers have opted to remain anonymous.

Our decision has been reached after consultation between the reviewers. Based on these discussions and the individual reviews below, we regret to inform you that your work will not be considered further for publication in *eLife*.

In this paper, the authors investigated sarcomere assembly in human iPSC- derived cardiomyocytes and provide novel molecular insights into this process, which are potentially of broad interest. However, all three reviewers felt that there is a very significant number of issues that need to be addressed before this paper can be published. In particular, the referees found that some data were incomplete and some controls missing, that important quantifications were lacking, and that certain data needed to be analyzed differently. Furthermore, a number of points had been raised that would require addition of new data, such as evaluation of the spacings and dynamics of α-actinin puncta, two-color imaging of myosin II and β cardiac myosin II, and photoactivation experiments to link muscle stress fibers to sarcomere assembly more convincingly. We return the paper to you, hoping that you will find the detailed comments of the reviewers useful for improving your study. However, we will be prepared to consider a new submission, which would fully address the comments of all three reviewers. In case you decide to resubmit your paper to *eLife*, it will be sent to the same Reviewing Editor, who will then decide, based on your revisions, whether the paper warrants re-review and whether the same reviewers should be consulted.

Reviewer #1:

In this manuscript Fenix et al., have reported a novel phenomenon in sarcomere assembly in human IPS derived cardiomyocytes (hiCM), which builds on prior knowledge in the field. Using confocal, super-resolution and live cell imaging assays in hiCM, the authors demonstrate that sarcomeres assemble from F-actin stress fibers organized in arcs near the edge of the spreading cells, that they term muscle stress fibers (MSF). These resemble actin arcs in spreading cells from HeLa and U2OS cell lines but have quantitative differences. They showed that retrograde movement of the MSFs results in maturation of sarcomeres (ie formation of striated myofibrils) and that the rate of retrograde flow of MSFs is significantly slower than that of F-actin arcs in the non-muscle cell types. These observations of myofibril assembly in the hiCMs are overall similar to a number of previous studies using fluorescent-tagged α-actinin to image the transitions of stress fiber-like structures (SFLSs)/pre-myofibrils into myofibrils in primary cardiac myocytes, but numbers and rates have now been quantified here (note the terminology is also different here). In a new advance, they go onto show that the formin FHOD3 rather than Arp2/3 plays pivotal roles in MSF directed myofibril assembly/sarcomere formation (by contrast, Arp2/3 is important for assembly of non-muscle cell F-actin arcs). In addition, they demonstrate novel functions for NMIIA versus NMIIA in myofibril assembly, showing that in the MSF to sarcomere transformation process NMIIB is necessary, while NMIIA appears to play a redundant role, using an acute knockdown approach. Finally, the mechanism of β-cardiac myosin II thick filament assembly into A bands was also studied, and live cell imaging was used to evaluate the mechanism of thick filament assembly into sarcomeres in striated myofibrils. Through these experiments Fenix et al., have convincingly shown that MSFs (which likely correspond to previously-termed SFLSs or pre-myofibrils) are precursors for maturing sarcomeres and require a different non-muscle myosin isoform as compared to the arc-like stress fibers found in non-muscle cell lines. However, some data are weak and do not add to the main points (e.g., the zebrafish data), and there are some missing pieces in the data (e.g., lack of direct comparisons to the previous literature using α-actinin2 as a marker for myofibril assembly, questions regarding location of FHOD3, potential myosin isoform compensation), that need to be addressed to place this study in the context of previous work. Many of the figures also need to be improved to strengthen the points and clarify some ambiguous results.

Main points:

1) The authors claim in the Introduction that the hiCM system allows them to study de novo myofibril assembly in vitro for the first time, unlike the previous primary cardiac cell cultures from embryonic chick hearts or neonatal mouse hearts. However, Figure 5 shows that the hiCM cells contain sarcomeres before replating and spreading! Thus, the hiCM system is actually quite similar to the primary cell systems, which disassemble and then reassemble sarcomeres after replating and spreading. Nevertheless, the hiCM cells are a great system, and the authors present some interesting and novel findings regarding the roles of actin arcs and myosin isoforms in myofibril assembly. The Introduction and Results section should be rewritten and shortened to emphasize the new work on formins and myosin isoforms.

2) There is another very important point regarding previous work on myofibril assembly. The authors state that they are unable to find I-Z-I bodies in the hiCMs, but I can clearly see small α-actinin puncta along the F-actin arcs in Figure 2A. The authors may have been misled by the vast and confusing literature; I-Z-I bodies are simply small α-actinin2 puncta linearly arranged along F-actin bundles that stain continuously for F-actin. These do appear to be present in the hiCMs. The authors should reexamine their control (untreated) hiCMs and present some high mag images of the cell edge and the F-actin arcs stained for α-actinin2. How does the spacing of α-actinin2 puncta in the arcs/MSFs of hiCMs compare with the literature on myofibril assembly in other cardiac cell types? The distances between the α-actinin2 puncta should be measured in the various experiments, including the formin and myosin KD experiments. It is important to compare the so-called MSFs in this study with the various so-called "SFLSs" and "pre-myofibril" etc structures studied by other groups. Most of the live imaging in previous work used fluorescent-α-actinin2 to study formation of myofibrils; thus, one would like to know whether this study obtains similar results. For example, closely-spaced α-actinin puncta can be seen transitioning into the wider striations in the myofibrils as they move inwards at the spreading edge of a chick cardiac myocyte in Figure 5 in Dabiri et al., 1997. I think these could be the arcs that are shown here in the hiCMs. There are other types of examples in the literature.

3) As a related point, what is the direct evidence that the F-actin arcs in the hiCMs are similar to the F-actin arcs in non-muscle cells? The first paragraph of the Results section states: "Strikingly, super-resolution imaging revealed the MSFs in hiCMs resembled a classic actin stress fiber found in non-muscle cells, referred to as actin arcs (Figure 1C and 1D)." I expected the figure would show high mag images of the hiCM arc substructure compared to the non-muscle cell arcs, but this figure shows relatively low mag images of the arcs in U2OS and HeLa cells, revealing continuous F-actin along the fibers in all cell types. How does this show that the MSFs are similar to the arcs in the non-muscle cells? We need to see some high-resolution images of the arcs in the hiCMs and the non-muscle cells – for example what are the spacings between the Z bodies (α-actinin puncta) in the arcs in each cell type? I would like to know whether the NMIIA and NMIIB are located in between the α-actinin2 puncta of the MSFs, which would also be similar to the SFs in the non-muscle cell types (but with a different α-actinin isoform).

4) Actin arcs are shown in the live cell imaging experiments to be precursors for "myofibrils containing sarcomeres" (note- this is a more precise terminology- a sarcomere is a small unit of a myofibril- and the arc may have a small actomyosin 'mini-sarcomere' type structure similar to the stress fibers of non-muscle cells). Can other types of actin stress fibers like dorsal and ventral stress fibers also mature into sarcomeres in hiCM? Or, do these hiCM cells only possess actin arcs? This seems unlikely, as I think I can see some radial stress fibers in the cells in Figure 3E and in Figure 1H near the edge.

5) The authors have shown that FHOD3 knockdown in hiCM cells results in reduced sarcomere formation. However, they have not shown the cellular distribution of FHOD3 by immunofluorescence, unlike Arp2/3. Where does FHOD3 localize in the hiCM in the initial MSF stage, and later when the sarcomeres are formed? Does FHOD3 directly interact with the MSFs and/or the sarcomeres?

6) The authors use a pan-formin inhibitor, SMIFH2, to show that formins are important for formation of the actin arcs and sarcomeres. While FHOD3 is clearly important, their experiments do not rule out roles for other formins. For example, see Rosado et al., 2014. Also, the authors claim that mDia1 is not involved in the MSFs, but this has not been tested here. This could be tested, or at the least, discussed and clarified.

7) NMIIA and NMIIB can form heteropolymers in hiCM, yet NMIIB knockdown alone disrupts sarcomere formation. Is this because there is a compensatory rise in NMIIB expression upon NMIIA knockdown, which restores the total NMII levels in the cell? However, another possibility is that the NMIIA levels are not enhanced upon NMIIB knockdown, resulting in a drop of cellular total NMII levels, bringing about a disruption of sarcomere assembly. The authors should examine the NMIIA levels upon NMIIB knockdown, and vice versa, to address this issue. It would also be helpful to provide a quantitative estimate of NMIIA/NMIIB distribution in the soluble and insoluble fractions of hiCM cells during the MSF and sarcomere stages. This may confirm why one isoform is more important than the other in sarcomere formation.

8) Although the results show that NMIIB is essential for de-novo sarcomere assembly in spreading cells, is it also important for the maintenance of the sarcomeres?

9) I appreciate the authors' efforts to study NMIIB in vivo, but the experiments are not convincing. The zebrafish morpholino experiments are potentially exciting but are underdeveloped and missing important controls. What is the result of MO for MYH9 (NMIIA)? MO of the MYH10 (NMIIB) could also affect heart development via effects on non-cardiac cell types, e.g., vascular endothelium. A rescue using a cardiac-specific MYH10 transgene would address this. Also, MOs often have off target effects, and CRISPR approaches are now used to deal with this issue. Due to these issues, I recommend omitting the zebrafish experiments and following this up in a future more complete study. In addition, Figure S3C of β-cardiac myosin and NMIIB staining in heart sections could simply be showing NMIIB in a small gap between two adjacent cardiac myofibrils. One would like to see a linear array of the BCMII/NMIIB along some continuously stained F-actin fibers to be convinced. This could be omitted also.

10) I agree wholeheartedly with the authors that their work on the NMIIA and IIB indicates new refined models are necessary and that we should not get hung up on the terminology. The authors may want to refine their model shown in Figure 11 based on data regarding α-actinin2 localization and assembly, as well as the other additional experiments and clarifications I have suggested.

Specific Points for improving Figures:

1) Figure 1F. I can't really see the transition of the MSFs to myofibrils with sarcomeres in Figure 1F. There are too many other myofibrils crossing it and near it. Also, is this indicated "MSF" really an actin arc? It is practically in the middle of the cell. The wide-field example in Figure 1E is better; we need a confocal example similar to this one.

2) Figure 1H. The kymograph in Figure 1H is strange- the retrograde flow of the MSF is convincing, but the purple line through the "sarcomere" seems to be parallel to the myofibrils, which are orthogonal to the MSFs. The authors need to find another example where the myofibrils with sarcomeres are parallel to the MSFs and to the cell edge – similar to Figure 1E, or to the cell and kymograph in Figure 3E-F.

3) Figure 2. The images of Arp2/3 staining in the control and CK666 cells in Panel C appear to show the edge of a spreading cell; with the actin fibers well-behind the edge (a lamellipodium, presumably). However, the image of the cell in Panel A is not the edge of a spreading cell since the actin fibers are right up at the edge. Where is the Arp2/3 in cells like those in Panel A?

4) Figure 3. It is hard to compare the F-actin distribution in Panel A (SMIFH2 inhibition) with that in Panel C for the FHOD3 siRNA experiment. The F-actin in Panel A could be shown at higher magnification and brighter to match that of Panel C.

5) Figure 4. The NMIIA appears to be enriched right at the edge of the cell, while the NMIIB is more abundant further away from the edge. However, the line scan shown in Figure 4B does not match the images. The image in Panel C where the NMIIA and IIB are co-localized also shows that the IIA is relatively more abundant near the cell edge. The line scan should be replaced with data that better matches the images.

6) Figure 4E. The NMIIA vs NMIIB knockdown is cool! The MYH10 knockdown cell appears to form abundant F-actin donut structures that resemble podosomes. I can see them in Figure 6F also. Are they podosomes? If so, they would contain vinculin. What might this mean? Competition between different F-actin structures? Effects of NMIIB in cell adhesion?

7) Figure 6. Can a high mag field of the NMIIB/β-cardiac myosin at the edge of the cell be shown to illustrate better the authors' point that the filaments are smaller and not lined up? The individual filaments in Panel C show the co-assembly, but not the overall pattern of the β-cardiac myosin compared to the NMIIB. Also, does IIA co-assemble with β-cardiac myosin?

8) Figure 6F. The movie of the β-cardiac myosin in the control cells (Video 6) is nice and clearly shows the thick filament assembly at the very edge of the cell, where the NMIIA is located. Since a major point of this figure is that NMIIB helps coordinate the β-cardiac myosin assembly, I would also like to see a live cell movie of the β-cardiac myosin in the MYH10 knockdown cells.

9) Figure 6F-G. The result that the MYH10 KD reduces F-actin assembly but allows β-cardiac myosin assembly to form thick filaments is interesting. However, a caveat is that it might be difficult to image very thin F-actin fibrils due to the very bright podosomes. Staining for α-actinin2 would address whether there are residual myofibril-like scaffolding structures; as might be suggested by the end-to-end lining up of the β-cardiac myosin structures.

10) Figure 6H. The β cardiac myosin appears to form star like structures in the Latrunculin treated cells. What might these be? If the cells are stained for F-actin, are there residual F-actin fibrils connecting the stars? Where is the α-actinin2? This reminds me of the polygonal arrays of F-actin and α-actinin2 in spreading cardiac myocytes reported by Holtzer's group. Lin et al., 1989. Does Latrunculin lead to changes in cell spreading? Or size?

11) Figure 7. Can the distinction between myosin stack formation by expansion versus concatenation mechanisms be explained more clearly? I don't get it. In the expansion image sequence in Figure 7D, it appears that a thick filament is splitting into two separate adjacent thick filaments- this could later lead to a branched myofibril (a well-known process in myofibril assembly). I don't see how this image shows that the thick filaments are adding on to one another make a thicker myofibril.

Reviewer #2:

This study by Fenix et al., builds on decades-old models of muscle sarcomere formation that postulated that non-muscle myosin assemblies served as templates for the formation of muscle myosin II-containing sarcomeres. This group was optimally positioned to tackle this issue that included important questions, for example the existence of actomyosin filaments containing non-muscle and muscle myosin II elements, the differential role of the major non-muscle myosin II isoforms, etc. This study elegantly answers several of these questions. Specifically, the authors demonstrate that: (1) formins are required for sarcomere formation; not so much the Arp2/3 complex; (2) the non-muscle myosin II isoforms play differential roles as sarcomeric precursors; (3) that this is likely the case in vivo.

Overall, I think this is a terrific study, one that definitively merits publication in *eLife*. Having stated this, I have some issues with several data sets, which in my opinion require some additions, modifications, and clarifications. However, I am convinced the authors can perform these experiments in a reasonable amount of time.

Specific point:

1) In Figure 1G-H-I, it seems clear that MSF and actin arcs are not the same entities in terms of dynamics, e.g. rates of retrograde flow. However, a major distinctive feature of stress fibers is that they connect focal adhesions, thus they are confined to the lower planes of the cell: coverslip interface. I think it would be a good idea to compare the retrograde flow of these structures using TIRF, not spinning disk as they do.

2) In Figure 2, I am fine with the effect of CK-666 in the localization of the Arp2/3 complex (as measured by anti-p34 staining, I assume; it'd be useful to have this stated in the figure legend). However, I am surprised by the lack of effect on the shape of the leading edge, which is very similar. I would like to see kymographs of the leading edge in control and CK-666-treated cells (similar to Figure 3F, but also showing the extension of the leading edge).

3) My main issue with this study pertains to the interpretation of Figure 4E. While I agree some sarcomeres can be seen in Myh9-depleted hiCM cells, the difference between scramble and Myh9 siRNA is too striking to be described as a mere "disorganization" issue. As such, the metric used in Figure 4G is very misleading. I remain convinced Myh9 plays a role in this process, and this description does not adequately address this. To confirm these data, I would like to see data using p-nitroblebbistatin (so they can do dynamics) and a better way to represent the disorganization caused by Myh9 depletion.

4) Figure 4A, the localization of NMII-A and NMII-B is described as equivalent, yet NMII-B remains "stubbornly" behind NMII-A, which is okay. Although I agree that the existence of co-polymeric forms of the two isoforms reflect the existence of a transitional area, the image in Figure 4A and the quantification shown in Figure 4B are dramatically different and may lead to confusion.

5) I do love the experiments depicted in Figure 5. However, I am confused as to their meaning. In the original paper from the Adelstein group (Tullio et al., 1997), a major issue was the size of the cardiomyocytes, which seems unaffected here. A similar result was seen with the conditional II-B/-C ablation in the Ma et al. paper from 2009. Is the size of the cells significantly larger in these experiments?

6) How is betaCMII organized before sarcomeres form? Does it form mini-filaments? In the same direction, if betaCMII is depleted, does NMII-B distribute evenly throughout the cytoplasm? Or is it still confined to the leading edge?

Reviewer #3:

This paper addresses some very interesting questions regarding sarcomere formation using a nice model cell system. While some of the data looks quite good, a considerable amount has significant issues with quantitation, lack of depth, and/or interpretation. Below are my specific comments.

I am surprised that there is no change in hiCMs after CK666 treatment (Figure 2) other than a reduction in Arp2/3 localization at the cell edge. In most cells Arp2/3 is consuming the bulk of the monomer at steady state. Is there a change in total actin content? Retrograde flow rate? Ability to spread? Do the formin-dependent MSFs get more robust because of the increase in monomer supply when Arp2/3 is inhibited? Of note, their CK666 treatment displaced only ~50% of the Arp2/3 signal from the edge. The remaining Arp2/3 signal/activity could well be enough to drive a significant fraction of normal branched nucleation. I wonder if CK666 works in their cells as expected, and/or if the CK666 in the media after 24 hours of imaging is still active? Does robust KD of any Arp component give the same result? Basically, this aspect of their work needs additional effort before the authors can categorically exclude a contribution from Arp23/ to MSF formation.

I recommend in subsection "Formins but not the Arp2/3 complex are required for MSF-based sarcomere formation…" that the authors move the fourth paragraph in front of the third paragraph. When I read the second paragraph about how SMIFH2 blocks sarcomere formation, I immediately wondered about the MSFs, given that they are supposed to give rise to the sarcomeres. But before I got the answer, I had to read about FHOD3. This seems out of order. In addition, the authors should determine FHOD3's localization relative to MSFs and sarcomeres to see if it supports their idea (and to provide some additional information on FHOD3's role in sarcomere formation since we already know from previous work that sarcomeres disappear when it is knocked down).

The SMIFH2 data in Figure 3 needs a time lapse movie to show how MSFs and sarcomeres disappear, and it needs a washout experiment (preferably with movie) showing the SMIFH2 treatment is readily reversible in these cells.

I think they need at least more incisive experiment to support the idea that MSFs give rise to sarcomeres. One good way to accomplish this would be to use photoactivation or photoconversion to put a fiducial mark in MSFs and then show that the mark persists for some period of time in nascent sarcomeres. This should be readily doable.

The authors use the terminology "MSFs acquire sarcomeres" in various places in the text. To me this does not describe what I think the authors are trying to convey, which is that MSFs transition into sarcomeres or MSFs template sarcomere assembly or something like that. I encourage the authors to think more carefully about their wording.

I am not sure the idea that MIIB is required for sarcomere formation but not for sarcomere maintenance makes sense. I assume that, like everything else, these structures are constantly turning over, so if one blocks sarcomere formation, sarcomeres must disappear over time because of turnover (or the system is more complicated than they think).

I have several significant issues with the data in Figure 4 as regards MIIA KD. First, the authors need to show the level of KD. Without that, one can have confidence in the conclusions. Second, the statement "KD of MIIA did not result in inhibition of MSF or de novo sarcomere assembly" appears to be at the very least a major over-simplification. First, it looks to me like the MSFs are largely missing in the KD cell shown in the middle panel in Figure 4E. Moreover, no quantitation is provided to support the claim that the "KD of MIIA does not result in an inhibition of MSF formation". Second, the effect on sarcomeres looks to be much greater that just disorganization as they call it. Indeed, the scoring for sarcomere assembly in Panel G (percent cells with sarcomeres) is not adequate. Something much more quantitative is required (e.g. total sarcomeres per cell, or per unit). It is my guess that proper quantitation of this data will alter their conclusions significantly.

What I see in Figure 4A and B is very different from what the authors conclude. The authors say that, unlike mesenchymal cells where MIIA is peripheral and MIIB is more central, these two isoforms are in the same place within MSFs of spreading hiCMs. To me, Figure 4A shows very clearly that MIIA is enriched closer to the leading edge than MIIB, exactly as seen in mesenchymal cells. I would need to see simultaneous two-color imaging to be convinced that the distribution MIIA and MIIB in hiCMs is different from that in mesenchymal cells. Similarly, the left most SIM mage in Figure 4C clearly shows (especially on the right side of the cell) mostly red (MIIA) nearest the cell edge, then white moving in (presumably overlap between MIIA and MIIB), and finally mostly green (MIIB) furthest in. My guess is that better analyses and quantitation of all the data in Figure 4A-4C will alter their conclusions significantly. Finally, I suggest the authors determine the ratio of MIIA/MIIB within individual filaments going from the cell edge inward (as in Beach et al., 2014).

I wonder if the MIIB KD effect is the result of not having MIIB or of removing much of the type II myosin in these cells. For example, if MIIB represents 90% of total MII in these cells, then it is possible that the KD effect stems from removing the vast majority of type II myosin, and not specifically MIIB. To address this, the authors need to determine the ratio of MIIA to MIIB (and MIIC?) in these cells. Along these lines, can overexpression of MIIA rescue the MIIB KD phenotype?

I was not clear to me that their quantitation of control and MIIB KD zebrafish hearts is representative because the high mag view of the KD tissue stained with α-actinin in Figure 5B looks pretty much like the high mag view of the control tissue in Figure 5A. In other words, they don't appear to differ by anything like ~3-fold, as presented in the quantitation in Figure 5D. Regarding Figure 5D, I assume that this data was taken from the boxed areas in the images stained for actin in Figure 5C. If I am correct, then this is not appropriate for two reasons. First, the data needs to be collected in a nonbiased way, i.e. not by picking areas but by scoring the entire sample area. Second, because sarcomeres are much clearer in the α-actinin-stained images than in the actin-stained images, the measurements need to be made using α-actinin staining. My guess is that proper scoring of this data will alter the conclusions to some extent.

Regarding Figure 6, the authors say, "We noted that near the leading edge of the cell, BCMII filaments were smaller and not organized into stacks resembling A-bands (Figure 3C)". I do not see data on filament size in Figure 3C. Regarding the conclusion that mixed MIIB/BCMII filaments populate a specific region behind the leading edge, the authors need to provide high mag insets and quantitation of double-stained, SIM imaged cells to support this conclusion. How well was MIIB knocked down in Figure 6F and 6G? I found the conclusion from the latrunculin data- "MSFs and sarcomeric actin filaments are serving as "tracks" with which BCMII filaments are loading onto as they from larger BCMII filament stacks of the A-band"- to be a huge over statement of the data. That things don't work when you blow up all the actin does not even remotely prove this elaborate mechanism. This would require at a minimum very detailed dynamic imaging of the normal maturation process.

Regarding Figure 7, the authors state that they do not "believe MII/BCMII co-filaments are not the major mechanism through which MIIB is facilitating BCMII filament stack assembly". Later, however, they state that "MIIB may be serving as a template to seed the formation of BCMII filaments". I don't see anywhere in the text that the authors clearly articulate what they think the role of MII is in this process. If it is not contributing as a co-filament/template, then what is it doing? If it is contributing as a co-filament/template, then how is their thinking different from the Sanger model? Is there a reason the authors cannot image dynamically MII and BCMII at the same time? It seems to me that the process is slow enough that doing multi-channel imaging would work. Doing this would greatly strengthen this central part of the paper. For example, it would show whether BCMII filaments always appear on MIIB filaments, or if they appear independently? Also, if the authors are looking to combine two previous models, then maintaining those names is more appropriate (I suggest the Sanger-Stitch Model).

In the end I did not get much sense how MSFs transition/mature into sarcomeres. Some additional insight into this would strengthen the paper considerably. For example, where is capping protein in this process? At some point, I would expect CapZ to incorporate to build a functional sarcomere. Can the authors shed some light on this part of the process with some staining and/or dynamic imaging of capping protein?

Please use pagination. Please drop the italics on "after" in the third paragraph of the Discussion section. The paper contains a lot of missing words, repeated words, etc. The Introduction could be made significantly more readable with some effort.

Finally, after the submission of this manuscript, a paper appeared in Dev. Cell from Chris Chen's group addressing this same topic. Notably, they came to very different conclusions regarding a role for NMII. While I think disagreement is very often good for science, I would expect any future versions of this manuscript to contain extensive discussions about these differences (and similarities) and possible explanations for the discrepancies.

[Editors’ note: what now follows is the decision letter after the authors submitted for further consideration.]

Thank you for submitting your article "Muscle specific stress fibers give rise to sarcomeres in cardiomyocytes" for consideration by *eLife*. Your article has been reviewed by Anna Akhmanova as the Senior Editor, a Reviewing Editor, and three reviewers. The following individual involved in review of your submission has agreed to reveal her identity: Velia Fowler (Reviewer #2).

The reviewers have discussed the reviews with one another and the Reviewing Editor has drafted this decision to help you prepare a revised submission.

Summary:

The revised manuscript is much improved with many new experiments, and substantial rewriting, so that it now presents a coherent story that is well-grounded in the historical context (presented very nicely in the Introduction), answering long-standing questions about the roles of NMIIA and NMIIA in sarcomere formation, using elegant live cell and super-resolution light microscopy approaches in human induced pluripotent stem cell cardiac myocytes (hiCM). The response to reviewers is very thorough and thoughtful. The authors have now used Lifeact-FPs and α-actinin2-FPs to study the transitions of the 'muscle stress fibers' (MSFs) into the myofibrils as the MSFs move dorsally and used the α-actinin2 spots/Z line distances to quantify sarcomere formation very convincingly. They also use photo-activation of Eos-α-actinin2 to show that the α-actinin2 dots in the in the MSFs become Z lines in the sarcomeres. The work shows that the two non-muscle myosin IIs are both important but play somewhat different roles in sarcomere and myofibril assembly, and also that formin FHOD3 but not Arp2/3 actin nucleation is required for sarcomere assembly. The controls are all in place and the conclusions are strong. The authors also show that NMIIA and NMIIB can form heteropolymers with the β-cardiac myosin II (betaCMII) in the cultured hiCMs, a novel finding. The comparisons of the behavior of the arc-like MSFs in the hiCMs with the actin arcs in spreading non-muscle cells such as U20S cells are also interesting and important to the field. Their careful presentation of the outstanding questions and their own new data in a historical context is valuable and balanced.

Essential revisions:

1) While most concerns have been addressed by the substantial, and high quality new experiments and extensive rewriting, there were still a few concerns with the figure presentation and some of the interpretations. Most importantly, the major messages of the manuscript (MSFs are precursors to sarcomeres, NMII participates in sarcomere formation in a precursor role) became somewhat diluted in the revised version. A few conclusions are also somewhat overstated. Below, we suggest improvements to clarify and strengthen the results and conclusions. These can all be addressed by minor revisions to figures and text.

2) It appears from the data shown in the paper that the new sarcomeres are formed by rearrangement of the MSFs, starting at the spreading edge of the cell. The preexisting sarcomeres present before replating appear to be in the center of the replated cells so that they do not play a role in the formation of the new sarcomeres. Since this is not entirely clear, the authors could provide better 3D imaging of a newly replated cell to show where the preexisting sarcomeres are, and their relationship to the newly forming sarcomeres at the edge of the cell.

3) Based on the photoconversion experiments using α-actinin-2, if the authors were to do similar experiments photoconverting NMIIA and/or IIB, where would it go? This is important because of their observation that NMII-A and IIB do not incorporate into the sarcomere. It would be nice, if possible, to address this point experimentally.

4) The zebrafish data is rather weak, and furthermore does not address the main message that "MSF are precursors to sarcomeres, NMII participates in sarcomere formation in a precursor role", and thus it can better be omitted.

Suggestions for revisions of text and figures:

1) Figure 6, Panel E. Very nice data, but the images would be clearer if the individual channels were shown in gray scale, with a merge for both the low and high mag crops included. Also, the high mag crops on the right would be better presented horizontally (i.e., time along X axis), so that we can follow the assembly over time more easily. The authors should comment on their intriguing observation that the FHOD3 appears to assemble into a pair of widely spaced stripes in the middle of the sarcomere. It is not at the Z line, based on the F-actin staining, nor at the pointed end - which would be one stripe in the middle of the sarcomere in the unstretched sarcomeres of the cultured cardiomyocytes. Is the FHOD3 staining similar to previous studies of FHOD3 localization? Or is it new? It looks as if it is in the region of A-I junction, near the locations of the myosin heads. This could be commented about.

2) Figure 8. Very interesting experiments! The effects of MYH9 knockdown versus MYH10 knockdown are very different. While they both are important for sarcomere/myofibril assembly, the MYH9 KD looks more important for lateral myofibril alignment – the KD cells have many branched and wispy looking myofibrils, although they clearly can make sarcomeres; the quantification shows this too. However, the MYH10 KD cells don't make any sarcomeres at all, so NMIIB is required for all (initial?) aspects of sarcomere assembly, while NMIIA is not. Figure 8—figure supplement 3 and Figure 8—figure supplement 4 time-lapse images show these differences very clearly also. The results and other parts of manuscript should be rewritten somewhat to clarify and emphasize these intriguing differences between the NMIIA and NMIIB which are very interesting. Have the authors looked at MSF retrograde flow in either of the knockdowns? This could be interesting (but not required). It was difficult to find any mention of this.

3) Figure 9. The zebrafish data remains unconvincing to me. One can find many areas of sarcomeres in the low mag images in the top panels, both in the myh9bMO and myh10MO hearts. The high mag images in the lower panels are blurry. Moreover, the myofibrils curve in and out of the XY plane in these cells in the heart, and thus some extend in the Z dimension, making it complex to identify all the sarcomeres due to the Z stretch in the confocal. Also, if the myofibrils are even a tiny bit more contracted, then sarcomeres can be very hard to identify from F-actin staining. α-actinin labeling could help. Finally, even if the images and quantification were reliable (which are not convincing to me), the experiment does not address the central point of the study, which is the role of NMIIA and NMIIB in the transition of MSFs to myofibrils containing sarcomeres. There are no clearly distinguishable MSFs in these zebrafish heart cells. We suggest the authors remove this data and save it for a future more extensive study where they use live cell imaging to study the MSF – sarcomere transitions in vivo- probably need lattice light sheet.

4) Figure 10. Panel H. These odd structures that form in the MYH10 KD cells are intriguing. It appears that rods of betaCMII extend between large donut-like foci of F-actin. Where is the α-actinin2? Is it in the center of the F-actin donut? It was speculated in the previous comments that they might be podosomes, and the response to reviewers provided a clearer explanation of why this is unlikely, based on where they are in the cell and what else they may contain. Some this explanation could be incorporated into the manuscript to make the figure clearer to the reader. We are not asking the authors to do more experiments, but at the least they should describe these images more precisely and then speculate about what the structures might be. Also, the betaCMII rods extending between the F-actin donuts could be individual A bands; are they included in the quantification shown in J? Or are A bands only counted if they are in a linear sequence along and F-actin bundle? Panel H high mag panels need a scale bar.

5) Figure 11. Panel I. The image from the Latrunculin-treated cell is confusing. Where are the cells in this image? What is the large fluorescent half-moon at the bottom of the image? Is this an accumulation of β-CMII around the nucleus or center of cell? Are there A bands in this densely-stained area – it looks like there might be. A counter-stain with a nucleus marker or cytoplasmic marker could be helpful. Another question is what type of betaCMII structure is in the little box? Are these 'stars' related to the donut like F-actin structures in panel H? i.e., would F-actin be in the hole in the middle? Was Panel J quantified from I? If so, then wouldn't it make sense to show both the betaCMII and the F-actin staining for the Latrunculin treated cells?

6) Figure 10. The authors' demonstration of NMIIA or NMIIB co-assembly in bipolar filaments with betaCMII in the cultured hiCMs is convincing. However, the reviewers were still not completely convinced by the claim that they can find these co-assembled filaments in vivo in the mouse heart. The heart tissue sections in Figure S3B and Figure S18 are a low mag field of many myofibrils (not sure why Figure S18 is a separate figure, also no scale bar in Figure S3B). Both show that the NMIIB staining (green) blobs are excluded from the myofibrils and squeezed in between the closely packed myofibrils. Are these supposed to be periodic NMIIB filaments located along an MSF type of structure? Or aggregates? One would like to see a zoom in of the blobs region so we can see the NMIIB and betaCM colocalized (along the putative MSFs?) in the context of the heart tissue, ie, we need an intermediate magnification field within which one can find the filaments! Not just the super-resolution images of individual filaments cropped out, as shown.

7) Figure 9—figure supplement 3A. The betaCMII in the siRNA MYH9-treated cells looks remarkably OK. One can see lots of A bands. However, they appear to be lined up along the length of myofbrils that are very thin so that the betaCMII stacks are not evident. It would be nice to show the F-actin colocalization with the betaCMII staining. This also fits with the data in Figure 8 showing that sarcomere formation in MYH9 KD cells is not as disrupted as in MYH10 KD cells (see point 2 above). Presumably the fourier transform does not reveal the periodicity as well as for the controls in Figure 10G, due to the wispy myofibrils and reduced alignment of the stacks. But a line scan along the length of the myofibrils would likely still reveal periodicity. We are not asking for authors to do new experiments here (unless they want to!), rather just tone down interpretation of MYH9 KD images in Figure 9—figure supplement 3, as related to comments in point 2 above about NMIIA function.

---

## [Author Response]

[Editors’ note: the author responses to the first round of peer review follow.]

Our decision has been reached after consultation between the reviewers. Based on these discussions and the individual reviews below, we regret to inform you that your work will not be considered further for publication in eLife.In this paper, the authors investigated sarcomere assembly in human iPSC- derived cardiomyocytes and provide novel molecular insights into this process, which are potentially of broad interest. However, all three reviewers felt that there is a very significant number of issues that need to be addressed before this paper can be published. In particular, the referees found that some data were incomplete and some controls missing, that important quantifications were lacking, and that certain data needed to be analyzed differently. Furthermore, a number of points had been raised that would require addition of new data, such as evaluation of the spacings and dynamics of α-actinin puncta, two-color imaging of myosin II and β cardiac myosin II, and photoactivation experiments to link muscle stress fibers to sarcomere assembly more convincingly. We return the paper to you, hoping that you will find the detailed comments of the reviewers useful for improving your study. However, we will be prepared to consider a new submission, which would fully address the comments of all three reviewers. In case you decide to resubmit your paper to eLife, it will be sent to the same Reviewing Editor, who will then decide, based on your revisions, whether the paper warrants re-review and whether the same reviewers should be consulted.Reviewer #1:In this manuscript Fenix et al., have reported a novel phenomenon in sarcomere assembly in human IPS derived cardiomyocytes (hiCM), which builds on prior knowledge in the field. Using confocal, super-resolution and live cell imaging assays in hiCM, the authors demonstrate that sarcomeres assemble from F-actin stress fibers organized in arcs near the edge of the spreading cells, that they term muscle stress fibers (MSF). These resemble actin arcs in spreading cells from HeLa and U2OS cell lines but have quantitative differences. They showed that retrograde movement of the MSFs results in maturation of sarcomeres (i.e. formation of striated myofibrils) and that the rate of retrograde flow of MSFs is significantly slower than that of F-actin arcs in the non-muscle cell types. These observations of myofibril assembly in the hiCMs are overall similar to a number of previous studies using fluorescent-tagged α-actinin to image the transitions of stress fiber-like structures (SFLSs)/pre-myofibrils into myofibrils in primary cardiac myocytes, but numbers and rates have now been quantified here (note the terminology is also different here).

We thank the reviewer for pointing out a number of strengths of this manuscript and acknowledging some of what we have added to the question of sarcomere assembly. We would like to point out that this is the first study we know of to visualize the first sarcomere/myofibril formation event. As the reviewer points out, there are classic studies with montages showing puncta of fluorescently tagged α-actinin 2 joining pre-existing Z lines in myocytes (Dabiri et al., 1997; McKenna et al., 1986). In the original manuscript, we only cited (Dabiri et al., 1997) on this point but, upon further reflection, believe that (McKenna et al., 1986) paper should be cited also. Per all three reviewers’ suggestions below, we now build upon these classic results using photo-activation of α-actinin 2. In addition, to the best of our knowledge no study has visualized the actin cytoskeleton or myosin II during the transition from non-sarcomeric organization to sarcomeric organization in live cells as we show here. The reviewer is correct that many studies (which we referenced in the manuscript) have localized components of putative precursor structures in fixed cells and tissues, but no one has visualized their transition to sarcomeres in live cells as we do here.

We are using the term “MSFs” for two reasons. We apologize for not making this clear in the original version of our Introduction. First, they were named and acknowledged as “stress fiber like structures” by Howard Holtzer years before they were renamed “pre-myofibrils”. We are aware that a portion of the cardiovascular cell biology field uses pre-myofibrils as if the Pre-myofibril Model has been thoroughly tested (Sanger et al., 2005; Sanger et al., 2017). There are just as many, if not more groups that promote other models, such as the Stitching model, as if it too has been thoroughly tested (Chopra et al., 2018; Rui et al., 2010). As such, we feel that calling these stress fibers pre-myofibrils is premature and may lead to those that do not “believe” in the Pre-Myofibril Model to dismiss our results outright. Thus, we have chosen to revert back to the simple term “stress fiber” and in the case of this particular manuscript “muscle stress fibers” as we need to distinguish between seemingly similar stress fibers in non-muscle cells and muscle cells. We believe future work which is not directly comparing non-muscle and muscle cells should simply call these stress fibers. This affords an unambiguous discussion of the mechanisms of sarcomere assembly, as using terms such as “pre-myofibril”, “I-Z-I bodies”, or “floating A-bands” implies pre-conceived notions of how the system functions (Holtzer et al., 1997; Rui et al., 2010).

Finally, distinguishing between these models is not a trivial point, especially in light of the recent work by (Chopra et al., 2018). This manuscript was published in a top tier journal and failed to properly introduce or discuss any previous models of sarcomere assembly, such as those mentioned above. Indeed, they present what is more or less the Stitching model but with the added twist of sarcomeres streaming from the adhesions. Although, this in itself is reminiscent of the previous proposal from (Quach and Rando, 2006) in skeletal muscle- also not cited by (Chopra et al., 2018). We of course are not addressing the relationship between adhesions and myofibrils in our current manuscript as it is well outside the scope of this study and is an ongoing project in our lab. We merely mention this in the response to emphasize that a lack of discussion of the previous models leads to proposals of “new” models. We feel this simply muddies an already dense and confusing literature. As such, we have revised our Introduction to introduce the Major models in the fields more explicitly and in the Discussion section, put our findings in this context.

In a new advance, they go onto show that the formin FHOD3 rather than Arp2/3 plays pivotal roles in MSF directed myofibril assembly/sarcomere formation (by contrast, Arp2/3 is important for assembly of non-muscle cell F-actin arcs). In addition, they demonstrate novel functions for NMIIA versus NMIIA in myofibril assembly, showing that in the MSF to sarcomere transformation process NMIIB is necessary, while NMIIA appears to play a redundant role, using an acute knockdown approach. Finally, the mechanism of β-cardiac myosin II thick filament assembly into A bands was also studied, and live cell imaging was used to evaluate the mechanism of thick filament assembly into sarcomeres in striated myofibrils. Through these experiments Fenix et al., have convincingly shown that MSFs (which likely correspond to previously-termed SFLSs or pre-myofibrils) are precursors for maturing sarcomeres and require a different non-muscle myosin isoform as compared to the arc-like stress fibers found in non-muscle cell lines.

To clarify, we have never claimed that the MSFs were not corresponding to the previously described structures. We are grateful to the reviewer for pointing out that one could interpret our wording in a way that we are claiming novelty. Our intent here is to build upon the work of others, which we cited in the previous version of the manuscript and are now more explicit in our Introduction and Discussion section so that the reader understands the previous work and our perspective.

However, some data are weak and do not add to the main points (e.g., the zebrafish data), and there are some missing pieces in the data (e.g., lack of direct comparisons to the previous literature using α-actinin2 as a marker for myofibril assembly, questions regarding location of FHOD3, potential myosin isoform compensation), that need to be addressed to place this study in the context of previous work. Many of the figures also need to be improved to strengthen the points and clarify some ambiguous results.

We thank the reviewer for pointing out a number of points that need addressing. Doing so has made our manuscript stronger. We address these points in depth below.

Main points:1) The authors claim in the Introduction that the hiCM system allows them to study de novo myofibril assembly in vitro for the first time, unlike the previous primary cardiac cell cultures from embryonic chick hearts or neonatal mouse hearts. However, Figure 5 shows that the hiCM cells contain sarcomeres before replating and spreading! Thus, the hiCM system is actually quite similar to the primary cell systems, which disassemble and then reassemble sarcomeres after replating and spreading.

We partially agree with the reviewer on this point and have revised our language about the primary cultures of embryonic cardiomyocytes. We have removed “de novo” from the manuscript. We did not mean to imply that primary cardiomyocytes (e.g., embryonic) do not form sarcomeres de novo after isolation and re-plating. They probably do, and we suspect that several groups have unpublished data showing this point. However, the data in the published literature using these systems to visualize and test the mechanisms of sarcomere assembly has thus far only provided data after cells contain sarcomeres. Here, we utilize a re-plating protocol, which was not trivial to establish, to visualize the first sarcomeres assembled in live cardiac myocytes. Nevertheless, we have softened our language in the current version of this manuscript considerably on this point.

Of note, the recent report by (Chopra et al., 2018) has also used the term “de novo*”* to refer to sarcomere assembly throughout their manuscript. As discussed at the end of this response, we believe there are significant problems with (Chopra et al., 2018) paper and they are, indeed, not showing the first sarcomeres that assemble in hiCM. Nonetheless, we have removed the term “de novo” in lieu of more descriptive wording.

Nevertheless, the hiCM cells are a great system, and the authors present some interesting and novel findings regarding the roles of actin arcs and myosin isoforms in myofibril assembly. The Introduction and Results section should be rewritten and shortened to emphasize the new work on formins and myosin isoforms.

As implied above, we have re-written the Introduction, Results section and Discussion section of the current manuscript to more clearly emphasize our results within the context of the larger literature.

2) There is another very important point regarding previous work on myofibril assembly. The authors state that they are unable to find I-Z-I bodies in the hiCMs, but I can clearly see small α-actinin puncta along the F-actin arcs in Figure 2A. The authors may have been misled by the vast and confusing literature; I-Z-I bodies are simply small α-actinin2 puncta linearly arranged along F-actin bundles that stain continuously for F-actin.

We agree with the reviewer that there is little evidence in the literature as proposed in the Stitching model. Indeed, others seem to find this confusing as well as the localization of α-actinin 2 puncta has been used to discuss both the Pre-myofibril (Kan et al., 2012) and Stitching Models (Rui et al., 2010). Therefore, we have chosen to test the prediction of relative positions of assembling sarcomeres as proposed by the Templating/Pre-Myofibril Models and Stitching Model. There is no discussion of I-Z-I bodies in the current Results section or Discussion section of our manuscript.

These do appear to be present in the hiCMs. The authors should reexamine their control (untreated) hiCMs and present some high mag images of the cell edge and the F-actin arcs stained for α-actinin2. How does the spacing of α-actinin2 puncta in the arcs/MSFs of hiCMs compare with the literature on myofibril assembly in other cardiac cell types? The distances between the α-actinin2 puncta should be measured in the various experiments, including the formin and myosin KD experiments.

We thank the reviewer for this comment as it has substantially strengthened our manuscript. In the original submission of our manuscript, we did not want to exhaustively characterize the α-actinin 2 in MSFs of hiCMs, as this has been thoroughly characterized in other systems as the reviewer mentions. We now provide thorough characterization of α-actinin 2 spacing in hiCMs in control (untreated), and all experimental conditions. This has not only led to a more thorough characterization and quantification of MSFs, but also a more complete understanding of the effects of our perturbations. Most importantly, these additional quantifications will more easily allow others to repeat our experiments and compare their own experimental results to what we present here.

It is important to compare the so-called MSFs in this study with the various so-called "SFLSs" and "pre-myofibril" etc structures studied by other groups. Most of the live imaging in previous work used fluorescent-α-actinin2 to study formation of myofibrils; thus, one would like to know whether this study obtains similar results.

We are confused. Before submission of our original manuscript we conducted a thorough review of the literature. The only live cell examples we could find of live-cell imaging of α-actinin 2 showed cells that already robust Z-lines in the first frame of the montage (Dabiri et al., 1997; McKenna et al., 1986). We also cited one of these papers in the Introduction of the previous version of our manuscript:

“Though a pre-myofibril structure containing actin, NMIIB, α-actinin has never been directly shown to transition into a sarcomere containing myofibril, previous live cell data using fluorescently tagged injected α-actinin has shown “Z-bodies” at the edge come together as the Z-bodies undergo retrograde flow and assemble into existing sarcomeres (Dabiri et al., 1997).”

We believe we have substantially clarified this issue in the current version of the Introduction. Furthermore, we would be grateful to know if we are missing or overlooking literature showing the first sarcomeres assembling in live cells. Of course, there are many images of fixed cells and tissues that were used to predict the Template and Pre-myofibril models.

For example, closely-spaced α-actinin puncta can be seen transitioning into the wider striations in the myofibrils as they move inwards at the spreading edge of a chick cardiac myocyte in Figure 5 in Dabiri et al., 1997. I think these could be the arcs that are shown here in the hiCMs. There are other types of examples in the literature.

We agree with the reviewer that the mentioned example showed nicely that α-actinin 2 puncta transitions into the wider striations of the myofibrils. The reviewer is also correct that in our system the smaller α-actinin 2 puncta do localize to MSFs while the larger α-actinin 2 Z-lines localize to myofibrils. We have now included photo-conversion experiments of α-actinin 2 to repeat and expand on the results of Dabiri et al. (New Figure 2). Again, we never meant to make it sound that MSFs were novel structures.

3) As a related point, what is the direct evidence that the F-actin arcs in the hiCMs are similar to the F-actin arcs in non-muscle cells? The first paragraph of the Results section states: "Strikingly, super-resolution imaging revealed the MSFs in hiCMs resembled a classic actin stress fiber found in non-muscle cells, referred to as actin arcs (Figure 1C and 1D)." I expected the figure would show high mag images of the hiCM arc substructure compared to the non-muscle cell arcs, but this figure shows relatively low mag images of the arcs in U2OS and HeLa cells, revealing continuous F-actin along the fibers in all cell types. How does this show that the MSFs are similar to the arcs in the non-muscle cells? We need to see some high-resolution images of the arcs in the hiCMs and the non-muscle cells--for example what are the spacings between the Z bodies (α-actinin puncta) in the arcs in each cell type? I would like to know whether the NMIIA and NMIIB are located in between the α-actinin2 puncta of the MSFs, which would also be similar to the SFs in the non-muscle cell types (but with a different α-actinin isoform).

We thank you for this comment. There are several similarities between actin arcs and MSFs that we failed to articulate. First, they both exist on the dorsal surface of the spreading cells. They stain along their lengths continuously with phalloidin indicating that there are few if any regions without actin. They are also both arranged parallel to the edge of the cell and move in a retrograde manner away from the edge. We have substantially revised Figure 1 and first part of the Results section to show evidence for these points.

4) Actin arcs are shown in the live cell imaging experiments to be precursors for "myofibrils containing sarcomeres" (note- this is a more precise terminology- a sarcomere is a small unit of a myofibril- and the arc may have a small actomyosin 'mini-sarcomere' type structure similar to the stress fibers of non-muscle cells). Can other types of actin stress fibers like dorsal and ventral stress fibers also mature into sarcomeres in hiCM? Or, do these hiCM cells only possess actin arcs? This seems unlikely, as I think I can see some radial stress fibers in the cells in Figure 3E and in Figure 1H near the edge.

There are several actin filament-based structures in hiCMs that we are not focusing on in this particular manuscript. Our data does indicate that there are similar structures in spreading hiCMs to the three major types of stress fibers in non-muscle cells: actin arcs (sometimes called transverse arcs), dorsal stress fibers and ventral stress fibers. With that said, we have not detected sarcomeres arising from either the dorsal stress fiber-like population or the ventral stress fiber population. It is important to note that our manuscript focuses on the events that occur before the hiCMs have been on the plate for 24 hours. There are several interesting changes that occur post 24 hours. Sarcomeres do appear on the bottom of the cell and sometimes along what appear to be dorsal stress fibers. Ongoing work in our lab is addressing these changes. In the current manuscript we do show two examples of sarcomeres appearing on the ventral surface of the cell and is more thoroughly discussed in response to reviewer 3’s final point.

5) The authors have shown that FHOD3 knockdown in hiCM cells results in reduced sarcomere formation. However, they have not shown the cellular distribution of FHOD3 by immunofluorescence, unlike Arp2/3. Where does FHOD3 localize in the hiCM in the initial MSF stage, and later when the sarcomeres are formed? Does FHOD3 directly interact with the MSFs and/or the sarcomeres?

We thank the reviewer for this point. Yes, FHOD3 is localized to both MSF and sarcomeres (Figure 6E). We now include localization of a FHOD3-mEGFP construct, which was a gift from and previously published by Dr. Elizabeth Ehler’s group.

6) The authors use a pan-formin inhibitor, SMIFH2, to show that formins are important for formation of the actin arcs and sarcomeres. While FHOD3 is clearly important, their experiments do not rule out roles for other formins. For example, see Rosado et al., 2014. Also, the authors claim that mDia1 is not involved in the MSFs, but this has not been tested here. This could be tested, or at the least, discussed and clarified.

We have used RNAseq to identify the formins expressed in the hiCMs used in this study (Figure 6A). As we report in the revised version of the manuscript, there are three highly expressed formins in hiCMs; FHOD3, DAAM1, and mDia1. These are actually different from the most highly expressed Formin paralogs in mouse hearts as reported by Rosado et al. We now provide a discussion of these results in the Discussion of the current manuscript. Furthermore, we have knocked down both DAAM1 and mDia1. Both knockdowns produce clear phenotypes, which warrant future investigations (Figure 6—figure supplement 1).

We did not mean to imply that mDia1 is not involved in MSFs. The only mention of mDia1 was as follows in the Discussion section:

“However, we found that the formin paralog, FHOD3, was responsible for MSF and sarcomere assembly, whereas in non-muscle cells mDia1 is required for actin arcs (Murugesan et al., 2016).”

We do understand how this sentence could appear that we are implying mDia1 is not required for MSFs or sarcomere assembly. That was not our intention and it has been removed.

7) NMIIA and NMIIB can form heteropolymers in hiCM, yet NMIIB knockdown alone disrupts sarcomere formation. Is this because there is a compensatory rise in NMIIB expression upon NMIIA knockdown, which restores the total NMII levels in the cell? However, another possibility is that the NMIIA levels are not enhanced upon NMIIB knockdown, resulting in a drop of cellular total NMII levels, bringing about a disruption of sarcomere assembly. The authors should examine the NMIIA levels upon NMIIB knockdown, and vice versa, to address this issue. It would also be helpful to provide a quantitative estimate of NMIIA/NMIIB distribution in the soluble and insoluble fractions of hiCM cells during the MSF and sarcomere stages. This may confirm why one isoform is more important than the other in sarcomere formation.

We thank the reviewer for these points. Of note, there has been extensive changes to our interpretation of the results from the NMIIA KDs in hiCMs based on comments from all three reviewers. We now provide quantifications which suggest NMIIA is playing major roles during sarcomere assembly in hiCMs (Figure 8). We also provide protein level changes of NMIIA levels upon NMIIB knockdown and vice-versa (Figure 8—figure supplement 2). The reviewer’s comments on further investigating the mechanism of how NMIIA and NMIIB drive sarcomeres formation is a good one. We have begun another project mapping the domains of each myosin that is required. While overexpression of NMIIA does not rescue we have found that we can rescue with only the motor domain of NMIIB (i.e., a chimera with the rod of NMIIA and the motor of NMIIB). Immuno- fluorescence was used to confirm that endogenous NMIIB was knocked down in each expressing cell. While these results are interesting it has become clear to us that the role of each domain and specific levels of each in filaments and the cytosol is beyond the scope of our current study. The reviewer is asking for at least one (maybe more) follow-up studies to be added to an already massive manuscript.

8) Although the results show that NMIIB is essential for de-novo sarcomere assembly in spreading cells, is it also important for the maintenance of the sarcomeres?

No, our data would suggest that neither NMIIB or NMIIA are required for sarcomere maintenance, at least over the course of 11 days in culture. We provided this data for NMIIB in Figure 5A of the previous version of this manuscript. Due to comments from all reviewers it was clear that this was a very confusing way to present this data, as it was included in the same figure as the Zebrafish data. The maintenance data is now included in Figure 8F and Figure 8—figure supplement 6. While our data suggests there may be separate mechanisms of assembly and maintenance, which is not necessarily a surprise, we feel it is outside the scope of this current manuscript to delineate these mechanisms.

9) I appreciate the authors' efforts to study NMIIB in vivo, but the experiments are not convincing. The zebrafish morpholino experiments are potentially exciting but are underdeveloped and missing important controls. What is the result of MO for MYH9 (NMIIA)? MO of the MYH10 (NMIIB) could also affect heart development via effects on non-cardiac cell types, e.g., vascular endothelium. A rescue using a cardiac-specific MYH10 transgene would address this. Also, MOs often have off target effects, and CRISPR approaches are now used to deal with this issue. Due to these issues, I recommend omitting the zebrafish experiments and following this up in a future more complete study.

We disagree the zebrafish data should be omitted. As mentioned above, comments from reviewers 2 and 3, caused us to re-evaluate the role of NMIIA during sarcomere assembly.

This included knocking down NMIIA via MO in zebrafish as reviewer 1 suggests here. NMIIA KD animals failed to form sarcomeres as assessed by Z-lines which can be clearly seen with an actin stain (current manuscript Figure 9). We also provide more thorough quantification of both NMIIA and NMIIB KD hearts. These included number of sarcomeres, Z-line lengths, and the persistence of sarcomeres along myofibrils for all Zebrafish conditions (current manuscript Figure 9). While the suggested rescue experiment using a cardia-specific transgene is a wonderful idea, it is not realistic or standard in the field of in vivo KDs of motor proteins in KD animals. Furthermore, the proper controls for the off-target effects of MOs were conducted, as previously shown by Dr. Jennifer Gutzman and colleagues. We cited this work in the previous version of the manuscript, but clearly did not include enough detail in our Methods, which we now provide. To further address this concern, we also provide quantification of NMIIA/NMIIB KD efficiency and compensation by the other paralog (Figure 9 and Figure S12). We also agree with the reviewer that further studies based on these data should be undertaken. Indeed, multiple studies have reported less than what we report here yet have stated more. We simply state here that NMIIA and NMIIB are required for sarcomere formation in zebrafish. We make no claims this is cell-type specific.

In addition, Figure S3C of β-cardiac myosin and NMIIB staining in heart sections could simply be showing NMIIB in a small gap between two adjacent cardiac myofibrils. One would like to see a linear array of the BCMII/NMIIB along some continuously stained F-actin fibers to be convinced. This could be omitted also.

This data is not showing NMIIB next to βCMII, it is showing NMIIB and βCMII are in the same filament. We provide multiple examples of these structures in hiCMs, mouse tissue, and human tissue. We used the same technique shown by (Beach et al., 2014) to show multiple myosin II species in co-filaments in cells. As this is the first work showing NMII-βCMII co-filaments in both cells and tissue, this data should not be omitted.

10) I agree wholeheartedly with the authors that their work on the NMIIA and IIB indicates new refined models are necessary and that we should not get hung up on the terminology. The authors may want to refine their model shown in Figure 11 based on data regarding α-actinin2 localization and assembly, as well as the other additional experiments and clarifications I have suggested.

We have done so.

Specific Points for improving Figures:1) Figure 1F. I can't really see the transition of the MSFs to myofibrils with sarcomeres in Figure 1F. There are too many other myofibrils crossing it and near it. Also, is this indicated "MSF" really an actin arc? It is practically in the middle of the cell. The wide-field example in Figure 1E is better; we need a confocal example similar to this one.

Thank you for this comment. We have now added several more examples of the assembly of the initial myofibril (Figure 1E-G). Some MSF travel further than others before they gain a myofibril like organization of actin and the density of stress fibers/myofibrils make it difficult to clearly see the transition. Figure 1G is an example of this problem. As such, we are only using Figure 1G to show sarcomeres appear on the dorsal (top) surface of hiCM.

2) Figure 1H. The kymograph in Figure 1H is strange- the retrograde flow of the MSF is convincing, but the purple line through the "sarcomere" seems to be parallel to the myofibrils, which are orthogonal to the MSFs. The authors need to find another example where the myofibrils with sarcomeres are parallel to the MSFs and to the cell edge – similar to Figure 1E, or to the cell and kymograph in Figure 3E-F.

We agree and have provided a different example with sarcomeres parallel to the edge. Thank you for pointing this out.

3) Figure 2. The images of Arp2/3 staining in the control and CK666 cells in Panel C appear to show the edge of a spreading cell; with the actin fibers well-behind the edge (a lamellipodium, presumably). However, the image of the cell in Panel A is not the edge of a spreading cell since the actin fibers are right up at the edge. Where is the Arp2/3 in cells like those in Panel A?

The cell in Panel A was treated with CK666 for 24 hours. For the reviewer’s benefit, we have cropped this Figure out using the “snipping tool” in Microsoft Windows and stretched the image in FIJI (ImageJ) (Author response image 1). As can be seen in the stretched image, there is actin all the way to the edge where Arp2/3 is localized. We intentionally displayed the actin at a lower level so that the Arp2/3 localization can be seen clearly.

**Author response image 1. respfig1:** Figure 4F stretched.

4) Figure 3. It is hard to compare the F-actin distribution in Panel A (SMIFH2 inhibition) with that in Panel C for the FHOD3 siRNA experiment. The F-actin in Panel A could be shown at higher magnification and brighter to match that of Panel C.

The actin and merge of actin and _α_-actinin 2 in SMIFH2 has been increased in size (current Figure 5A).

5) Figure 4. The NMIIA appears to be enriched right at the edge of the cell, while the NMIIB is more abundant further away from the edge. However, the line scan shown in Figure 4B does not match the images. The image in Panel C where the NMIIA and IIB are co-localized also shows that the IIA is relatively more abundant near the cell edge. The line scan should be replaced with data that better matches the images.

Thank you for this point. All 3 reviewers brought up this issue. The line scan was created from averaging the localization of NMIIA and NMIIB from multiple individual cells (previous Figure 4B, Current Figure 7B). As such, we chose to change the images to reflect the average. The examples shown in the original manuscript (previous Figure 4A) were from two different cells and not directly comparable. We have replaced the original Figure 4A with localizations of NMIIA and NMIIB in the same hiCM (current Figure 7A).

6) Figure 4E. The NMIIA vs NMIIB knockdown is cool! The MYH10 knockdown cell appears to form abundant F-actin donut structures that resemble podosomes. I can see them in Figure 6F also. Are they podosomes? If so, they would contain vinculin. What might this mean? Competition between different F-actin structures? Effects of NMIIB in cell adhesion?

We think these structures are unlikely to be podosomes (at least not the ones attached to the substrate). They are distributed throughout the cell and are not localized at the ventral surface (Author response image 2). However, there is a likely possibility vinculin and other podosome proteins will localize to them. We do detect podosome-like structures on the ventral surface of control hiCM particularly early after plating. We are currently working on addressing the podosomes more thoroughly in an ongoing project in the lab focusing on how substrate adhesions link to myofibrils.

**Author response image 2. respfig2:** 3D projection of actin- NMIIB KD hiCM.

7) Figure 6. Can a high mag field of the NMIIB/β-cardiac myosin at the edge of the cell be shown to illustrate better the authors' point that the filaments are smaller and not lined up? The individual filaments in Panel C show the co-assembly, but not the overall pattern of the β-cardiac myosin compared to the NMIIB. Also, does IIA co-assemble with β-cardiac myosin?

Thank you for this comment. This was an obvious oversight on our part. We also did not provide such a view for the co-filaments of NMIIA and NMIIB (current Figure 7D- same cell as previous Figure 4C). We mentioned that almost all of the NMII filaments contained both paralogs but, obviously, failed to show a field of them. We have also now provided a high mag field of endogenous β-cardiac myosin and NMIIA at the edge (Figure 9—figure supplement 2). As shown, it is relatively easy to “find” co-filaments in regions less populated by filaments (Figure 9—figure supplement 2B, boxes 1 and 2). However, in more dense regions it is more difficult to discern them with the spatial resolution afforded by SIM (Figure 9—figure supplement 2B, box 3). We feel this is important information for anyone who will work on the potential roles of co-filaments in the future and thank the reviewer again for this comment.

8) Figure 6F. The movie of the β-cardiac myosin in the control cells (Video 6) is nice and clearly shows the thick filament assembly at the very edge of the cell, where the NMIIA is located. Since a major point of this figure is that NMIIB helps coordinate the β-cardiac myosin assembly, I would also like to see a live cell movie of the β-cardiac myosin in the MYH10 knockdown cells.

We do not see how adding a live cell movie of the β cardiac myosin in the MYH10 knockdown cells would add any new insight, as we already show these cells fail to form Abands comparable to control hiCMs (current Figure 10I).

9) Figure 6F-G. The result that the MYH10 KD reduces F-actin assembly but allows β-cardiac myosin assembly to form thick filaments is interesting. However, a caveat is that it might be difficult to image very thin F-actin fibrils due to the very bright podosomes. Staining for α-actinin2 would address whether there are residual myofibril-like scaffolding structures; as might be suggested by the end-to-end lining up of the β-cardiac myosin structures.

Indeed, β cardiac myosin II filaments are associated with remaining actin structures in the NMIIB KD hiCMs. We failed to mention this in the previous version of the manuscript. We now provide a high magnification view of a βCMII filament on residual actin filaments in the NMIIB KD hiCMs (current Figure 10H). We thank the reviewer for catching this oversight. This association was the reason we performed the subsequent Latrunculin experiments to more thoroughly remove the actin cytoskeleton.

10) Figure 6H. The β cardiac myosin appears to form star like structures in the Latrunculin treated cells. What might these be? If the cells are stained for F-actin, are there residual F-actin fibrils connecting the stars? Where is the α-actinin2? This reminds me of the polygonal arrays of F-actin and α-actinin2 in spreading cardiac myocytes reported by Holtzer's group. Lin et al., 1989. Does Latrunculin lead to changes in cell spreading? Or size?

The star-like structures the reviewer mentions, which we now present a blow-up of (current Figure 10I), are similar in structure to the aggregates of cytoskeletal proteins shown in non-muscle cells treated with Latrunculin or Cytochalasin. These aggregates contain small actin filaments that somehow did not get recycled by endogenous mechanisms during treatment

(precise mechanism unknown). This is probably what the βCMII filaments are binding to. Indeed, in non-muscle cells, NMII and other cytoskeletal components, such as some formins localize to these such puncta (Luo et al., 2016; Luo et al., 2013), while _α_-actinin does not (Luo et al., 2013).

We did not notice any major changes to cell size. As stated in the text, we treated these cells after they had already spread for 18 hours, not during cell spreading.

11) Figure 7. Can the distinction between myosin stack formation by expansion versus concatenation mechanisms be explained more clearly? I don't get it. In the expansion image sequence in Figure 7D, it appears that a thick filament is splitting into two separate adjacent thick filaments- this could later lead to a branched myofibril (a well-known process in myofibril assembly). I don't see how this image shows that the thick filaments are adding on to one another make a thicker myofibril.

Thank you again for asking for this clarification. We think about the concepts of expansion and concatenation quite often and, as a result, did not notice that we failed to fully describe them in manuscript. We have changed the text to differentiate between these two models with the following text in subsection “NMII and FHOD3 are required for organized A-band formation”:

“In non-muscle U2OS cells, A-band-like stacks of NMIIA filaments are often formed through a process called “Expansion” (Fenix et al., 2016). During Expansion, NMIIA filaments that are close to each other (e.g., in a tight bundle) move away from each other in space but remain part of the same ensemble, where they align into a stack like in the A-band. In addition to Expansion, NMIIA filaments also, but more rarely, “Concatenated” (Fenix et al., 2016).

Concatenation was where spatially separated NMIIA filaments moved towards one another to create a stack.”

Indeed, the reviewer points out that previous Figure 7D (current figure 11D) does not show myosin II Concatenation or the creation of a thicker myofibril. This example now Figure 11D, shows myosin II Expansion, or the movement of filaments away from each other. In hiCMs our data suggests Expansion actually results in a smaller and less organized myosin II ensemble, while Concatenation results in a thicker myofibril which is shown in (previous 7C, current Figure 11C). The arrow heads in current Figure 11C denote βCMII ensembles which concatenate with separate βCMII to create a larger βCMII filament stack.

The phenomenon of myofibril splitting is interesting. However, we rarely see myofibril splitting in live hiCMs. Many phenomena occur at later time points than we are studying and splitting is likely to be among them.

Reviewer #2:This study by Fenix et al., builds on decades-old models of muscle sarcomere formation that postulated that non-muscle myosin assemblies served as templates for the formation of muscle myosin II-containing sarcomeres. This group was optimally positioned to tackle this issue that included important questions, for example the existence of actomyosin filaments containing non-muscle and muscle myosin II elements, the differential role of the major non-muscle myosin II isoforms, etc. This study elegantly answers several of these questions. Specifically, the authors demonstrate that: (1) formins are required for sarcomere formation; not so much the Arp2/3 complex; (2) the non-muscle myosin II isoforms play differential roles as sarcomeric precursors; (3) that this is likely the case in vivo.Overall, I think this is a terrific study, one that definitively merits publication in eLife. Having stated this, I have some issues with several data sets, which in my opinion require some additions, modifications, and clarifications. However, I am convinced the authors can perform these experiments in a reasonable amount of time.Specific point:1) In Figure 1G-H-I, it seems clear that MSF and actin arcs are not the same entities in terms of dynamics, e.g. rates of retrograde flow. However, a major distinctive feature of stress fibers is that they connect focal adhesions, thus they are confined to the lower planes of the cell: coverslip interface. I think it would be a good idea to compare the retrograde flow of these structures using TIRF, not spinning disk as they do.

Thank you for this comment. We agree that TIRF’s signal-to-noise would be ideal for live imaging of the MSF to myofibril/sarcomere transition. Unfortunately, the transition occurs on the dorsal surface of the cells 1-2 microns away from the substrate. Much like actin arcs in many cells types (e.g., U2OS), the MSF in hiCM travel in a retrograde manner along the dorsal surface of the cell. In our original submission, we failed to articulate the 3D position of the MSF and sarcomeres and showed only showed maximum projections. At the time, it did not seem unreasonable to do this as the top of the cell is relatively flat. However, as the reviewer insightfully points out, not defining the position explicitly will lead to several questions on the reader’s part. As such, we have chosen to revise Figure 1 to include 3D renderings of actin filaments in hiCM. We have also included a time montage in Figure 1G showing the actin on the ventral surface of the cell during the MSF to sarcomere transition. This data is particular important in light of new data published (Chopra et al., 2018) claiming that the first sarcomeres in hiCM form on the ventral surface of the cell at much later time points than we have found.

2) In Figure 2, I am fine with the effect of CK-666 in the localization of the Arp2/3 complex (as measured by anti-p34 staining, I assume; it'd be useful to have this stated in the figure legend). However, I am surprised by the lack of effect on the shape of the leading edge, which is very similar. I would like to see kymographs of the leading edge in control and CK-666-treated cells (similar to Figure 3F, but also showing the extension of the leading edge).

The reviewer is indeed pointing out an interesting phenomenon that has also piqued our interest. The lamellipodium is typically smaller in hiCMs than in some non-muscle cells lines. As such, reducing the hiCM’s lamellipodium is often a minor change. However, this is not always the case. We now also present live-cell data showing P16B-mEGFP and Lifeact-mApple before and after acute treatment with CK666. In addition, as per the reviewer’s suggestion, we have also now added dynamic data from cells treated with CK666. Of note, leading edge dynamics are reduced. In addition, retrograde flow of the MSF is not significantly different. After the revision, this data can be found in Figure 4.

3) My main issue with this study pertains to the interpretation of Figure 4E. While I agree some sarcomeres can be seen in Myh9-depleted hiCM cells, the difference between scramble and Myh9 siRNA is too striking to be described as a mere "disorganization" issue. As such, the metric used in Figure 4G is very misleading. I remain convinced Myh9 plays a role in this process, and this description does not adequately address this. To confirm these data, I would like to see data using p-nitroblebbistatin (so they can do dynamics) and a better way to represent the disorganization caused by Myh9 depletion.

We are indebted to the reviewer for bringing this up and pointing out the weakness of the measurements in what are now Figure 8A and 8B. This has actually resulted in a major reanalysis and change in our interpretation. We did not intend to imply that *MYH9* siRNA had a small effect on sarcomere assembly- just that the *MYH9* siRNA-treated hiCM had sarcomeres. There is clearly a large effect. We have now stated that there were less sarcomeres and have added new quantifications comparing siRNA scrambled and *MYH9* siRNA-treated hiCM. While the distances between Z-lines were not dramatically changed (Figure 8D), there was a significant difference in the length of Z-lines (Figure 8E). We feel these quantifications clarify the phenotype and will make it easier for future studies to directly compare their results to ours.

This line of quantification led us to use MO mediated KD of NMIIA in Zebrafish (Figure 9). MO mediated KD of NMIIA in Zebrafish also resulted in a decrease in the number of sarcomeres (Figure 9C), and sarcomere persistence (Figure 9E). Importantly, the loss of sarcomeres in the NMIIA KD hearts were equivalent. As such, our data suggests that both NMIIA and NMIIB play major roles in sarcomere assembly. Again, we thank the reviewer for this point.

We have also now presented data using blebbistatin. Blebbistatin treatment indeed stopped the assembly of MSFs and sarcomeres (Figure S10).

4) Figure 4A, the localization of NMII-A and NMII-B is described as equivalent, yet NMII-B remains "stubbornly" behind NMII-A, which is okay. Although I agree that the existence of co-polymeric forms of the two isoforms reflect the existence of a transitional area, the image in Figure 4A and the quantification shown in Figure 4B are dramatically different and may lead to confusion.

Thank you for suggesting this clarification. We agree that our language and measurements could lead to confusion. We were trying to convey that neither NMIIA or NMIIB extended as far from the edge as we would expect in non-muscle cells such U2OS. Again, our focus on NMII-B has changed due to the new measurements described in point 3 above. As such, we now report the offset at the edge and then simply state the NMII localization does not persist into regions with sarcomeres. We have also replaced the images of two separate cells with a single cell showing endogenous NMIIA and NMIIB localizations (now Figure 7A). This better reflects the localization and measurements averaged over multiple cells (previously Figure 4B and now Figure 7B). While we did place lone NMIIA filaments at the edge of hiCM in our cartoon model, we clearly failed to articulate the difference in localization. To clarify this issue, we now say in the text on Page 6:

“The vast majority of the NMIIA and NMIIB filaments overlapped, except at the very leading edge where NMIIA is localized slightly ahead of NMIIB in hiCMs (Figure 7A and 7B)”

5) I do love the experiments depicted in Figure 5. However, I am confused as to their meaning. In the original paper from the Adelstein group (Tullio et al., 1997), a major issue was the size of the cardiomyocytes, which seems unaffected here. A similar result was seen with the conditional II-B/-C ablation in the Ma et al. paper from 2009. Is the size of the cells significantly larger in these experiments?

We are hesitant to directly compare our experiments with in vivo experiments with mice. With that said, we do not see larger cells in our experimental model. This is likely due to these cells sitting on a 2D surface.

6) How is betaCMII organized before sarcomeres form? Does it form mini-filaments? In the same direction, if betaCMII is depleted, does NMII-B distribute evenly throughout the cytoplasm? Or is it still confined to the leading edge?

This is a very interesting question. βCMII does indeed form short filaments (similar in length to NMII filaments), specifically at the edge of hiCMs. As shown in Figure10E, the length of βCMII filaments are shorter at the edge and increase in length further away from the edge. We also now present another view of β cardiac myosin II with NMIIA where short filaments of β cardiac myosin II are shown more clearly (Figure 9—figure supplement 2).

The second question the reviewer poses has led to some strange results. First, as further discussed at the end of this response, we do not find that reducing β cardiac myosin II has much effect on sarcomere assembly. In addition, we find that NMIIB stays at the edge even when cells are spread in the presence of blebbistatin (Author response image 3). There is noticeably less NMIIB at the edge, but it does not re-localize throughout the cell as we would expect. This result warrants further investigation and is part of an on-going project in the lab investigating what factors are responsible for the localization of NMIIB (and NMIIA).

**Author response image 3. respfig3:** hiCM spread in presence of 100µM Blebbistatin; 24 hours after plating.

Reviewer #3:This paper addresses some very interesting questions regarding sarcomere formation using a nice model cell system. While some of the data looks quite good, a considerable amount has significant issues with quantitation, lack of depth, and/or interpretation. Below are my specific comments.I am surprised that there is no change in hiCMs after CK666 treatment (Figure 2) other than a reduction in Arp2/3 localization at the cell edge. In most cells Arp2/3 is consuming the bulk of the monomer at steady state. Is there a change in total actin content? Retrograde flow rate? Ability to spread?

We did not detect a change in total actin content. We have now included experiments measuring the rate of retrograde flow of actin in the presence of CK666, and hiCMs show no change in the rate of retrograde flow in response to CK666 (Figure 4D). As can be seen in Figure 4A, hiCMs are spread similar to control hiCMs in the presence of CK666.

Do the formin-dependent MSFs get more robust because of the increase in monomer supply when Arp2/3 is inhibited? Of note, their CK666 treatment displaced only ~50% of the Arp2/3 signal from the edge. The remaining Arp2/3 signal/activity could well be enough to drive a significant fraction of normal branched nucleation. I wonder if CK666 works in their cells as expected, and/or if the CK666 in the media after 24 hours of imaging is still active?

To quantify sarcomere formation in presence of CK666 (now Figure 4A and 4B), hiCMs were not imaged for 24 hours as suggested by the reviewer. These cells were allowed to spread for 24 hours in the presence of CK666 in the dark as noted in the text and methods, then fixed and stained before imaging. To address the reviewer’s comment and provide a more acute experiment testing the effect of CK666, we imaged live hiCMs expressing a subunit of the Arp2/3 complex, P16B-mEGFP. As can be seen in Figure 4H, treatment of hiCMs with CK666 results in a rapid loss of P16B-mEGFP from the leading edge of hiCMs. In addition, the concentration of CK666 used is well above saturating concentration. Furthermore, inhibition of the Arp2/3 complex via CK666 did not affect retrograde flow rates of MSFs (Figure 4D). Finally, it is important to note that we are not claiming that Arp2/3 plays no role in sarcomere assembly. However, whatever role it might play does not require it to be at the edge.

Does robust KD of any Arp component give the same result? Basically, this aspect of their work needs additional effort before the authors can categorically exclude a contribution from Arp23/ to MSF formation.

We agree with the reviewer that further evidence is required to show that we are inhibiting the Arp2/3 complex. As such, we now provide both acute (Figures 4D and 4H) and long term (24 hours) data showing CK666 treatment results in de-localization of the Arp2/3 complex in fixed and live cells (Figures 4F-I). The reviewer is correct in that we are not categorically ruling out a role for the Arp2/3 complex in sarcomere biology. As such, we have also softened the language we use in our interpretation. The conclusion to subsection “Formins, but not the Arp2/3 complex, are required for MSF-based sarcomere formation” now reads:

“Taken together, our data suggests that the Arp2/3 complex does not need to be localized at the leading edge for sarcomeres to be assembled.”

Given that Arp2/3 plays multiple roles in the cell- including major roles in membrane trafficking, we are hesitant to knock down any individual component. Knock down in our system takes multiple days and we have no way of predicting potential ramifications of long-term removal of Arp2/3 activity.

I recommend in subsection "Formins but not the Arp2/3 complex are required for MSF-based sarcomere formation" that the authors move the fourth paragraph in front of the third paragraph. When I read the second paragraph about how SMIFH2 blocks sarcomere formation, I immediately wondered about the MSFs, given that they are supposed to give rise to the sarcomeres. But before I got the answer, I had to read about FHOD3. This seems out of order.

Thank you for pointing this out. We now see that order of this section was awkward. As such, we now present the SMIFH2 data in Figure 5 and then the FHOD3 data in Figure 6.

In addition, the authors should determine FHOD3's localization relative to MSFs and sarcomeres to see if it supports their idea (and to provide some additional information on FHOD3's role in sarcomere formation since we already know from previous work that sarcomeres disappear when it is knocked down).

Thank you for suggesting this addition. We have now provided the localization of FHOD3 in Figure 6E.

The SMIFH2 data in Figure 3 needs a time lapse movie to show how MSFs and sarcomeres disappear, and it needs a washout experiment (preferably with movie) showing the SMIFH2 treatment is readily reversible in these cells.

We agree with that this data would add further evidence that formin inhibition is preventing sarcomere assembly. We now provided live examples of cells spreading out in SMIFH2 and they do not assemble sarcomeres (Figure S3), and a washout experiment where cells subsequently assemble sarcomeres in (Figure 5—figure supplement 2).

I think they need at least more incisive experiment to support the idea that MSFs give rise to sarcomeres. One good way to accomplish this would be to use photoactivation or photoconversion to put a fiducial mark in MSFs and then show that the mark persists for some period of time in nascent sarcomeres. This should be readily doable.

Thank you for this idea. As reviewer 1 suggested, we have now included experiments and analysis of α-actinin to the manuscript in Figure 2. Along these lines, we now present data showing that some puncta of converted α-actinin-2-mEOS2 in MSF do get incorporate into Zlines.

To address the reviewer’s request, we originally tried to use α cardiac actin for the photoactivation experiments as we thought this might be the most direct way of showing the transition. Unfortunately, this line of investigation did not work. We cloned α cardiac actin from a cDNA library we created from hiCMs and fused it to EOS. This construct did express in hiCMs but did not incorporate well into either MSFs or sarcomeres. We also expressed fluorescently tagged fusions of another actin paralog, βactin, that we know incorporate into actin arcs in U2OS cells. β actin showed a similar localization. It is not wholly surprising that actin monomers with a fluorescent protein hanging off them do not get incorporated into formin-generated actin filaments. Others have noted that fluorescent tagged actin monomers do not seem get past formins well. Nonetheless, we were still disappointed.

The authors use the terminology "MSFs acquire sarcomeres" in various places in the text. To me this does not describe what I think the authors are trying to convey, which is that MSFs transition into sarcomeres or MSFs template sarcomere assembly or something like that. I encourage the authors to think more carefully about their wording.

We agree that transition is a better word for what we are trying to convey. As such, we have modified the manuscript. It should be clear that we believe that MSF are a template for sarcomeres. Thank you for this suggestion.

I am not sure the idea that MIIB is required for sarcomere formation but not for sarcomere maintenance makes sense. I assume that, like everything else, these structures are constantly turning over, so if one blocks sarcomere formation, sarcomeres must disappear over time because of turnover (or the system is more complicated than they think).

Every biological system is more complicated than we think. Such is the burden of those of us who have chosen not to be reductionists. All we can do here is report the data and interpret the best we can. First, NMII filaments are absent from sarcomeres. In addition, after 10 days of KD of NMII (either NMIIA or NMIIB), myocytes have similar sarcomere organization as in siRNA treated controls (Figure 8—figure supplement 6). Based on published data of component turnover in mature sarcomeres, this is enough time for the sarcomeres to turn over several times. However, when asked to assemble sarcomeres without pre-existing sarcomeres in our spreading assay, the NMII KD hiCMs fail to do so. Over all, we think this data supports the proposal that there could be separate mechanisms governing sarcomere assembly and sarcomere maintenance. Clearly, quite a bit of future work will be needed to fully address the molecular mechanisms underlying the molecular differences between assembly and maintenance.

*I have several significant issues with the data in Figure 4 as regards MIIA KD. First, the authors need to show the level of KD. Without that, one can have confidence in the conclusions. Second, the statement "KD of MIIA did not result in inhibition of MSF or* de novo *sarcomere assembly" appears to be at the very least a major over-simplification. First, it looks to me like the MSFs are largely missing in the KD cell shown in the middle Panel in 4E. Moreover, no quantitation is provided to support the claim that the "KD of MIIA does not result in an inhibition of MSF formation". Second, the effect on sarcomeres looks to be much greater that just disorganization as they call it. Indeed, the scoring for sarcomere assembly in Panel G (percent cells with sarcomeres) is not adequate. Something much more quantitative is required (e.g. total sarcomeres per cell, or per unit). It is my guess that proper quantitation of this data will alter their conclusions significantly.*

Thank you for bringing this up and pointing out the weakness of the measurements in what are now Figure 8A and 8B. This has resulted in a major re-analysis and change in the interpretation of the data. We did not intend to imply that *MYH9* siRNA had a small effect on sarcomere assembly- just that the *MYH9* siRNA treated hiCM had sarcomeres. There is clearly a large effect. We have now stated that there were less sarcomeres and have added new quantifications comparing siRNA scrambled and *MYH9* siRNA treated hiCM. While the distances between Z lines were not dramatically changed (Figure 8D), there was a significant difference in the length of Z-lines (Figure 8E). We feel these quantifications clarify the phenotype and will make it easier for future studies to directly compare their results to ours.

This line of quantification in hiCM also led us to use MO mediated KD of NMIIA in Zebrafish (Figure 9). MO mediated KD of NMIIA in Zebrafish also resulted in a decrease in the number of sarcomeres (Figure 9C), and sarcomere persistence (Figure 9E). Importantly, the loss of sarcomeres in the NMIIA KD hearts were equivalent to that of NMIIB KD. Combined, the hiCM quantifications and the Zebrafish results have led to a major change in our interpretation.

Clearly our data suggests that both NMIIA and NMIIB play major roles in sarcomere assembly. Again, we thank the reviewer for this point.

What I see in Figure 4A and B is very different from what the authors conclude. The authors say that, unlike mesenchymal cells where MIIA is peripheral and MIIB is more central, these two isoforms are in the same place within MSFs of spreading hiCMs. To me, Figure 4A shows very clearly that MIIA is enriched closer to the leading edge than MIIB, exactly as seen in mesenchymal cells. I would need to see simultaneous two-color imaging to be convinced that the distribution MIIA and MIIB in hiCMs is different from that in mesenchymal cells. Similarly, the left most SIM mage in Figure 4C clearly shows (especially on the right side of the cell) mostly red (MIIA) nearest the cell edge, then white moving in (presumably overlap between MIIA and MIIB), and finally mostly green (MIIB) furthest in. My guess is that better analyses and quantitation of all the data in Figure 4A-4C will alter their conclusions significantly.

We apologize for the confusion. Previously, we presented NMIIA and NMIB in different cells (previous Figure 4A). As such, it was not possible to directly compare the relative localizations of NMIIA and NMIIB. This is why presented quantification of their localizations NMIIB averaged over several cells (previous Figure 4B; current Figure 7B). We did not intend to imply that NMIIA was not slightly in front. We even included this detail in our cartoon model. However, we clearly chose to show cells that were not completely representative of the quantification. In particular, the NMIIB localization that was previously shown in Figure 4A was slightly further from the edge than average. To eliminate confusion on this point, we now present NMIIA and NMIIB localized in the same cell (current Figure 7A). This required us to label one of the antibodies—to NMIIA— directly with a fluorophore.

The main point we were trying to make was that NMIIA/NMIIB was not about which paralog showed up first but that neither extended as far as one would see in mesenchymal cells. We have removed this comparison.

Finally, I suggest the authors determine the ratio of MIIA/MIIB within individual filaments going from the cell edge inward (as in Beach et al., 2014).

This is a confusing request. Beach et al., 2014 did not measure the ratio of MIIA/MIIB in individual filaments, nor did Shutova et al., 2014, which also presented a characterization of NMII co-filaments. (Beach et al., 2017) did not measure MIIA/MIIB ratios either. Beach et al., 2014 did measure NMIIA/NMIIB protein levels following over-expression of the other paralog (using Westerns).

Beach et al., 2014 did report that there were fewer co-filaments of NMIIA/NMIIB further away from the edge. While we do not detect this, it could be because the localization of NMIIA/NMIIB filaments in hiCM do not exist as far from the edge as in U2OS cells.

I wonder if the MIIB KD effect is the result of not having MIIB or of removing much of the type II myosin in these cells. For example, if MIIB represents 90% of total MII in these cells, then it is possible that the KD effect stems from removing the vast majority of type II myosin, and not specifically MIIB. To address this, the authors need to determine the ratio of MIIA to MIIB (and MIIC?) in these cells. Along these lines, can overexpression of MIIA rescue the MIIB KD phenotype?

The reviewer brings up an interesting point here, which is the focus of an ongoing project in the Burnette lab. We also wondered if the effect of NMIIB KD was simply the result of it being the more highly expressed paralog? Based on our preliminary data, we feel that the mechanisms are more complicated. We first performed RNAseq on hiCMs and the mRNA ratio was 1.31:1 for *MYH9:MYH10*. A difference in turnover kinetics could still result in higher NMIIB protein levels. Therefore, we overexpressed NMIIA after NMIIB KD in three independent experiments, which failed to rescue the NMIIB KD phenotype (response Figure 4). These data have led us to explore what domain of NMIIB is important for sarcomere assembly. We reasoned that the motor domain could be a factor as NMIIA and NMIIB have different duty ratios. To test this, we expressed a chimera with the motor domain of NMIIB fused to the rod domain of NMIIA after NMIIB knockdown. We found that this construct resulted in 80.9 +/- 10.5% of hiCMs containing sarcomeres (response Figure 4B); (N=3 independent experiments; Exp 1: 7/10 cells, Exp 2: 10/11 cells, Exp 3: 9/11 cells). We confirmed that endogenous NMIIB was knocked down in each cell that was analyzed by IF using an NMIIB antibody raised to its C-terminus. While these experimental results are intriguing and encouraging, they are clearly only the start of larger project in which we will need to further characterize these conditions.

**Author response image 4. respfig4:** Expression of NMIIA and NMIIA/NMIIB(motor) chimera in NMIIB KD hiCM.

I was not clear to me that their quantitation of control and MIIB KD zebrafish hearts is representative because the high mag view of the KD tissue stained with α-actinin in Figure 5B looks pretty much like the high mag view of the control tissue in Figure 5A. In other words, they don't appear to differ by anything like ~3-fold, as presented in the quantitation in Figure 5D. Regarding Figure 5D, I assume that this data was taken from the boxed areas in the images stained for actin in Figure 5C. If I am correct, then this is not appropriate for two reasons. First, the data needs to be collected in a nonbiased way, i.e. not by picking areas but by scoring the entire sample area. Second, because sarcomeres are much clearer in the α-actinin-stained images than in the actin-stained images, the measurements need to be made using α-actinin staining. My guess is that proper scoring of this data will alter the conclusions to some extent.

We thank the reviewer for these points. We agree the way in which we presented the data in what previously was Figure 5 in the original submission was quite confusing. In the original manuscript, Figure 5A and 5B were hiCMs before re-plating, and Figure 5C and D was from Zebrafish. Though this was noted in the text, figure and figure legend, we agree it was confusing. All data from Zebrafish is now in one main figure (Figure 9) and one supplementary figure (Figure S12). Similar to the NMIIA KD in hiCMs, we have more thoroughly quantified sarcomere assembly from the Zebrafish experiments (Figure 9). As noted above, we also include a NMIIA KD via MO in Zebrafish. In addition, we measure NMIIA and NMIIB protein levels in the KD condition of the other paralog, number of sarcomeres, lengths of Z-lines, and sarcomere persistence (Figure 9C-9E and Figure S12). Importantly, measurements in the original manuscript (previously Figure 5) and the current manuscript (new Figure 9) were not made from the yellow boxed area, which was shown simply to indicate where the high mag image was taken from. As the reviewer notes, it is important to quantify the whole image, which is exactly how these images were quantified. We thank the reviewer for these points, as the Zebrafish data is now much clearer and more thoroughly characterized.

Regarding Figure 6, the authors say, "We noted that near the leading edge of the cell, BCMII filaments were smaller and not organized into stacks resembling A-bands (Figure 3C)". I do not see data on filament size in Figure 3C. Regarding the conclusion that mixed MIIB/BCMII filaments populate a specific region behind the leading edge, the authors need to provide high mag insets and quantitation of double-stained, SIM imaged cells to support this conclusion.

The data was in Figure 6 and Figure S3 and was appropriately cited in the previous version of this manuscript. We presented multiple quantifications to support this statement. First, we presented the localizations of NMIIB and ββCMII in hiCMs and quantified their relative localizations over multiple cells and multiple experiments (previously Figure 6A and 6B). We also measured the distributions of the filament lengths of both NMII and βCMII through the cell (previous Figure 6D and current Figure 10D). We also measure βCMII filament lengths with respect to their position in the cell (i.e., measured from the leading edge of the hiCMs) (previously Figure 6E and current Figure 10E). These measurements suggest that the smaller βCMII filaments are found closer to the leading edge of the cell and largely restricted from the longer βCMII filaments of the A-bands (current Figure 10E). In addition, we do provide hi-mag images of the shorter βCMII filaments (previously Figure 6C and current Figure 10C; previously Figure S3 and current Figure 9- figure supplement 1 and Figure 9- figure supplement 2). These quantifications were all performed from SIM images. All figure legends now explicitly state which imaging modality was used.

How well was MIIB knocked down in Figure 6F and 6G?

On average, we achieved a 70% KD of NMIIB. The quantification of the knock down percentage of both NMIIA and NMIIB is now presented in current Figure 8—figure supplement 2.

I found the conclusion from the latrunculin data- "MSFs and sarcomeric actin filaments are serving as "tracks" with which BCMII filaments are loading onto as they from larger BCMII filament stacks of the A-band"- to be a huge over statement of the data. That things don't work when you blow up all the actin does not even remotely prove this elaborate mechanism. This would require at a minimum very detailed dynamic imaging of the normal maturation process.

We have softened the language simply to, “These results argue that actin filaments are required for the organization of the βCMII filaments of the A-band”.”

Regarding Figure 7, the authors state that they do not "believe MII/BCMII co-filaments are not the major mechanism through which MIIB is facilitating BCMII filament stack assembly". Later, however, they state that "MIIB may be serving as a template to seed the formation of BCMII filaments". I don't see anywhere in the text that the authors clearly articulate what they think the role of MII is in this process. If it is not contributing as a co-filament/template, then what is it doing? If it is contributing as a co-filament/template, then how is their thinking different from the Sanger model? Is there a reason the authors cannot image dynamically MII and BCMII at the same time? It seems to me that the process is slow enough that doing multi-channel imaging would work. Doing this would greatly strengthen this central part of the paper. For example, it would show whether BCMII filaments always appear on MIIB filaments, or if they appear independently? Also, if the authors are looking to combine two previous models, then maintaining those names is more appropriate (I suggest the Sanger-Stitch Model).

We thank the reviewer for catching this statement, as this was a relic of a previous version of the manuscript written before we performed siRNA-mediated knockdown of NMIIA and FHOD3 and localized βCMII filaments (current Figure 9—figure supplement 4). Indeed, we do believe NMII-βCMII cofilaments is a potential mechanism that could lead to A-band assembly and as such this speculation has been removed from the text.

In the end I did not get much sense how MSFs transition/mature into sarcomeres. Some additional insight into this would strengthen the paper considerably. For example, where is capping protein in this process? At some point, I would expect CapZ to incorporate to build a functional sarcomere. Can the authors shed some light on this part of the process with some staining and/or dynamic imaging of capping protein?

No. While we are also fascinated with how MSFs transition/mature into sarcomeres, we strongly feel that pursuing the roles of other proteins in the process is beyond the scope of this manuscript. Future work from our lab and others will investigate interesting potential players such as CapZ. Indeed, even in more rigorously studied systems, such as crawling cells, if/how/when exactly different stress fiber populations transition to one another is not completely understood.

Please use pagination. Please drop the italics on "after" in the third paragraph of the Discussion section. The paper contains a lot of missing words, repeated words, etc. The Introduction could be made significantly more readable with some effort.

We have dropped the italics on “after” in the Discussion section. In addition, a complete rework of the introduction and discussion has been made.

Finally, after the submission of this manuscript, a paper appeared in Dev. Cell from Chris Chen's group addressing this same topic. Notably, they came to very different conclusions regarding a role for NMII. While I think disagreement is very often good for science, I would expect any future versions of this manuscript to contain extensive discussions about these differences (and similarities) and possible explanations for the discrepancies.

We agree with the reviewer that we should address the paper from Chris Chen’s group in our manuscript. As such, we have provided the following in the Discussion section of the current manuscript including two examples of what we believe they are imaging when they claim that sarcomeres assemble from adhesions. Please see below for additional comments on this manuscript we have provided for the Editor and reviewers’ consideration.

Text from the Discussion section.

“Our data shows the transition of a MSF to a sarcomere-containing myofibril occurs on the dorsal (top) surface of the cell. However, a recent study also imaging iPSC derived cardiac myocytes claims that sarcomeres are formed on the ventral (bottom) surface of the cell near extracellular matrix adhesions (Chopra et al., 2018). This group also report that the first sarcomeres forms between 24-48 hours after plating, well after we detect the first sarcomere appearing (Chopra et al., 2018). This led us to question what could be leading to these two seemingly opposite results. Importantly, it appears that this group was imaged the ventral (bottom) surface of their myocytes. Close inspection of their time-lapse movies revealed faint structures corresponding to sarcomeric pattern that show up in the frame right before the appearance of sarcomeres. This supports the notion that they are imaging sarcomeres that are coming into focus, and not assembling “de novo”. To test this idea, we imaged hiCMs with 3D confocal microscopy after they had been plated for 24 hrs (Figure S17). While we also see similar patterns of sarcomeres appearing on the ventral surface as (Chopra et al., 2018), our data revealed these sarcomeres are moving down from the dorsal surface and not assembling on the ventral surface (Figure S17 and Supplemental Movie 3). Of interest, the phenomenon of actin arcs moving down to the ventral surface of non-muscle cells has also been reported previously (Gao et al., 2012; Hotulainen and Lappalainen, 2006). Finally, (Chopra et al., 2018) also claim that neither NMIIA nor NMIIB are required for sarcomere assembly. We also find this strange. While their double NMIIA/NMIIB knockout (KO) cardiomyocyte cell line has α-actinin 2 positive structures, they do not contain continuous labeled Z-lines aligned parallel to each other comparable to the control cell line. The authors did not measure Z-line lengths, spacing, or other criteria needed to define sarcomeres. These discrepancies between our study and theirs, including the role of NMII, need to be harmonized, as it will directly affect our interpretation of future in vivo data concerning sarcomere assembly.”

We feel confident in the arguments that we make in the Discussion section of the main text and feel that they clarify the differences between the study by Chen’s group and the study we present here. In our arguments, we have taken their data at face value. However, there are some concerning aspects of their data and analysis that we do not understand. We have chosen not to comment on them in our manuscript, and instead plan to have a dialogue with the corresponding authors directly. With that said, we believe the points below are germane to the current discussion between ourselves and the reviewers and hopefully will give the reviewers an idea of what we intend to do in the future to further address the findings in (Chopra et al., 2018).

To test whether NMIIA and/or NMIIB were required for sarcomere assembly, (Chopra et al., 2018) used NMIIA, NMIIB, and double NMIIA/IIB knockout (KO) cell lines. These cell lines were presented as, “stable CRIPSR knockout cell lines” in their methods. This is curious for a number of reasons. First, as opposed to the TTNtv, MYH6, and MYH7 KO cell lines which were generated in the iPSC cell stage, (Chopra et al., 2018) created the MYH9 and MYH10 (NMIIA and NMIIB, respectively) KO cell lines by performing CRISPR on differentiated cardiomyocytes. Despite a 3-4 day selection with puromyocin, every micrograph presented of these stable KO cell lines contains cells that are clearly positive for NMIIA and/or NMIIB. In fact, 66.7% of the cells shown have robust NMII localization. This is not in line with their Western Blots, which show robust KD of both NMIIA and NMIIB—though no quantification is provided. Secondly, (Chopra et al., 2018) also measured a surprisingly low number of cells for each experimental condition (i.e., 6 cells over 3 experiments for the NMIIA/NMIIB double knock out cell line). We do not know the reason for this but suspect it is because few cells had lower NMII levels. Finally, it is difficult to even access whether the cells (Chopra et al., 2018) highlight are in fact devoid of protein. The images are displayed in such a way that the background is flat black. With that said, we still think that there could be protein localization in these cells as revealed by stretching the images.

(Chopra et al., 2018) claim that neither NMIIA or NMIIB are required for sarcomere assembly. We do not understand how this they arrived at this conclusion. In the NMIIA/IIB double KO, there are clearly no Z lines comparable to control cells or striated myofibrils containing multiple sarcomeres as seen in control cells (Figure 3A and 3B in Chopra et al., 2018). The authors show a blow-up where there are clear diffraction limited α-actinin 2 puncta but no continuous Z line staining. Indeed, the image used to suggest the NMIIA/IIB double KO cardiomyocytes form sarcomeres appears quite similar to the image in the same figure showing that β MHC is required for sarcomere assembly. The only clear difference is that there is less α-actinin 2 signal between the conditions, which is a major component of the sarcomere content analysis Chopra et al. use to quantify sarcomere assembly. This prompted us to more fully investigate their sarcomere content analysis.

While their methods section is brief, Chopra et al. refer readers to a 2015 paper, (Hinson et al., 2015), where the sarcomere content analysis was developed. It is impossible to fully use this analysis, as multiple variables in the equations used are not defined, and strange terms such as “the energy of sarcomere assembly” are provided. The custom ImageJ and MATLAB codes are also not provided. It does not help that the only paper they cite in their methodology for performing a Fourier transform analysis does not actually include a single reference to Fourier transformation in the entire manuscript. We were already in the process of automating the manual measurements we are presenting in our current manuscript. As such, we have decided to pursue our interest in repeating the “sarcomere content” analysis in the context of this ongoing project. For this we will contact the corresponding authors and request their code so that we can first recapitulate their quantifications and compare it to the physical measurements of Z bodies and Z lines that the software we are developing depends on.

[Editors' note: the author responses to the re-review follow.]

Summary:The revised manuscript is much improved with many new experiments, and substantial rewriting, so that it now presents a coherent story that is well-grounded in the historical context (presented very nicely in the Introduction), answering long-standing questions about the roles of NMIIA and NMIIA in sarcomere formation, using elegant live cell and super-resolution light microscopy approaches in human induced pluripotent stem cell cardiac myocytes (hiCM). The response to reviewers is very thorough and thoughtful. The authors have now used Lifeact-FPs and α-actinin2-FPs to study the transitions of the 'muscle stress fibers' (MSFs) into the myofibrils as the MSFs move dorsally and used the α-actinin2 spots/Z line distances to quantify sarcomere formation very convincingly. They also use photo-activation of Eos-α-actinin2 to show that the α-actinin2 dots in the in the MSFs become Z lines in the sarcomeres. The work shows that the two non-muscle myosin IIs are both important but play somewhat different roles in sarcomere and myofibril assembly, and also that formin FHOD3 but not Arp2/3 actin nucleation is required for sarcomere assembly. The controls are all in place and the conclusions are strong. The authors also show that NMIIA and NMIIB can form heteropolymers with the β-cardiac myosin II (betaCMII) in the cultured hiCMs, a novel finding. The comparisons of the behavior of the arc-like MSFs in the hiCMs with the actin arcs in spreading non-muscle cells such as U20S cells are also interesting and important to the field. Their careful presentation of the outstanding questions and their own new data in a historical context is valuable and balanced.Essential revisions:1) While most concerns have been addressed by the substantial, and high quality new experiments and extensive rewriting, there were still a few concerns with the figure presentation and some of the interpretations. Most importantly, the major messages of the manuscript (MSFs are precursors to sarcomeres, NMII participates in sarcomere formation in a precursor role) became somewhat diluted in the revised version. A few conclusions are also somewhat overstated. Below, we suggest improvements to clarify and strengthen the results and conclusions. These can all be addressed by minor revisions to figures and text.2) It appears from the data shown in the paper that the new sarcomeres are formed by rearrangement of the MSFs, starting at the spreading edge of the cell. The preexisting sarcomeres present before replating appear to be in the center of the replated cells so that they do not play a role in the formation of the new sarcomeres. Since this is not entirely clear, the authors could provide better 3D imaging of a newly replated cell to show where the preexisting sarcomeres are, and their relationship to the newly forming sarcomeres at the edge of the cell.

We believe the organization of Figure 1—figure supplement 1 is leading to confusion. We positioned 1.5 hours after plating and 24 hours after plating next to each other for each sarcomere marker we localized. We have now combined the 1.5-hour time points together in Figure 1—figure supplement 1Aand the 24-hour time points in Figure 1—figure supplement 1B. As shown in Figure 1—figure supplement 1A, there are no discernable sarcomeres at the 1.5-hour time points.

3) Based on the photoconversion experiments using α-actinin-2, if the authors were to do similar experiments photoconverting NMIIA and/or IIB, where would it go? This is important because of their observation that NMII-A and IIB do not incorporate into the sarcomere. It would be nice, if possible, to address this point experimentally.

We thank the reviewers for this suggestion. We did photo-convert myosin IIA-mEOS2 and myosin IIB-mEOS2 at the edge of hiCM. Unfortunately, both paralogs of NMII turned over much faster than α-actinin 2, so it was difficult to follow the marked population for long (Author response image 5). We can say that we did not see photo-converted NMIIs move the middle of the cell where sarcomeres typically assemble. As this result was less than satisfying, we tried an alternative approach. We now present two supplemental movies (Video 2 and Video 3) that show the position of NMIIA with sarcomeres assembling visualized with lifeact and the position of NMIIB with sarcomeres assembling visualized with α-actinin 2. While these are not experiments per se, the movies do demonstrate that region where sarcomeres are appearing are largely lacking NMII.

**Author response image 5. respfig5:** Time montage of Myosin IIB-/mEOS2 in a hiCM.

4) The zebrafish data is rather weak, and furthermore does not address the main message that "MSF are precursors to sarcomeres, NMII participates in sarcomere formation in a precursor role", and thus it can better be omitted.

We have removed the zebrafish data.

Suggestions for revisions of text and figures:1) Figure 6, Panel E. Very nice data, but the images would be clearer if the individual channels were shown in gray scale, with a merge for both the low and high mag crops included. Also, the high mag crops on the right would be better presented horizontally (ie, time along X axis), so that we can follow the assembly over time more easily. The authors should comment on their intriguing observation that the FHOD3 appears to assemble into a pair of widely spaced stripes in the middle of the sarcomere. It is not at the Z line, based on the F-actin staining, nor at the pointed end-- which would be one stripe in the middle of the sarcomere in the unstretched sarcomeres of the cultured cardiomyocytes. Is the FHOD3 staining similar to previous studies of FHOD3 localization? Or is it new? It looks as if it is in the region of A-I junction, near the locations of the myosin heads. This could be commented about.

Thank you for these suggestions. We now present the data as suggested. The reviewers are correct in noticing that FHOD3 does not localize to the Z-line. It localizes in two distinct stripes, one on either side of each Z line. To our knowledge, this not been shown before. This is most likely due to the previous used of diffraction-limited imaging. We have now addressed this issue with the following text added to the Discussion section:

“Future work will be required to elucidate the precise mechanism of how and where FHOD3 potentially nucleates actin filaments in sarcomeres. Canonical sarcomeric actin filaments have their barbed ends embedded within a Z-line (see schematic in Figure 1A). As such, this is where we would have predicted FHOD3 would localize. However, our SIM data shows that FHOD3 does not localize to Z lines (Figure 6F, boxes 4, 5 and 6). Instead, FHOD3 appears to localize on either side of each Z-line. This could indicate that FHOD3 is only transiently associated with the barbed ends of canonical sarcomeric actin filaments. Alternatively, there could barbed ends of actin filaments in this region, which could indicate that the organization and/or dynamics of actin filaments within a sarcomere is more complex.”

2) Figure 8. Very interesting experiments! The effects of MYH9 knockdown versus MYH10 knockdown are very different. While they both are important for sarcomere/myofibril assembly, the MYH9 KD looks more important for lateral myofibril alignment – the KD cells have many branched and wispy looking myofibrils, although they clearly can make sarcomeres; the quantification shows this too. However, the MYH10 KD cells don't make any sarcomeres at all, so NMIIB is required for all (initial?) aspects of sarcomere assembly, while NMIIA is not. Figure 8—figure supplement 3 and Figure 8—figure supplement 4 time-lapse images show these differences very clearly also. The results and other parts of manuscript should be rewritten somewhat to clarify and emphasize these intriguing differences between the NMIIA and NMIIB which are very interesting. Have the authors looked at MSF retrograde flow in either of the knockdowns? This could be interesting (but not required). It was difficult to find any mention of this.

Thank you for these suggestions. We have added the following text to the Discussion section to address these differences more clearly. We hope the reviewers do not mind us taking the term “wispy”. It works well as a descriptor. Thank you again.

“We also show that NMIIA and NMIIB are required for proper sarcomere assembly in our hiCM model system. Interestingly, our data suggest that each motor may be playing a different role(s) in sarcomere assembly. NMIIB KD resulted in no detectable sarcomere assembly (Figure 8, Figure 8—figure supplement 4), suggesting that NMIIB could be required for the initial and possibly subsequent steps of assembly. On the other hand, NMIIA KD-hiCMs were able to assemble sarcomeres, although these were wispy with significantly shorter Z-lines (Figure 8 and Figure 8—figure supplement 3). This NMIIA KDphenotype could be a result of several possibilities including a role of NMIIA in sarcomere maturation, alignment, or stability. Obviously, this is not an exhaustive list of potential mechanisms.”

We have not quantified retrograde flow in these conditions but are part of consortium at Vanderbilt that is currently assembling a lattice light sheet. With this low-dose and high-resolution modality, we plan on performing a detailed 3D study of retrograde flow during sarcomere assembly.

3) Figure 9. The zebrafish data remains unconvincing to me. One can find many areas of sarcomeres in the low mag images in the top panels, both in the myh9bMO and myh10MO hearts. The high mag images in the lower panels are blurry. Moreover, the myofibrils curve in and out of the XY plane in these cells in the heart, and thus some extend in the Z dimension, making it complex to identify all the sarcomeres due to the Z stretch in the confocal. Also, if the myofibrils are even a tiny bit more contracted, then sarcomeres can be very hard to identify from F-actin staining. α-actinin labeling could help. Finally, even if the images and quantification were reliable (which are not convincing to me), the experiment does not address the central point of the study, which is the role of NMIIA and NMIIB in the transition of MSFs to myofibrils containing sarcomeres. There are no clearly distinguishable MSFs in these zebrafish heart cells. We suggest the authors remove this data and save it for a future more extensive study where they use live cell imaging to study the MSF – sarcomere transitions in vivo- probably need lattice light sheet.

We have removed the zebrafish data from the manuscript.

4) Figure 10. Panel H. These odd structures that form in the MYH10 KD cells are intriguing. It appears that rods of betaCMII extend between large donut-like foci of F-actin. Where is the α-actinin2? Is it in the center of the F-actin donut? It was speculated in the previous comments that they might be podosomes, and the response to reviewers provided a clearer explanation of why this is unlikely, based on where they are in the cell and what else they may contain. Some this explanation could be incorporated into the manuscript to make the figure clearer to the reader. We are not asking the authors to do more experiments, but at the least they should describe these images more precisely and then speculate about what the structures might be. Also, the betaCMII rods extending between the F-actin donuts could be individual A bands; are they included in the quantification shown in J? Or are A bands only counted if they are in a linear sequence along and F-actin bundle? Panel H high mag panels need a scale bar.

Thank you for these comments. In the methods section, we have now more clearly defined how we quantified how we quantified the betaCMII data. The following text has been added to the Materials and methods section:

“β cardiac myosin II (βCMII) filament and stack quantification

A similar methodology was used to quantify βCMII A-band filament stacks using endogenous βCMII staining and SIM instead of the actin cytoskeleton. A βCMII filament was quantified as the minimum SIM resolvable βCMII unit which had a bipolar organization (a filament with motor domains on each side), as represented in Figure 10A. hiCMs were quantified as containing βCMII A-band filament stacks if they contained even one βCMII filament stack in the cell. A βCMII filament stack was defined as being thicker than the minimum SIM resolvable βCMII filament (Figure 11A), indicating multiple SIM resolvable βCMII filaments. Indeed, by this metric, βCMII filament stacks have more resolvable “motor-domains” than βCMII filaments.”

5) Figure 11. Panel I. The image from the Latrunculin-treated cell is confusing. Where are the cells in this image? What is the large fluorescent half-moon at the bottom of the image? Is this an accumulation of β-CMII around the nucleus or center of cell? Are there A bands in this densely-stained area – it looks like there might be. A counter-stain with a nucleus marker or cytoplasmic marker could be helpful. Another question is what type of betaCMII structure is in the little box? Are these 'stars' related to the donut like F-actin structures in panel H? i.e., would F-actin be in the hole in the middle? Was Panel J quantified from I? If so, then wouldn't it make sense to show both the betaCMII and the F-actin staining for the Latrunculin treated cells?

We have added several things to clarify this data. First, we have denoted the cell body in the latruculin treated cell shown in Figure 9J with an arrow. Indeed, there is clumping of actin and βCMII that occurs near the nucleus. Both networks are dense and chaotic and were not used for the quantification shown in Figure 9K. This is now stated in the figure legend. We have also added a new supplemental figure where we have stained actin, βCMII and DNA in a cell treated with latruculin as the reviewers suggest (Figure 9—figure supplement 3). We agree that there could be A-bands in the cell body. However, it would take higher resolution imaging than SIM to fully explore this possibility. Given that the experiment was simply performed to further test if disrupting actin filament organization also disrupted βCMII filament organization, we feel that further characterization of the clumping in the cell body is beyond the scope of this study.

6) Figure 10. The authors' demonstration of NMIIA or NMIIB co-assembly in bipolar filaments with betaCMII in the cultured hiCMs is convincing. However, the reviewers were still not completely convinced by the claim that they can find these co-assembled filaments in vivo in the mouse heart. The heart tissue sections in Figure S3B and Figure S18 are a low mag field of many myofibrils (not sure why Figure S18 is a separate figure, also no scale bar in Figure S3B). Both show that the NMIIB staining (green) blobs are excluded from the myofibrils and squeezed in between the closely packed myofibrils. Are these supposed to be periodic NMIIB filaments located along an MSF type of structure? Or aggregates? One would like to see a zoom in of the blobs region so we can see the NMIIB and betaCM colocalized (along the putative MSFs?) in the context of the heart tissue, ie, we need an intermediate magnification field within which one can find the filaments! Not just the super-resolution images of individual filaments cropped out, as shown.

Thank you for these comments. We have now added several arrows in, what is now Figure 12B, showing NMIIB localizations that are close to βCMII. As this is a confocal image, we have now added an intermediate SIM image of a human septal muscle along with 4 high mag insets (Figure 12C). A scale bar has been added to Figure S12B.

7) Figure 9—figure supplement 3A. The betaCMII in the siRNA MYH9-treated cells looks remarkably OK. One can see lots of A bands. However, they appear to be lined up along the length of myofbrils that are very thin so that the betaCMII stacks are not evident. It would be nice to show the F-actin colocalization with the betaCMII staining. This also fits with the data in Figure 8 showing that sarcomere formation in MYH9 KD cells is not as disrupted as in MYH10 KD cells (see point 2 above). Presumably the fourier transform does not reveal the periodicity as well as for the controls in Figure 10G, due to the wispy myofibrils and reduced alignment of the stacks. But a line scan along the length of the myofibrils would likely still reveal periodicity. We are not asking for authors to do new experiments here (unless they want to!), rather just tone down interpretation of MYH9 KD images in Figure 9—figure supplement 3, as related to comments in point 2 above about NMIIA function.

Thank you for this comment. Indeed, there are organized A-bands in MYH9 KD condition. We have changed the title of the Results section to: “NMIIB and FHOD3 are required for organized A-band formation”

And changed the text in the Results section to:

“As FHOD3 KD also resulted in disorganized actin filament architecture, we localized βCMII in this condition. Indeed, FHOD3 KD hiCM had disorganized βCMII filaments compared to control hiCMs (Figure 9—figure supplement 4).”